# TGFβ1+CCR5+ neutrophil subset increases in bone marrow and causes age-related osteoporosis in male mice

Jinbo Li [1,7] ✉, Zhenqiang Yao [1,2], Xin Liu[3], Rong Duan[1], Xiangjiao Yi[4], Akram Ayoub[1,5], James O. Sanders [6,8], Addisu Mesfin [6], Lianping Xing [1,2] & Brendan F. Boyce [1,2,6] ✉

TGFβ1 induces age-related bone loss by promoting degradation of TNF receptor-associated factor 3 (TRAF3), levels of which decrease in murine and human bone during aging. We report that a subset of neutrophils (TGFβ1+CCR5+) is the major source of TGFβ1 in murine bone. Their numbers are increased in bone marrow (BM) of aged wild-type mice and adult mice with TRAF3 conditionally deleted in mesenchymal progenitor cells (MPCs), associated with increased expression in BM of the chemokine, CCL5, suggesting that TRAF3 in MPCs limits TGFβ1+CCR5+ neutrophil numbers in BM of young mice. During aging, TGFβ1-induced TRAF3 degradation in MPCs promotes NF-κB-mediated expression of CCL5 by MPCs, associated with higher TGFβ1+CCR5+ neutrophil numbers in BM where they induce bone loss. TGFβ1+CCR5+ neutrophils decreased bone mass in male mice. The FDA-approved CCR5 antagonist, maraviroc, reduced TGFβ1+CCR5+ neutrophil numbers in BM and increased bone mass in aged mice. 15-mon-old mice with TGFβRII specifically deleted in MPCs had lower numbers of TGFβ1+CCR5+ neutrophils in BM and higher bone volume than wild-type littermates. We propose that pharmacologic reduction of TGFβ1+CCR5+ neutrophil numbers in BM could treat or prevent age-related osteoporosis.

Low-level chronic inflammation occurring in the absence of overt infection during aging, termed inflammaging[1], is common and implicated in the pathogenesis of cancer[2], atherosclerosis[3], diabetes[3], Alzheimer's[3], arthritis[4], sarcopenia[5], and osteoporosis[6]. Osteoporosis, the most common bone disease in aging humans, is characterized by decreased bone mass and strength due to increased osteoclastic bone resorption and decreased osteoblastic bone formation, leading to increased fracture risk, morbidity and mortality[7]. Treatment includes

anti-resorptive and anabolic drugs[8,9]; the former have side effects that now limit patients' willingness to take them[10], and the latter can be given for only 2 years to patients with severe osteoporosis[11,12], after which bone loss accelerates[13]. Thus, there a growing need to develop new treatments, based on better understanding of the molecular mechanisms causing bone loss.

We reported that during aging: RANKL, an essential osteoclastogenic cytokine[14], stimulates bone resorption in part by degrading TNF

[1]Department of Pathology and Laboratory Medicine, University of Rochester Medical Center, Rochester, NY 14642, USA. [2]Center for Musculoskeletal Research, University of Rochester Medical Center, Rochester, NY 14642, USA. [3]Department of Orthopedics, Tianjin Hospital, Tianjin, China. [4]The First Affiliated Hospital of Anhui University of Chinese Medicine, Hefei, Anhui, China. [5]Leica Biosystems, Deer Park, IL 60010, USA. [6]Department of Orthopaedics and Rehabilitation Medicine, University of Rochester Medical Center, Rochester, NY 14642, USA. [7]Present address: Institute of Health and Medical Research, Hebei Medical University, Shijiazhuang, Hebei 050017, China. [8]Present address: Department of Orthopaedics, University of North Carolina, Chapel Hill, NC 27514, USA. ✉e-mail: Jinbo_Li@hebmu.edu.cn; brendan_boyce@urmc.rochester.edu

receptor-associated factor 3 (TRAF3) and thus activating NF-κB signaling in osteoclast precursors to enhance osteoclastogenesis[15]; and TGFβ1, a cytokine released from bone during resorption[16], inhibits bone formation[16] by activating NF-κB, a master regulator of inflammatory responses[17] and bone remodeling[18], in mesenchymal progenitor cells (MPCs). Both cytokines mediate signaling in cells through TRAFs, which can positively and negatively regulate signaling[19,20]. TRAF3 typically inhibits NF-κB by promoting degradation of NF-κB-inducing kinase[21]. TRAF3 limits RANKL-induced osteoclastogenesis by suppressing NF-κB signaling in osteoclast precursors[15], and restricts TGFβ1-induced inhibition of osteoblast differentiation from MPCs by inhibiting GSK-3β-mediated degradation of β-catenin[16], expression of which is essential for bone formation[22]. TRAF3 protein levels are reduced in bone from aged humans and mice, associated with increased levels of active TGFβ1 in BM[16]. Mice with TRAF3 conditional knockout (cKO) in osteoclast precursors develop early onset osteoporosis because of excessive bone resorption[15], while mice with TRAF3 cKO in MPCs develop osteoporosis due to reduced bone formation and increased resorption[16]. By degrading TRAF3 in MPCs, TGFβ1 promotes increased resorption and decreased bone formation[15,16], implicating it in the pathogenesis of low-level chronic inflammation-related osteoporosis. However, there are small numbers of MPCs in bone marrow (BM) of normal mice, and bone resorption decreases in mice and humans during aging[23], resulting in less TGFβ1 being released from bone in aged mice, suggesting that there is another more common source(s) of TGFβ1 in BM causing osteoporosis.

Inflammaging is accompanied by alterations in immune cells, in the microenvironment in lymphoid/non-lymphoid tissues, and in chemokines and cytokines that mediate interactions between immune cells and the microenvironment[24]. However, it remains largely unknown if age-related alterations of immune cells in BM cause bone loss during aging. Hematopoietic stem cells (HSCs) continuously regenerate immune and blood cells and expand numerically in BM during aging, when they commit preferentially to myeloid over lymphoid differentiation[25], leading to impaired adaptive immune responses and increased susceptibility to systemic infection[26]. Leukocytes are the most abundant cell type in BM and are heterogenous. As the most abundant sub-population of leukocytes, neutrophils express TGFβ1 in tumor cells[27], normal intestinal cells[28], and in respiratory cells in asthmatic subjects[29], but whether TGFβ1-expressing neutrophils or other immune cells are involved in inflammaging and age-related osteoporosis remains unclear.

Despite these advances, the major sources of TGFβ1 in inflammaging-induced osteoporosis remain unknown. Here, we report that TGFβ1+CCR5+ neutrophils (which we call TCNs), an immune cell subset, are the major cellular source of TGFβ1 in BM. Their numbers are increased in BM of aged mice, associated with increased expression by MPCs of CCL5 because of TGFβ1-induced degradation of TRAF3 in MPCs. Injection of TCNs into male NGS mice significantly reduced bone mass. Aged mice given the FDA-approved CCR5 inhibitor, maraviroc, and mice lacking TGFβRII in MPCs have reduced numbers of TCNs in BM and increased bone mass, supporting our posit that these neutrophils cause age-related bone loss.

## Results

### Neutrophils are the major cellular source of TGFβ1 in BM from aged mice

To examine potential sources of TGFβ1 in the bone/BM microenvironment of aged mice, we measured TGFβ1 protein levels in bone, BM cells and BM plasma, and found that they were higher in BM cells than in bone or BM plasma from 2 to 4-mon-old male mice, and that they were significantly higher in BM cells from 18 to 22-mon-old male mice (Fig. 1a). TGFβ1 is expressed in the cytosol and on the surface of many cell types[30,31]. Thus, we next assessed cell surface and cytoplasmic expression of TGFβ1 in BM cells and found significantly more

TGFβ1+ cells in 22-mon-old than 4-mon-old male mice, and that significantly more of them had cytoplasmic than cell surface expression of TGFβ1 (Fig. 1b). To further characterize these TGFβ1+ BM cells, we examined TGFβ1 expression in non-hematopoietic cells (CD45-), MPCs (CD45-Sca1+), myeloid (CD11b+Gr1+), B (B220+), T (CD3e+), and dendritic (CD11c+) cells. We found that myeloid cells were the most abundant TGFβ1+ BM cells and comprised ~80% of the total TGFβ1+ BM cells (Fig. 1c; Suppl. Fig. 1). We next compared Ly6G and Ly6C expression by these TGFβ1+CD11b+ myeloid cells (Fig. 1d, e) and found that the percentage of Ly6C-6G+ cells (granulocyte subset) was higher and the percentage of Ly6Chi6G- cells (monocyte subset) was lower in cells from aged than from young mice (Fig. 1f, g). In addition, aged male mice had significantly more TGFβ1+CD11b+Ly6C-6G+ granulocytic cells in BM than young mice (Fig. 1h, i), and significantly more of these cells had intracellular than surface TGFβ1 expression (Fig. 1h, i). We found that TGFβ1+ cells comprised ~12% of total BM Ly6C-6G+ granulocytic cells in 2-mon-old young mice, that this fraction increased significantly to ~16% in aged male mice, and that most TGFβ1+ cells had high expression of Ly6G, a marker of neutrophil differentiation (Suppl. Fig. 2). In contrast with BM, the percentages of TGFβ1+CD11b+Ly6C-6G+ cells were significantly lower in peripheral blood, mesenteric lymph node and spleen from aged than from young mice (Fig. 1j). FACS-sorted, cytospun, and H&E-stained TGFβ1+CD11b+Ly6C-6G+ cells had multilobed nuclei, typical of neutrophils (Fig. 1k).

### Neutrophils inhibit osteoblast differentiation from MPCs and promote osteoclast formation through TGFβ

To determine the effects of neutrophils on osteoblast differentiation, we sorted Ly6G+ BM cells from young and aged male mouse BM using magnetic-activated cell sorting (MACS) and co-cultured them at different ratios with MPCs. We found that Ly6G+ cells from BM of young mice stimulated osteoblast differentiation at relatively low ratios (0.5:1 and 1:1), but inhibited osteoblast differentiation when the Ly6G+/MPC ratio was increased to 4:1 and 8:1 (Fig. 2a, b). Notably, the differentiation of MPCs was significantly lower when they were co-cultured with Ly6G+ cells from aged mice than from young male mice (Fig. 2a, b). Ly6G+ cells isolated from BM from aged mice using MACS had significantly higher active TGFβ1 levels than cells from young male mice (Fig. 2c).

To test if TGFβ-expressing neutrophils (TNs) regulate osteoblast differentiation directly and if this process functions through TGFβ, we sorted TNs (TGFβ1+CD11b+Ly6C-6G+) from 3- and 20-mon-old C57 male mice using FACS, and co-cultured them with MPCs from BM of 3-mon-old C57 mice. We found that TNs from aged mice more efficiently inhibited osteoblast differentiation than TNs from young mice, and that this inhibition was blocked by 1D11, a TGFβ neutralizing antibody (Fig. 2d). To test if this process functions through TGFβ signaling in MPCs, we generated mice with conditional KO of TGFβ receptor II in mesenchymal lineage cells by crossing TGFβRIIfl/fl with Prx1Cre mice (TGFβRIIfl/fl;Prx1Cre, which we call TRII-cKO mice). We found that TNs sorted from aged C57 mice effectively inhibited osteoblast differentiation from WT MPCs, but not from TRII-cKO MPCs (Fig. 2e). In addition, TNs from aged male mice stimulated osteoclast formation from progenitor cells with no changes in the ratio of Rankl/Opg transcription levels, and this was inhibited by 1D11 (Fig. 2f, g, Suppl. Fig. 1b). To determine the effects of TNs in vivo, we sorted BM TNs from 3- and 24-mon-old mTmG male mice using FACS, and gently mixed them with 3rd passage bone-derived MPCs from young TGFβRIIfl/fl (WT) or TRII-cKO mice, and we implanted the cells into the dorsal subcutaneous tissues of 3-mon-old male NSG mice. We found that osteoblast differentiation and new bone formation from WT bone-derived MPCs were significantly lower when they were co-implanted with TNs from aged than from young mice (Fig. 2h, i). Importantly, this inhibition of osteoblast differentiation caused by aged TNs was abolished when

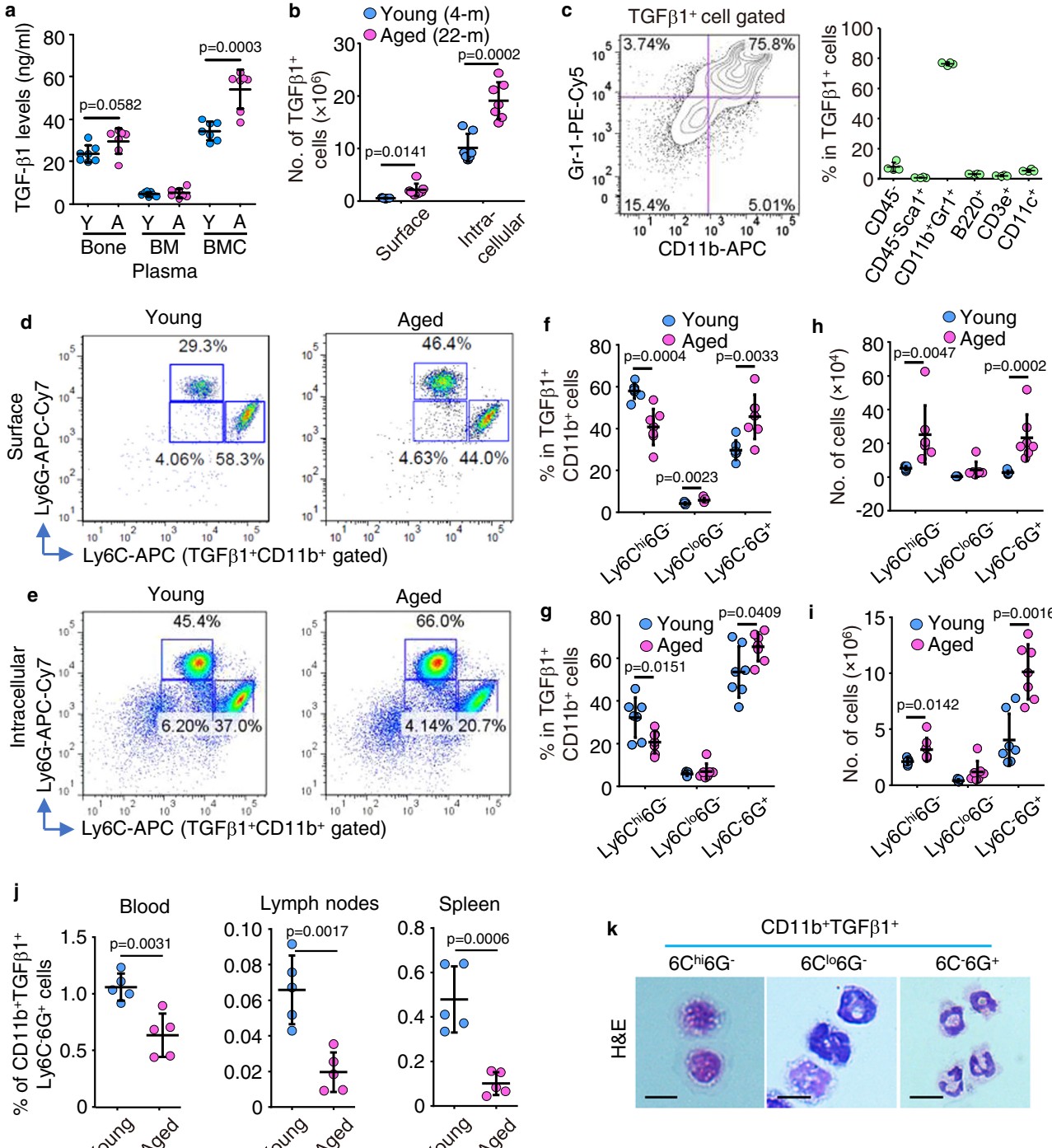

**Fig. 1 | Neutrophils are the major cellular source of TGFβ1 in BM from aged mice. a** Expression levels of TGFβ1 in protein lysates from bone (with BM flushed out), BM cells (BMC), and BM plasma from one leg from young (2–4-mon-old) and aged (18–22-mon-old) male C57 mice using ELISA. Mean±SD (*n* = 7 biologically independent male mice/group). **b** Surface and intracellular expression of TGFβ1 tested by FACS in BM cells from 4- and 22-mon-old C57 mice. Mean ± SD (*n* = 7 biologically independent male mice/group). **c** Enrichment of TGFβ1⁺ BM cells in CD11b⁺Gr1⁺ myeloid cells and percentages of various types of BM cells expressing TGFβ1 in BM. Mean ± SD (*n* = 4 biologically independent male mice). **d**, **e** FACS analysis of Ly6C⁺ and Ly6G⁺ subpopulations in TGFβ1⁺CD11b⁺ myeloid cells with

TGFβ1 surface and intracellular expression in BM from 4- and 22-mon-old C57 mice, and frequencies (**f**, **g**) and numbers (**h**, **i**) of monocytic (Ly6C^hi6G⁻), granulocytic (Ly6C⁻6G⁺) and intermediate-stage (Ly6C^lo6G⁻) cells in TGFβ1⁺CD11b⁺ myeloid cells in BM. Mean±SD (*n* = 7 biologically independent male mice/group). **j** Frequencies of TGFβ1⁺CD11b⁺Ly6C⁻6G⁺ cells in peripheral blood, mesenteric lymph nodes, and spleen. Mean ± SD (*n* = 5 biologically independent male mice/group). **k** H&E-stained Ly6C⁺ and Ly6G⁺ subpopulations from TGFβ1⁺CD11b⁺ myeloid cells following FACS and cyto-spinning. Bar = 10 µm. Analyses: Student's two-sided unpaired *t* test. Source data are provided as a Source data file.

they were co-implanted with bone-derived MPCs from TRII-cKO male mice, and these cells formed significantly more new bone (Fig. 2h, i). Osteoclastic bone resorption in this new bone was significantly higher when WT MPCs were co-implanted with TNs from

aged than from young male mice (Fig. 2j, k), while this stimulation of osteoclast formation caused by aged TNs was significantly reduced when the cells were co-implanted with bone-derived MPCs from TRII-cKO male mice (Fig. 2j, k).

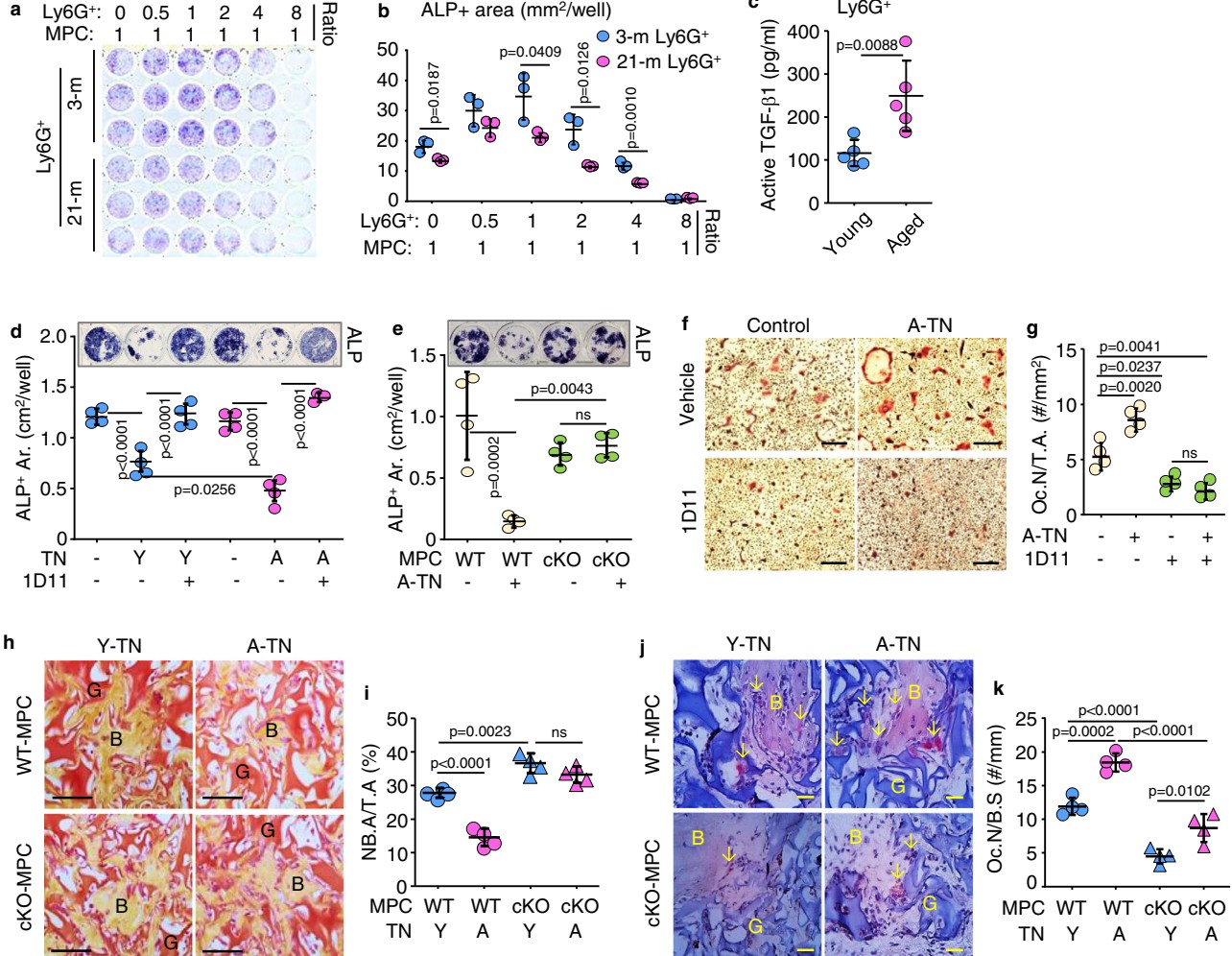

**Fig. 2 | Neutrophils inhibit osteoblast differentiation from MPCs and promote osteoclast formation through TGFβ. a** Mesenchymal progenitor cells (MPCs) co-cultured with Ly6G⁺ BM cells from 3- and 21-mon-old mice for 4 d and **b** ALP⁺ OB area. Mean ± SD (*n* = 3 biologically independent samples/group). **c** Active TGFβ1 protein levels in lysates of Ly6G⁺ BM cells magnetically-isolated from 4- and 22-mon-old mice tested using ELISA. Mean ± SD (*n* = 5 biologically independent male mice/group). **d** MPCs from 3-mon-old C57 mice co-cultured with TGFβ1⁺ neutrophils (TNs) sorted from BM of 3- and 20-mon-old C57 mice and treated with vehicle or the TGFβ neutralizing Ab, 1D11. ALP staining performed for osteoblast measurement. Mean±SD (*n* = 4 biologically independent samples/group). **e** MPCs from TGFβRII^fl/fl (WT) and Prx1^Cre;TGFβRII^fl/fl (cKO) male mice co-cultured with TNs sorted from 20-mon-old C57 mice (A-TN), and ALP⁺ stained osteoblasts. Mean ± SD (*n* = 4 biologically independent samples/group). **f, g** Osteoclast precursors from

3-mon-old C57 mice co-cultured with TNs from 20-mon-old mice (A-TN) plus addition of vehicle or 1D11, and TRAP-stained. Bar = 100 µm. Mean ± SD (*n* = 4 biologically independent samples/group). **h–k** MPCs from TGFβRII^fl/fl (WT) and Prx1^Cre;TGFβRII^fl/fl (cKO) male mice were implanted subcutaneously into NSG mice along with TNs sorted from 3- (Y) and 24-mon-old (A) mTmG mice. Implants were harvested 1 month later, processed through paraffin, and H&E- and TRAP-stained. G: implanted GelFoam (red). B: newly-formed bone (yellow). **h, i** Newly generated bone area. Bar = 50 µm. **j, k** Osteoclast (red) number in newly generated bone. G: implanted GelFoam (blue). B: newly-formed bone (light pink). Bar = 25 µm. Mean ± SD (*n* = 4 biologically independent male mice/group). Analyses: Student's two-sided unpaired *t* test in (**c**); one-way ANOVA with Tukey's post hoc test in all others. Source data are provided as a Source data file.

## NSG mice injected with TGFβ1-expressing neutrophils develop osteoporosis

To examine if TNs have systemic effects on bone homeostasis, we flow sorted TNs from BM of 3- and 20-mon-old male C57 male mice and confirmed that they have high expression of TGFβ, with MFI levels being 4.3-fold higher than in non-enriched neutrophils (Suppl. Fig. 3a–d) and injected them into 3-mon-old NSG male mice via tail vein. We found that NSG recipients injected with either young or aged TNs developed significant bone loss (Fig. 3a). Of note, NSG recipients injected with aged TNs had significantly lower trabecular bone volume (Fig. 3b) and trabecular number (Fig. 3c) and increased trabecular separation (Fig. 3d) than NSG recipients injected with young TNs. However, no significant changes were observed in trabecular or cortical bone thickness (Suppl. Fig. 3e, f). Consistent with this, NSG recipients injected with either young or aged TNs had significantly fewer

osteoblasts and more osteoclasts on trabecular bone surfaces than NSG mice without TN injection (Fig. 3e–h). NSG recipients injected with aged TNs had significantly more osteoclasts than recipients injected with young TNs (Fig. 3g, h).

## Mice with TRAF3 deleted in MPCs have increased numbers of TGFβ-expressing neutrophils in BM

TRAF3 negatively regulates NF-κB signaling, and levels of TRAF3 fall in human and murine bone during aging[16]. Consistent with this, protein levels of NF-κB p52 and RelB were higher in bone from old than from young male mice (Fig. 4a) as a result of NF-κB-inducing kinase-mediated processing of p100 to p52. Mice with TRAF3 deleted specifically in mesenchymal lineage cells (Prx1^Cre;TRAF3^fl/fl; called P-cKO) develop early onset osteoporosis by 9-mon-old[16]. This persisted in 12-mon-old P-cKO mice (Fig. 4b, c) and was

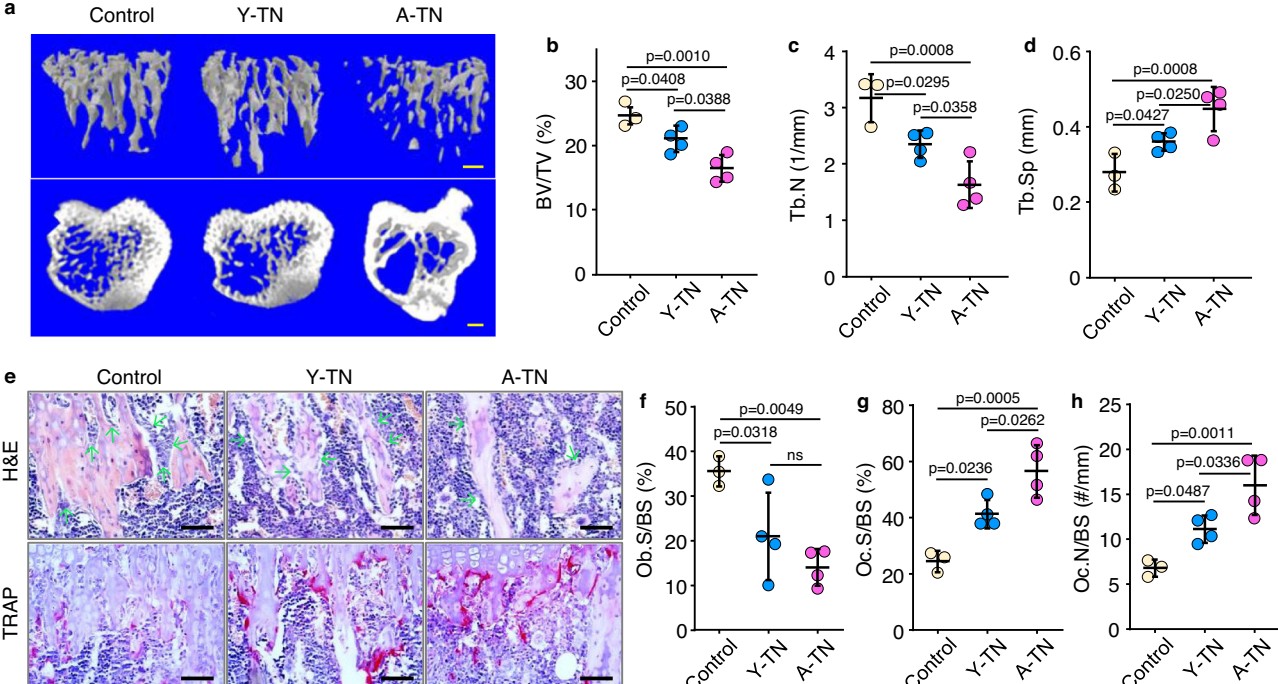

**Fig. 3 | NSG mice injected with TGFβ1-expressing neutrophils (TNs) develop osteoporosis. a** μCT 3D reconstruction of trabecular bone and coronal sections of tibial metaphyses of NSG mice injected with PBS (control), TNs sorted from young (Y-TN) or from aged mice (A-TN). Bar = 1 mm. **b**–**d** Microstructural parameters, including trabecular bone volume (BV/TV) (**b**), number (Tb.N) (**c**), and separation (Tb.Sp) (**d**) in tibial metaphyses. Mean ± SD (*n* = 3, 4, and 4 biologically independent male recipient mice for control, Y- and A-TN groups, respectively).

**e** Histomorphometric analysis of tibial metaphyseal bone of NSG recipients, and H&E and TRAP staining performed for osteoblast and osteoclast counting, respectively. Bar = 50 μm. **f**–**h** Osteoblast surfaces (Ob.S/BS) (**f**), osteoclast surfaces (Oc.S/BS) (**g**) and numbers (Oc.N/BS) (**h**) on metaphyseal trabecular bone surfaces. Mean ± SD (*n* = 3, 4, and 4 biologically independent male recipient mice for control, Y- and A-TN groups, respectively). Analyses: one-way ANOVA with Tukey's post hoc test. Source data are provided as a Source data file.

due to reduced bone formation (Fig. 4d) and increased bone resorption (Fig. 4e). In addition, the percentage and numbers of TGFβ-expressing neutrophils were significantly higher in BM from 12-mon-old P-cKO mice than from WT male mice (Fig. 4f, g). Thus, similar to aged WT mice, 12-mon-old P-cKO mice have increased TGFβ-expressing neutrophils in BM. To examine the specificity of targeted gene deletion driven by Prx1Cre, we crossed Prx1^Cre with Rosa26^mTmG reporter mice, in which Prx1^Cre+ cells switch tdTomato to GFP protein expression. We found that Prx1^Cre+ GFP cells were very rare in skeletal muscle, but widely distributed in bone cells, including in hypertrophic chondrocytes in growth plates, mesenchymal/osteoblastic cells on metaphyseal bone surfaces, and a relatively small fraction of osteocytes and spindle cells in BM (Suppl. Fig. 4a–c), which is consistent with the profiles of Prx1-tracing mesenchymal lineage cells reported by others[32]. In addition, osteoblastic cells on surfaces of newly generated bone surfaces were GFP-positive in fracture callus of 3-mon-old Prx1^Cre;Rosa26^mTmG male mice on day 14 post-fracture, consistent with the osteogenic potential of the Prx1^Cre-tracing cells in our mouse model (Suppl. Fig. 4d). We also found significantly higher Cre transcription levels (~50–700×) in vertebrae of Prx1^Cre mice than in WT mice from 2 wk to 12-mon of age (Suppl. Fig. 4e). Many Prx1^Cre+ GFP cells were present on endosteal and trabecular bones surfaces of 3-mon-old Prx1^Cre;Rosa26^mTmG mice (Suppl. Fig. 4f). We also found that Prx1^Cre drove *Traf3* gene deletion in MPCs, but not in B cells, T cells, leukocytes, or osteoclast precursor cells, in BM of 15-mon-old P-cKO mice (Suppl. Fig. 5). These findings verify the specificity of Prx1-driven gene deletion in mesenchymal lineage cells and suggest that TRAF3 in mesenchymal lineage cells limits the numbers of TGFβ1+ neutrophils in BM and bone loss in adult mice.

## Aged and P-cKO mice have MPCs with increased CCL5 expression and increased numbers of TGFβ-expressing neutrophils in BM

Various hematopoietic/immune cells are attracted to or maintained in BM by signaling through chemokine/chemokine receptors whose expression is regulated by transcription factors, particularly NF-κB[33]. NF-κB signaling is increased in MPCs of aged WT mice and adult TRAF3 P-cKO mice[16]. Data in Fig. 1j suggest that increased numbers of TGFβ-expressing neutrophils could be recruited from blood or peripheral lymphoid organs to BM during aging. To explore potential NF-κB-induced chemotactic signals shared by mesenchymal lineage cells from adult P-cKO and aged WT mice to promote TN accumulation in BM, we isolated CD45- cells from BM of adult (9-mon-old) WT and P-cKO mice and aged (24-mon-old) WT male mice and examined the expression levels of 38 chemokines by qPCR. We found higher mRNA expression levels of 4 chemokines (*Ccl12*, *Ccl5*, *Cxcl12*, and *Ccl24*) and lower expression of 3 other chemokines (*Cxcl4*, *Ccl2*, and *Ccl26*) in the CD45- cells from both adult P-cKO and aged WT mice than from adult WT mice (Fig. 5a). We then performed an antibody array, which included 40 different cytokines and chemokines to determine if protein levels of these chemokines were altered. We found that CCL5 protein levels were higher in BM protein lysates from old than from young mice (Fig. 5b; Suppl. Fig. 6a), and they were also higher in BM from 9-mon-old P-cKO mice than from WT littermates (Fig. 5c; Suppl. Fig. 6a). However, protein levels of CCL12, CXCL12 and CCL2 were very low, and no obvious changes were observed (Suppl. Fig. 6a; CCL24, CXCL4, and CCL26 were not included in the array). We next performed an ELISA assay and confirmed that CCL12, CXCL12, and CCL2 protein levels were comparable between 8-mon-old WT and P-cKO mice. Protein levels of CCL12, but not CXCL12 or CCL2, were significantly higher in BM of 18-mon-old mice than 6-mon-old male mice (Suppl. Fig. 6b, c).

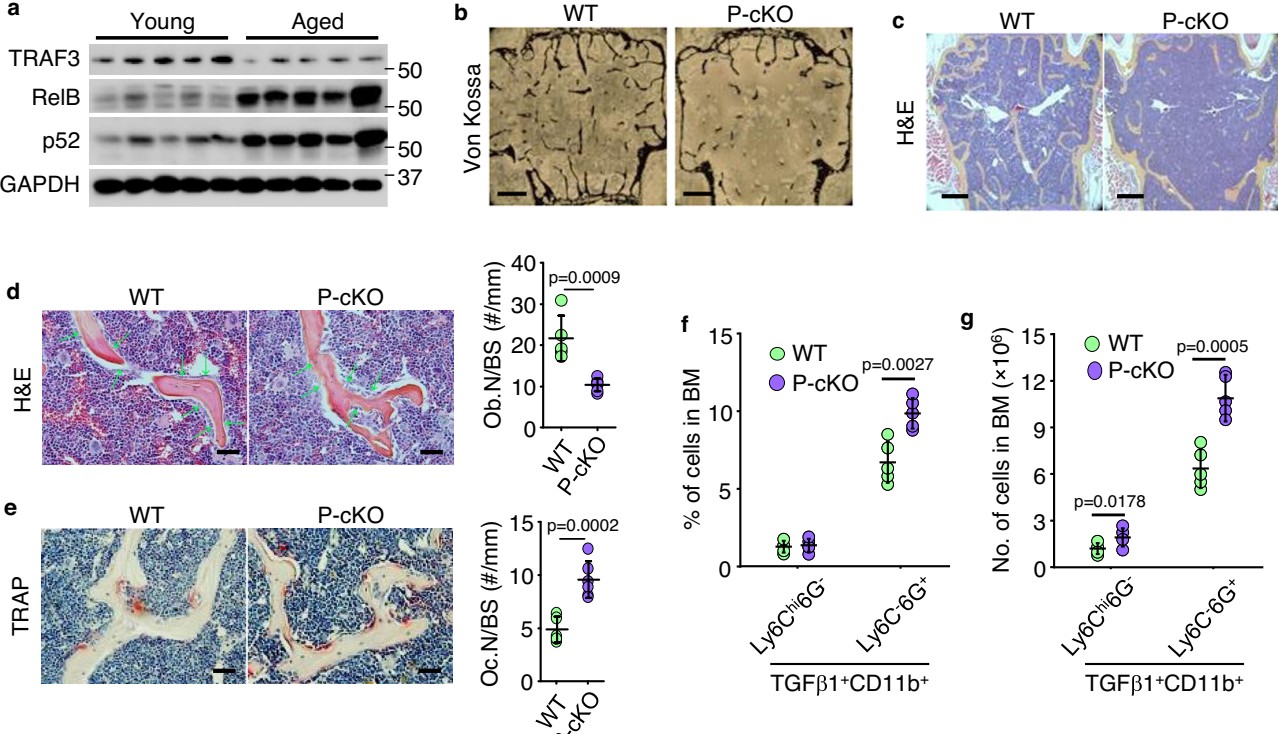

**Fig. 4 | Mice with TRAF3 deleted in MPCs have increased numbers of TGFβ-expressing neutrophils in BM. a** WBs of TRAF3, RelB, p52, and GAPDH expression in total protein lysate extracted from leg bones (with BM) from 3- (young) and 18-mon-old (aged) C57 mice. **b** von-Kossa-stained plastic sections of L1 vertebrae from 12-mon-old Prx1^Cre;TRAF3^fl/fl (P-cKO) mice and WT littermates. Bar = 400 μm. **c** H&E-stained paraffin sections of L2 vertebrae from 12-mon-old WT and P-cKO mice, and **d** OB surfaces (Ob.S/B.S). Bar = 400 μm in (**c**) and 50 μm in (**d**). Mean ± SD (*n* = 8 biologically independent male mice/group). **e** TRAP-stained paraffin sections of L2 vertebrae and OC numbers (Oc.N/B.S). Bar = 50 μm. Mean ± SD (*n* = 8 biologically independent male mice/group). **f, g** Frequencies and numbers of TGFβ1^+CD11b^+Ly6C^hi6G^- and TGFβ1^+CD11b^+Ly6C^-6G^+ in BM from 15-mon-old WT and P-cKO mice. Mean ± SD (*n* = 5 biologically independent male mice/group). Analyses: Student's two-sided unpaired *t* test. Source data are provided as a Source data file.

Notably, *Ccl5* mRNA levels were higher in MPCs from aged than from young mice (Fig. 5d), and they were also higher in human vertebral specimens from elderly subjects than from children (Fig. 5e).

To determine if NF-κB regulates *Ccl5* expression in MPCs, potential κB binding sites on the promotor of the *Ccl5* gene were identified and confirmed using ChIP assays. We then identified 3 κB binding sites within 500 bp from the start codon in the *Ccl5* promoter (Suppl. Fig. 7a). ChIP assay data supported higher NF-κB RelA and RelB binding to these sites in the *Ccl5* promotor in MPCs from 9-mon-old P-cKO mice than from WT mice of the same age (Fig. 5f, g). In addition, *Ccl5* mRNA levels were significantly higher in MPCs infected with pMX-RelA or -RelB lentiviruses than cells infected with pMX-GFP lentiviruses (Suppl. Fig. 7b).

To examine if there are differences in cell migration of TNs from young and aged mice, we performed a cell migration assay using chemokines. We found that, in response to CCL4, CCL5, and CXCL12, both neutrophils and TNs from aged C57 mice were attracted at a similar speed as cells from young C57 mice (Fig. 5h; Suppl. Fig. 7c). In contrast, in response to CCL4 and CCL5, but not to CXCL12, TNs from both young and aged male mice were attracted faster than vehicle-treated cells (Fig. 5h). We next tested the expression levels of CCR5, a co-receptor of CCL5 and CCL4, by various Ly6C and 6G subsets of TGFβ1^+ myeloid cells, including TNs. We found that CCR5 expression by aged TNs was comparable to that in young TNs (Suppl. Fig. 7d), and that CCR5 expression was significantly higher in TGFβ1^+ neutrophils (Ly6C^-6G^+) than in TGFβ1^+ monocytes (Ly6C^hi6G^-) (Fig. 5i, j; Suppl. Fig. 8a), which were characterized by higher expression of CSF-1R (Fig. 5k; Suppl. Fig. 8b). CCR5 expression by TGFβ1^- neutrophils was also higher than TGFβ1^- monocytes, but still much lower than TGFβ1^+

neutrophils (Suppl. Fig. 9). Hence, we called this newly identified neutrophil subset TGFβ1^+CCR5^+ neutrophils (TCNs for short).

To determine if CCR5 expression by TNs facilitated their recruitment toward MPCs in P-cKO mice, we next performed transwell experiments and found that MPCs from P-cKO mice attracted significantly more TNs than MPCs from WT mice. Of note, the increased attraction induced by P-cKO MPCs was efficiently blocked by addition of maraviroc, an FDA-approved CCR5 antagonist (Fig. 5l, m). These findings suggest that TCNs could be recruited into BM in response to increased CCL5 production by MPCs in P-cKO mice and aged mice.

**Maraviroc prevents bone loss caused by TCNs from aged mice**

To determine the effects of maraviroc on TCN recruitment into BM and bone homeostasis, we sorted TCNs from BM of aged ROSA^mTmG male mice and confirmed their high expression of TGFβ1 (Suppl. Fig. 10), We injected these sorted TCNs into male NSG mice and treated the mice with vehicle or maraviroc. We found that NSG mice treated with maraviroc had significantly fewer donor-derived TCNs in BM than vehicle-treated mice (Suppl. Fig. 11a–d), which was not associated with altered Ki-67-related cell proliferation (Suppl. Fig. 11e, f), or p16-related cell senescence (Suppl. Fig. 11g, h). Of note, maraviroc-treated recipients had significantly higher trabecular bone volume (BV/TV; Fig. 6a, b) and trabecular number (Tb.N; Fig. 6c), lower trabecular separation (Tb.Sp; Fig. 6d), and higher bone mineral density (BMD; Fig. 6e) than vehicle-treated recipients. However, no significant changes were observed in the thickness of trabecular or cortical bone between NSG recipients treated with vehicle or maraviroc (Suppl. Fig. 11i, j). Histomorphometric analysis showed that maraviroc-treated recipients had higher osteoblast surfaces (Fig. 6f, g) and lower osteoclast numbers and surfaces on

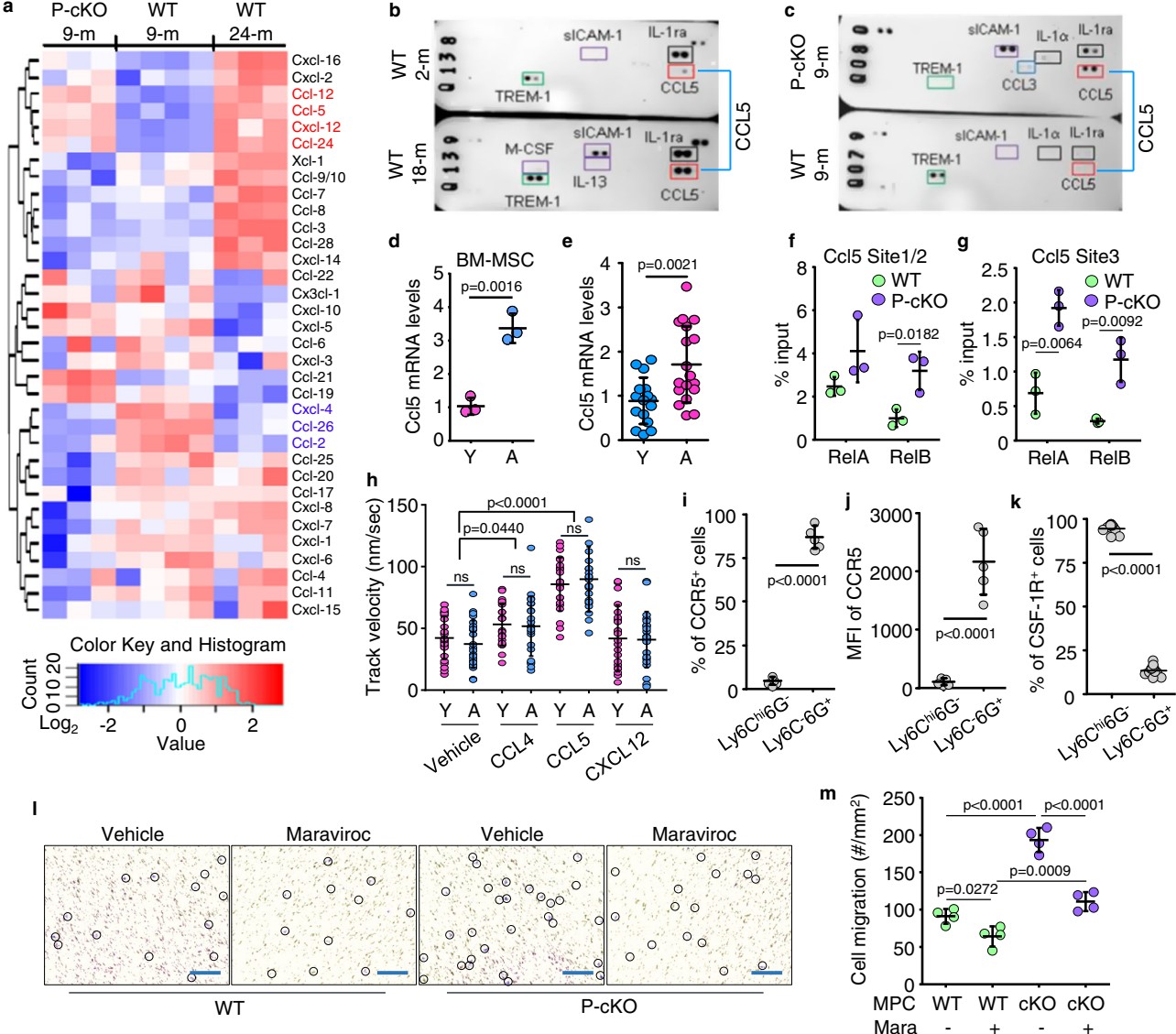

**Fig. 5 | Aged and P-cKO mice have MPCs with increased CCL5 expression and increased numbers of TCNs in BM. a** Heatmap of mouse chemokine levels tested using real-time qPCR in bulk RNA extracted from CD45- cells isolated from BM of 9-mon-old TRAF3fl/fl (WT) and Prx1CreTRAF3fl/fl (P-cKO) mice and 24-mon-old TRAF3fl/fl (WT) mice. $n = 4$, 3, and 3 biologically independent male mice, respectively. **b**, **c** Cytokine array of BM protein lysate from 2- and 18-mon-old male C57 mice (**b**) and from 9-mon-old WT and P-cKO male mice (**c**). **d** Relative levels of *Ccl5* mRNA in bulk RNA from 3rd passage BM-MSCs from 3- (young; Y) and 22-mon-old (aged; A) male C57 mice. Mean ± SD ($n = 3$ biologically independent samples/group). **e** Relative levels of *Ccl5* in bulk RNA from human vertebral specimens. Mean±SD ($n = 16$ young (Y) and 18 elderly (A) human subjects). **f**, **g** Sheared chromatin from WT and P-cKO BdMPCs used to perform DNA IP with RelA and RelB Abs or IgG control. RT-PCR using designed primers with putative κB binding sites in *Ccl5* gene promotors, normalized to input. Mean±SD ($n = 3$ biologically independent samples/group). **h** TNs sorted from BM of 3- (young) and 22-mon-old (aged) male C57 mice attracted by chemokines, including CCL4, CCL5 and CXCL12. Velocity of TN

migration. Mean ± SD ($n = 20$ biologically independent cells). **i** Frequencies of CCR5+ cells in Ly6Chi6G- and Ly6C-6G+ cells gated from TGFβ1+CD11b+ BM cells of 22-mon-old male C57 mice. Mean ± SD ($n = 5$ biologically independent male mice/group). **j** Mean fluorescence intensity (MFI) of CCR5 expression by Ly6Chi6G- and Ly6C-6G+ subpopulations in TGFβ1+CD11b+ BM cells from 22-mon-old male C57 mice. Mean ± SD ($n = 5$ biologically independent male mice/group). **k** Frequencies of CSF-1R+ cells in Ly6Chi6G- and Ly6C-6G+ cells gated from TGFβ1+CD11b+ BM cells of 22-mon-old male C57 mice. Mean ± SD ($n = 13$ biologically independent male mice/group). **l** Migration of TNs from 22-mon-old male C57 mice toward WT and P-cKO MPCs upon vehicle or maraviroc treatment tested using a transwell system. Cells maintained in the transwell membrane stained purple and manually marked with black line circles. Bar = 50 µm. **m** Cells in purple in transwell membrane. Mean ± SD ($n = 4$ biologically independent samples/group). Analyses: one-way ANOVA with Tukey's post hoc test in (**h**) and (**m**); Student's two-sided unpaired *t* test in all others. Source data are provided as a Source data file.

trabeculae in tibial metaphyses (Fig. 6f, h, i). These findings reveal that blockade of CCR5 by maraviroc efficiently prevented recruitment of donor-derived TCNs into BM and bone loss in NSG recipients.

## Aged mice treated with maraviroc have fewer TCNs in BM and increased bone mass

To determine if inhibition of CCR5 signaling prevents the accumulation of TCNs in BM, we treated 22-mon-old WT male mice with

maraviroc, once/day for one month. Maraviroc-treated aged mice had significantly lower percentage and numbers of TCNs in BM (Fig. 7a, b) than vehicle-treated mice, as well as higher values for trabecular bone volume and thickness (Fig. 7c–e), cortical bone thickness (Fig. 7f, g) and bone mineral density (Fig. 7h) in L1 vertebrae. Consistent with this, mineral apposition and bone formation rates and osteoblast surfaces (Fig. 7i–l) in L1 vertebrae and serum osteocalcin levels (Fig. 7m) were also higher in maraviroc- than vehicle-treated aged mice. In addition,

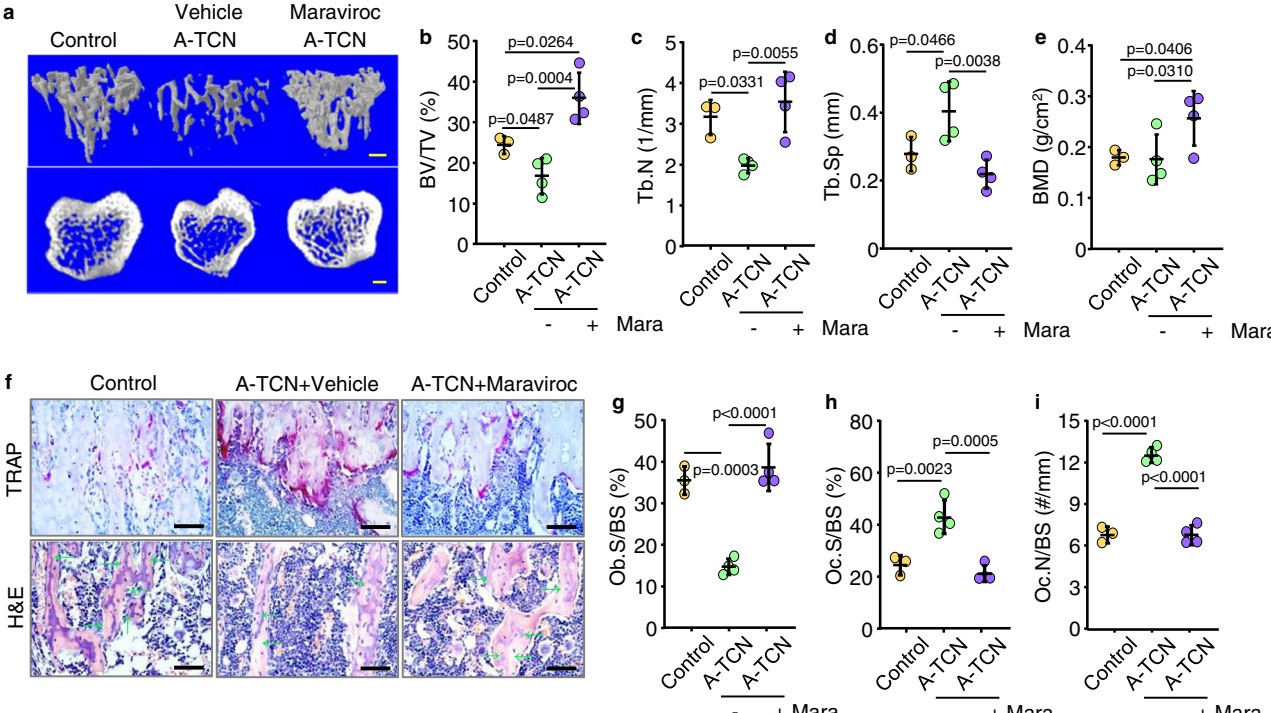

**Fig. 6 | Maraviroc prevents bone loss caused by aged TCNs. a** μCT 3D reconstruction of trabecular bone and coronal sections of tibial metaphyses of NSG mice injected with TCNs from aged C57 mice plus vehicle or maraviroc treatment. Bar = 1 mm. **b–d** Analysis of microstructure parameters in trabecular bones in (**a**), including trabecular bone volume (BV/TV) (**b**), number (Tb.N) (**c**) and separation (Tb.Sp) (**d**), and of bone mineral density (BMD) (**e**). Mean ± SD (n = 3, 4, and 4 biologically independent male mice for control, A-TCN and A-TCN plus maraviroc groups, respectively). **f** Representative images of TRAP- and H&E-stained tibial paraffin sections from NSG recipients as in (**a**) Bar = 50 μm. **g–i** Osteoblast surfaces (Ob.S/BS) (**g**), osteoclast surfaces (Oc.S/BS) (**h**), and numbers (Oc.N/BS) (**i**) on metaphyseal trabecular surfaces. Mean ± SD (n = 3, 4, and 4 biologically independent male mice, respectively). Analyses: one-way ANOVA with Tukey's post hoc test. Source data are provided as a Source data file.

aged mice treated with maraviroc had significantly lower osteoclast numbers and surfaces (Fig. 7n–p) and lower TRACP-5b levels in serum (Fig. 7q) than vehicle-treated mice. In contrast, maraviroc had no significant direct effects on osteoblast or osteoclast formation in vitro (Suppl. Fig. 12a, b), suggesting that its effects on osteoclast and osteoblast numbers in vivo are mediated by other cell types.

## Mice with TGFβRII deleted specifically in mesenchymal lineage cells have decreased numbers of TCNs in BM and increased bone mass

To determine if deletion of TGFβRII specifically in MPCs affects the age-related reduction in TRAF3 protein levels, TCN numbers in BM, and bone mass in mice, we generated TβRII$^{fl/fl}$Prx1$^{Cre}$ (TRII-cKO) mice and sacrificed them at 15 months of age. We found that TRAF3 protein levels in bone from 15-mon-old TRII-cKO male and female mice were similar to those in young WT mice (Fig. 8a) and that TGFβ1 did not induce transcription of *Ccl5* in TRII-cKO mouse MPCs (Suppl. Fig. 13a). Of note, the percentage (Fig. 8b) and numbers (Fig. 8c) of TCNs were significantly lower in BM from 15-mon-old TRII-cKO mice than from WT littermates, while the numbers of monocytic cells were similar (Fig. 8d). In addition, 15-mon-old TRII-cKO mice had significantly higher values for trabecular bone volume, number, and thickness, and reduced trabecular bone separation in L1 vertebrae (Fig. 8e, f) and femora (Fig. 8g, h) than male and female WT littermates. This increase in bone volume extended into the femoral diaphyses, including up to 4 mm from the growth plate where trabecular bone volumes were 9-fold higher than in WT littermates (BV/TV: 57 ± 12% vs. 6.2 ± 4.9%). Cortical bone thickness of L1 vertebrae (Fig. 8i) and femora (Fig. 8l) was significantly higher in TRII-cKO mice than WT littermates (Fig. 8k, n) and BMD of L1 vertebrae was significantly higher in TRII-cKO mice than WT littermates (Fig. 8j, m). In addition, mineral apposition rates

(Fig. 8o), bone formation rates (Fig. 8p), and osteoblast surfaces in trabecular bone of L1 vertebrae (Fig. 8q; Suppl. Fig. 13b) and serum osteocalcin levels (Fig. 8r) were markedly higher in these mice than in 15-mon-old WT mice. Of note, osteoclast surfaces and numbers in L1 vertebrae were significantly lower in TRII-cKO than WT mice (Fig. 8s, t; Suppl. Fig. 13c); consistent with this, serum TRACP-5b levels were lower in TRII-cKO mice than in WT littermates (Fig. 8u).

## Discussion

We have identified a subset of TGFβ/CCR5-expressing neutrophils (TCNs) that are increased in BM and decreased in blood, lymph nodes, and spleens of 18–22-mon-old WT mice. TCNs are also increased in 12-mon-old P-cKO mice, in which MPCs lack TRAF3 expression, associated with reduced bone mass. In contrast, 15-mon-old TRII-cKO mice, which lack TGFβ receptor II, have low numbers of TCNs in BM and high bone mass. Importantly, TCNs from aged male mice inhibited bone formation in vivo when they were injected into NSG mice or implanted into NSG mice along with MPCs from WT, but not from TRII-cKO mice, implicating TCNs as a cause of age-related bone loss through TGFβ. We propose that increased amounts of TGFβ1 released by TCNs during aging induce TRAF3 degradation in BM MPCs through TGFβ receptor II signaling, causing MPCs to increase their production of CCL5 through enhanced NF-κB signaling; in turn, increased CCL5 levels in BM are associated with increased numbers of TCNs in BM, and thus more TGFβ1 is released in the BM microenvironment. We also propose that in young and adult male mice, TGFβ levels and TCN numbers are physiological, which allows TRAF3 in MPCs to restrict NF-κB-mediated production of RANKL and CCL5 to levels that limit osteoclast formation and TCN numbers in BM, respectively. This holds bone resorption and formation at optimal levels to maintain skeletal integrity in young and adult mice. In contrast, TGFβ1 released by increased numbers of

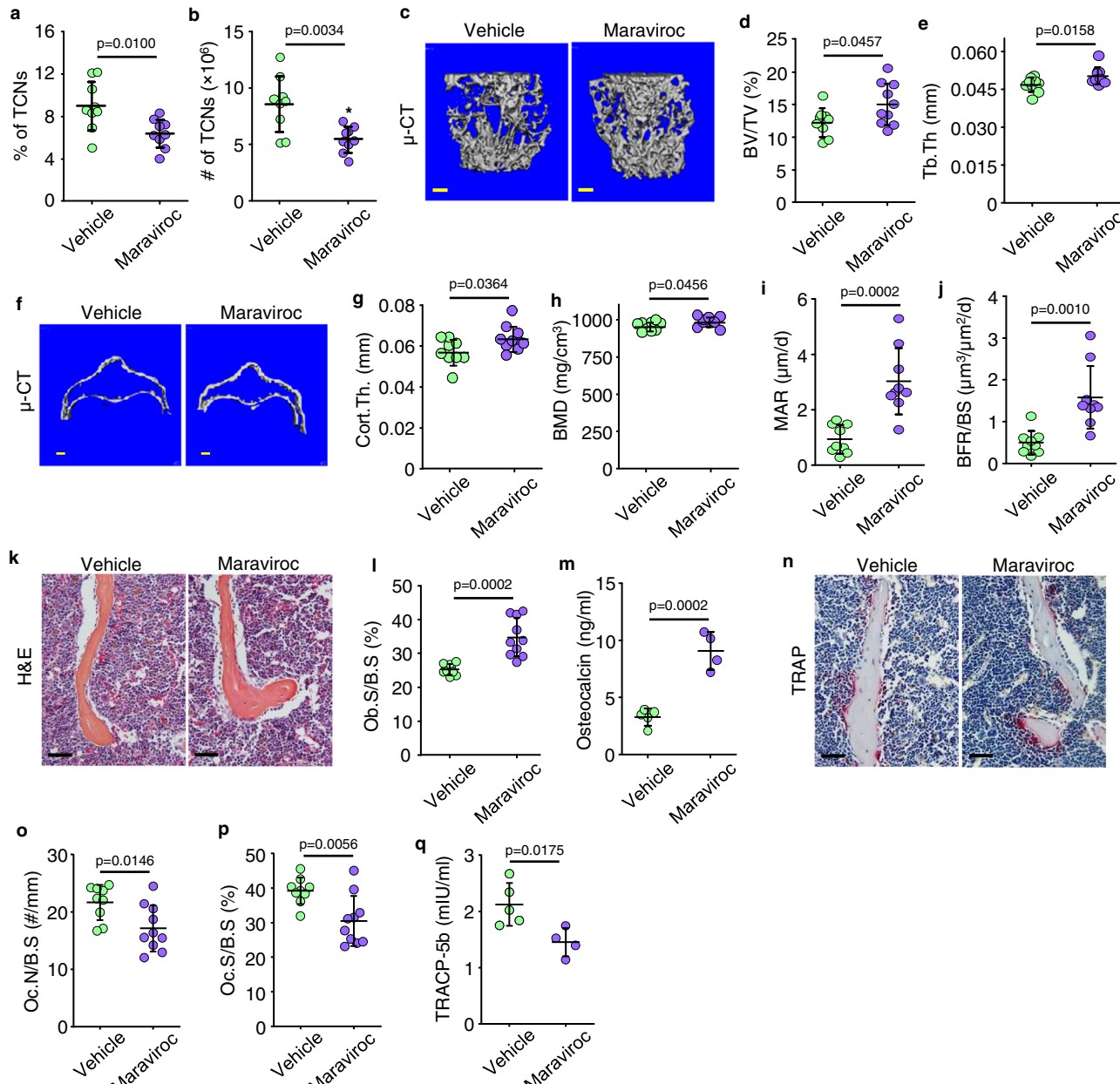

**Fig. 7 | Aged mice treated with maraviroc have fewer TCNs in BM and increased bone mass. a**, **b** Frequencies (**a**) and numbers (**b**) of TCNs in BM of 22-mon-old male C57 mice treated with vehicle or maraviroc (10 mg/kg) once/d s.c. for 1 month. Mean±SD ($n = 9$ and 10 biologically independent male mice for vehicle and maraviroc-treated groups, respectively). **c**–**e** µCT 3D reconstruction of L1 vertebrae (**c**) and values for trabecular bone volume (BV/TV) (**d**) and thickness (Tb.Th) (**e**). Bar = 1 mm. Mean ± SD ($n = 9$ and 10 biologically independent male mice, respectively). **f**–**h** Horizontal sections of µCT 3D reconstruction of cortical bones of L1 vertebrae (**f**) and values for cortical bone thickness (Cort.Th) (**g**) and bone mineral density (BMD) (**h**). Bar = 1 mm. Mean ± SD ($n = 9$ and 10 biologically independent male mice, respectively). (**i**, **j**) Bone formation parameters including mineral

apposition rate (MAR) and bone formation rate (BFR) in L1 vertebrae. Mean ± SD ($n = 9$ and 10 biologically independent male mice, respectively). **k** H&E-stained sections of L2 vertebrae and **l** OB surface values. Bar = 50 µm. Mean ± SD ($n = 9$ and 10 biologically independent male mice, respectively). **m** Serum osteocalcin by ELISA. Mean ± SD ($n = 5$ and 4 biologically independent male mice, respectively). **n** TRAP-stained paraffin sections as in (**k**), OC numbers (**o**), and surfaces (**p**). Bar = 50 µm. Mean ± SD ($n = 9$ and 10 biologically independent male mice, respectively). **q** Serum TRACP-5b by ELISA. Mean ± SD ($n = 5$ and 4 biologically independent male mice, respectively). Analyses: Student's two-sided unpaired $t$ test. Source data are provided as a Source data file.

TCNs in BM during aging degrades TRAF3 in MPCs and this leads to increased NF-κB RelA/RelB-induced transcription of *Tnfsf11* (encodes RANKL) and *Ccl5*. As a result, TCN numbers increase in BM during aging and cause osteoporosis in male mice by stimulating RANKL-induced resorption and inhibiting bone formation in a self-amplifying feedback loop (Fig. 9).

In support of this mechanism, total TGFβ1 protein levels are much higher in murine BM cells than in bone. We reported previously that

levels of active TGFβ are higher in samples of BM from 19-mon- than from 3-mon-old WT mice and in samples of vertebral bone containing BM from aged humans than from children[16], suggesting that TGFβ released from BM cells gets activated by a mechanism different from the osteoclast-derived acid that activates TGFβ released from bone matrix in resorption lacunae[34]. A limitation of our study is that we have not investigated how TGFβ from TCNs is activated in BM, but possibilities include MMP9[35], which can be activated by TNF[36] (a mediator of

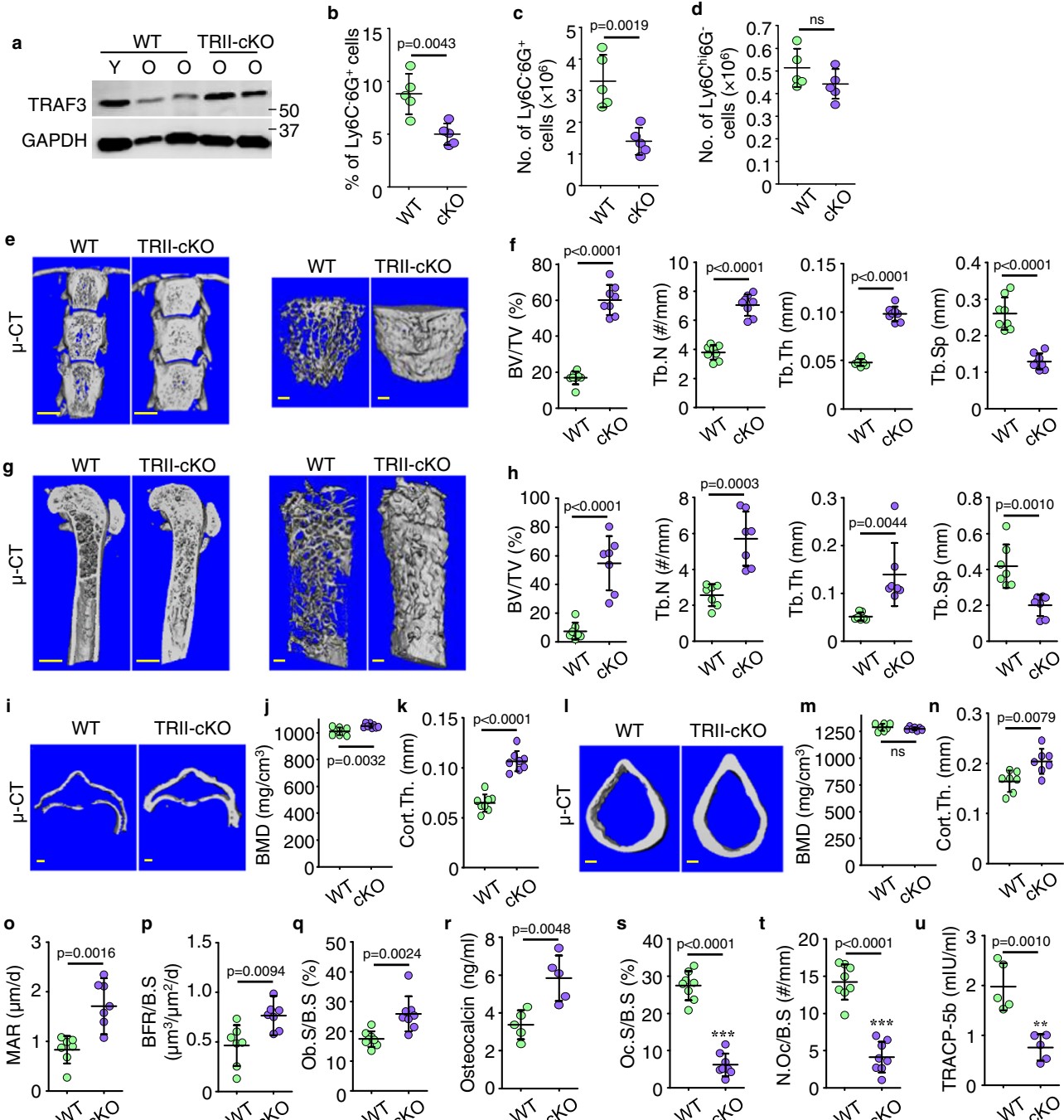

**Fig. 8 | Mice with TGFβRII deleted specifically in mesenchymal lineage cells have decreased numbers of TCNs in BM and increased bone mass. a** WB of bone protein lysates from 3- (young) and 15-mon-old (old) TGFβRII^fl/fl (WT) and 15-mon-old TGFβRII^fl/fl;Prx1^Cre (TRII-cKO) mice. **b, c** Frequencies (**b**) and numbers (**c**) of Ly6C⁻6G⁺ cells in TGFβ1⁺CD11b⁺ myeloid cells in BM from 15-mon-old WT and TRII-cKO mice. Mean ± SD (*n* = 5 biologically independent male mice/group). **d** Ly6C^hi6G⁻ cells in TGFβ1⁺CD11b⁺ myeloid cells in BM from 15-mon-old WT and TRII-cKO mice. Mean ± SD (*n* = 5 biologically independent male mice/group). **e** Representative images of coronal sections of T12-L2 vertebrae and 3D reconstruction of trabecular bone in L1 from TβRII-cKO and WT mice. Bar = 1 mm. **f** Trabecular bone volume, number, thickness, and separation in L1 from WT and TRII-cKO mice. Mean ± SD (*n* = 3 female and 5 male biologically independent mice/group). **g** Sagittal sections and 3D reconstruction of trabecular bone in femora from TRII-cKO and WT mice. Bar = 1 mm. **h** Trabecular bone volume, number, thickness, and separation within the area 1 mm beneath the growth plate in WT and TRII-cKO mice. Mean ± SD (*n* = 3 female and 4 male biologically independent mice/group). **i** Horizontal cross sections of cortical bones of L1. **j** Measurements of bone mineral density (BMD) and (**k**) cortical bone thickness. Mean ± SD (*n* = 3 female and 5 male biologically independent mice/group). **l** Cross sections of femoral cortices, and BMD values (**m**) and cortical bone thickness (**n**). Mean ± SD (*n* = 3 female and 4 male biologically independent mice/group). **o, p** Mineral apposition rate (MAR) and bone formation rate (BFR). Mean ± SD (*n* = 3 female and 4 male biologically independent mice/group). **q** Osteoblast surface (Ob.S/B.S) from H&E-stained paraffin sections of L3 vertebrae. Bar = 20 μm. Mean ± SD (*n* = 3 female and 5 male biologically independent mice/group). **r** Serum levels of osteocalcin tested using ELISA. Mean ± SD (*n* = 5 biologically independent male mice/group). **s, t** Osteoclast surface (Oc.S/B.S) and number (Oc.N/B.S) from TRAP-stained paraffin sections. Mean ± SD (*n* = 3 female and 5 male biologically independent mice/group). **u** TRACP-5b levels in serum samples in (**s**) were tested using ELISA. Mean ± SD (*n* = 5 biologically independent male mice/group). Analyses: Student's two-sided unpaired *t* test. Source data are provided as a Source data file.

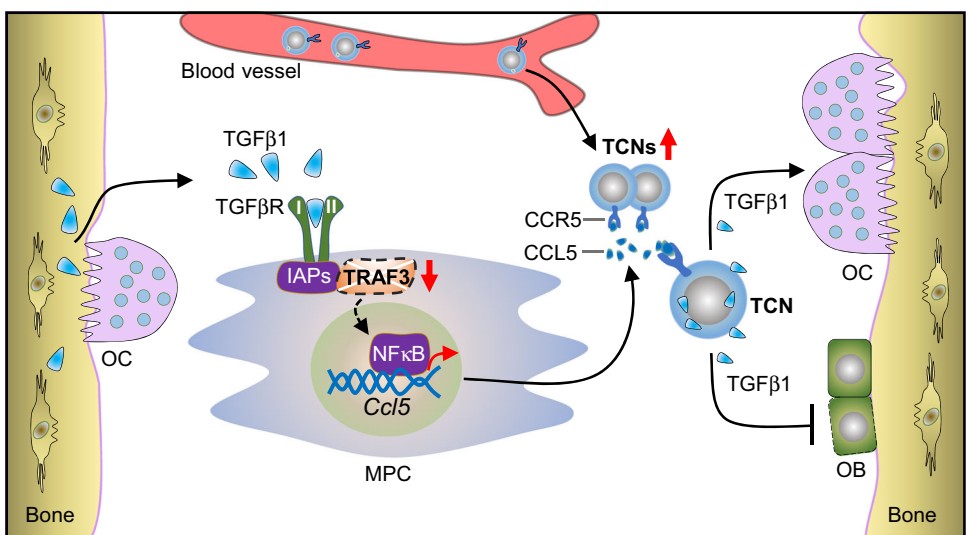

**Fig. 9 | Model for how TGFβ1-TRAF3 axis affects bone mass during aging.** During aging, increased TGFβ1 protein levels in the bone microenvironment cause decreased bone mass. TGFβ1 targets mesenchymal progenitor cells (MPCs) by binding to its receptors to trigger TRAF3 degradation, leading to excessive NF-κB-mediated CCL5 expression by MPCs. TGFβ1⁺CCR5⁺ neutrophils (TCNs), the major cellular source of TGFβ1 in BM, are recruited from peripheral blood into BM during aging, associated with increased CCL5 levels, resulting in inhibited osteoblast (OB) differentiation and enhanced osteoclast (OC) formation. As we reported previously[16], RANKL expression is increased in MPCs because of TRAF3 degradation and enhances osteoclast formation, which promotes release of more TGFβ1 into the bone microenvironment. This model supports a positive feedback loop in which TGFβ1 stimulates TRAF3 degradation and CCL5 expression by MPCs, and recruits more TCNs into BM. Interventions, such as specific deletion of TGFβRII expression by MPCs or CCR5 blockade by maraviroc, interrupt the positive feedback, and thus inhibit osteoclastogenesis and enhance osteoblast formation.

inflammaging[37]) and neutrophils themselves[38]. Although TGFβ is released from resorbing bone[39], bone remodeling is lower in aged than in adult mice and humans[23,40]. We propose that higher TGFβ expression levels in BM cells than in bone and even higher expression levels during aging suggest a key role for BM cells in the regulation of bone remodeling. We found that monocytes (Ly6C^hi6G⁻) also express TGFβ1. However, we focused here on TCNs because they are much more abundant in BM during aging (Fig. 2) and age-related osteoporosis is characterized by decreased bone formation, while bone-derived monocytes/macrophage have been reported to stimulate bone formation[41]. Given the large numbers of TGFβ-expressing neutrophils in BM of aged mice, we believe that these are a much more important source of TGFβ for degradation of TRAF3 in MPCs than TGFβ released from resorbing bone. However, definitive proof of this will require additional studies.

TGFβ1 is expressed by various cell types[42], and has been implicated in the pathogenesis of many fibrotic, inflammatory and neoplastic diseases[43] for which TGFβ inhibitors have been developed, with some success[44]. For example, the TGFβ Ab, 1D11, increased bone mass and prevented breast cancer bone metastasis-induced osteolysis in mice[45]. However, despite promising results[44,46] and an ongoing phase 1 trial of 1D11 in osteogenesis imperfecta (https://clinicaltrials.gov/ct2/show/NCT03064074), serious toxicities have hampered progress[44,46], and no TGFβ inhibitor has received FDA approval.

Although Prx1 is expressed predominantly in appendicular bones in embryos[32], it is also expressed in vertebrae[16]. Prx1 specifically tracks mesenchymal lineage cells, and mice with global knockout of Prx1 have defects in the skull, limbs, and vertebrae[16,32,47]. The 15-mon-old TGFβRII cKO mice we generated have markedly increased trabecular bone volume and cortical thickness in vertebrae and long bones, consistent with the expression pattern of Prx1. Increased bone mass has been reported in 2-mon-old mice with TGFβRII cKO driven in osteoblasts by the osteocalcin promoter[48]. Others have described defective limb formation in embryos of TGFβRII cKO mice[49] and osteoarthritis in 12-mon-old mice[50]. We did not observe these abnormalities in our cKO mice, which may reflect differences in genetic backgrounds.

We found that maraviroc, an FDA-approved small molecule CCR5 antagonist[51], given for 4 wk to 22-mon-old male WT mice or for 10 d to 3-mon-old male NSG mice injected with aged TCNs had remarkable skeletal effects: it not only reduced TCN numbers in BM, but also increased bone mass, associated with increased bone formation and decreased resorption. Lee et al. reported that a CCR5 Ab inhibited the function, but not formation, of human osteoclasts, and that Ccr5 global knockout mice have dysfunctional osteoclasts[52]. However, they also noted that the role of CCL5/CCR5 signaling in bone cells is controversial. The effects we observed with maraviroc could be different from those of an anti-human CCR5 Ab or from global deletion of CCR5, since maraviroc binds to a different site on CCR5 from the CCR5 Ab[53], and the range of its inhibitory effects is different from global CCR5 gene deletion. Lee et al. also reported that CCR5 mediates RANKL-induced bone resorption[52]. Our data suggest that CCL5/CCR5 chemotaxis signaling mediates accumulation of TCNs in BM resulting in enhanced bone resorption, inhibited bone formation, and bone loss during aging (Fig. 9) compatible with CCR5 KO mice being resistant to RANKL[52].

CCR5 is involved in recruitment of neutrophils in murine models of ischemia-reperfusion injury[54], endotoxin-induced lung injury, and pneumonia[55], and neutrophils are a source of TGFβ in tumors[27], the intestine[28], and in asthmatic subjects[29]. Here, we report CCR5⁺ neutrophils being a major cellular source of TGFβ in murine BM and important regulators of bone homeostasis, as evidenced by in vitro co-culture and in vivo ectopic implantation and adoptive transfer of this subset into NSG mice. We believe that these TCNs are distinct from other reported immune cell subpopulations involved in the pathogenesis of age-associated osteoporosis. We found that neutrophils have significantly higher CCR5 expression levels than monocytes (Figs. 5i and 5j; Suppl. Fig. 9), and that TGFβ1-expressing neutrophils (CD11b⁺TGFβ1⁺Ly6C⁻Ly6G⁺) have significantly higher CCR5 expression levels than TGFβ1-negative neutrophils (Suppl. Fig. 9).

Our findings are consistent with maraviroc acting by inhibiting TCN return to BM. Another limitation of our study is that we have not shown directly that CCL5/CCR5 signaling attracts increased numbers

of TCNs to the BM during aging. This would require generating mice with CCR5 knocked out in Ly6G[+] myeloid lineage cells, which is beyond the scope of this paper at this time. Maraviroc could also have worked by affecting the functions of other cells involved in suppression or promotion of bone remodeling. For example, although maraviroc binds to CCR5 to block interaction between HIV-1 and CCR5 and prevent HIV-1 entry into host T cells and macrophages[51], recent studies have shown that it also blocks interactions between cancer cells and immune and other cells in the tumor microenvironment[56], reprograms immunosuppressive myeloid cells, and enhances antitumor immunity by targeting an autocrine CCL5-CCR5 axis in BM[57]. In addition, maraviroc is being studied in clinical trials in breast, colorectal, and pancreatic cancer and Kaposi's sarcoma[58] following studies revealing that CCL5 and/or CCR5 are overexpressed in many cancers (including breast, gastrointestinal, prostatic, melanoma, Hodgkin lymphoma, and multiple myeloma), and they promote tumor growth, extracellular matrix remodeling, angiogenesis, immune and stromal cell recruitment, and immunosuppressive polarization of macrophages[58]. Thus, it will be important to determine exactly how maraviroc increases bone mass in aged mice. Maraviroc, overall, is well tolerated by HIV-1 patients[59], but it can have serious adverse effects (https://www.fda.gov/media/116857/download). We did not detect obvious adverse effects in the mice treated with this drug for 4 wk. However, it will also be important to determine if it can be administered long-term to mice to prevent age-related bone loss, since occasional severe side effects that may be acceptable in patients with malignancies and HIV infections[60] will not be in patients with benign diseases, such as osteoporosis. Our findings suggest that maraviroc could be repurposed in this way to treat and prevent age-related osteoporosis in humans.

Neutrophils are generally regarded as short-lived cells, generated in the BM, and released continuously into the circulation as inactive cells[61]. They migrate to sites of infection in response to chemotactic signals where they become primed to destroy invading pathogens through enhanced cytotoxic capacity and lifespan, and ultimately undergo apoptosis[62]. Unprimed neutrophils undergo senescence, and in response to chemokines, including CXCL12, return to the BM where they impair HSC mobilization[63] and undergo apoptosis[64]. *Cxcl12* gene transcription is higher in CD45[-] BMCs from adult P-cKO and old WT mice than from adult WT mice. Interestingly, recent studies have revealed that unprimed neutrophils en-route for elimination can be co-opted in tissues, such as lungs and intestines, where their lifespan can be extended, allowing them to adapt to local environments and develop non-canonical functions that tailor their properties to support organ homeostasis and repair mechanisms, including angiogenesis[65]. Growing evidence points to neutrophils having a broader range of functions than traditionally has been attributed to them, including in cancer[66], stroke[67], autoimmune, and renal disease[68], raising the likelihood that novel functions for them will be identified in additional diseases. It will be important to determine if TCNs are co-opted by cells in the BM and if their lifespan is extended to enhance the deleterious effects that we have discovered they have on bone mass during aging.

## Methods

All animal procedures were conducted in compliance with all applicable ethical regulations using procedures approved by the University of Rochester Committee for Animal Resources. For the studies with human study participants, we followed a protocol that was approved by the Research Subjects Review Board of the University of Rochester Medical Center. All patients or their guardians in the case of pediatric patients provided written informed consent. These human studies were performed with adherence to the relevant ethical regulations (Declaration of Helsinki).

### Animals

Mice with TRAF3 conditionally knocked out in osteoblast (OB) lineage cells were generated by crossing TRAF3[flox/flox] mice (C57BL/6 background) with Prx1[Cre] mice (Jackson Lab; Stock #: 005584; which we call P-cKO mice). P-cKO mice and their WT (TRAF3[flox/flox]) littermates were sacrificed at 12 months of age. Mice with TGFβRII conditionally knocked out in osteoblast lineage cells were generated by crossing TGFβRII[flox/flox] mice (Jackson Lab #012603) with Prx1[Cre] mice (Jackson Lab #005584). TGFβRII[flox/flox]Prx1[Cre] mice (which we call T-cKO mice) and their WT (Traf3[flox/flox]) littermates were sacrificed at 15 months of age. ROSA[mT/mG] mice (Jackson Lab #007676) have cell membrane-localized tdTomato (mT) fluorescence expressed widely in cells/tissues prior to Cre recombination, while, in Cre recombinase-expressing cells (and future cell lineages derived from these cells), cell membrane-localized EGFP (mG) fluorescence expression will be triggered and replace the red fluorescence. To examine the specificity of targeted gene deletion driven by Prx1Cre, we crossed Prx1Cre with Rosa26[mTmG] reporter mice, in which Prx1Cre[+] cells switch tdTomato to GFP protein expression. A non-stabilized tibia fracture model was performed on 3-mon-old Prx1[Cre];Rosa26[mTmG] mice, after anesthetizing the mice with isoflurane. Briefly, a skin incision was made along the anterior side of the mouse tibia, and a transverse osteotomy was unilaterally performed at the mid-shaft using a rotary bone saw. The incision was closed without fixation of fractured bones. Fractured bones were harvested on day 14 post-fracture for further analysis. All these mouse lines are in a C57BL/6 (C57) background. WT C57 mice were purchased at breeding age from the National Cancer Institute (Frederick, MD, USA) and bred to generate young WT mice that were sacrificed at 2–3 months of age. Aged C57 male mice (22-mon-old) were provided by the National Institute on Aging (Bethesda, MD, USA). Aged mice were injected intraperitoneally with vehicle or maraviroc (10 mg/kg/day) once/day for one month and sacrificed. No randomization was done to select animals for study. We used male mice in all experiments except those used in Fig. 8 (in which 3 pairs of female TGFβRII-cKO and WT mice were used) to avoid the effects of changing levels of female sex hormones on bone mass and metabolism during aging. Thus, our findings may not be applicable to female mice. Mice were housed with a 12/12 light cycle (typically 6 am ON and 6 pm OFF) at 72 °F (+/−2 degrees) and 30% to 70% humidity.

All mice used in this study were appropriately monitored under the University Committee on Animal Resources of the University of Rochester guidelines with the Animal Welfare Assurance number D16-00188 (A3292-01). $CO_2$ inhalation was used for mouse euthanasia. Briefly, no more than 5 mice were placed in a transparent euthanasia chamber (7½″ wide × 11½″ deep × 5″ high) and $CO_2$ was introduced at 3.0 l/min. Once the mice appeared unconscious (i.e., recumbent, without purposeful movement), the $CO_2$ influx speed was increased to 5.0 l/min. All mice were left in the $CO_2$ environment for at least 5 min until breathing and heartbeat stopped, and they had faded eye color, fully dilated pupils, and relaxation of anal/urinary sphincters.

### Reagents

The following Abs were purchased from Santa Cruz Biotechnology Inc.: TRAF3 (Cat #: sc-947), Ubiquitin (Cat #: sc-8017), RelB (Cat #: sc-226), and GAPDH (Cat #: sc-32233). Ab against human/mouse p52 (Cat #: 4882 s) was from Cell signaling. These primary Abs were used at the following concentrations: Ubiquitin (1:200) and all others (1:500). The following primary and secondary Abs were used for flow cytometry: PE-Cy7-conjugated anti-B220 (Biolegend; Cat #: 103222), FITC-conjugated anti-CD45 (eBioscience; Cat #: 11-0451-82), APC-conjugated anti-CD45 (eBioscience; Cat #: 17-0454-82), PE-Cy7-conjugated anti-Sca1 (eBioscience; Cat #: 25-5981-81), PE-conjugated anti-TGFβ1 (Biolegend; Cat #: 141403), APC-conjugated anti-CD11b (eBioscience; Cat #: 17-0112-83), PE-Cy5.5-conjugated anti-Gr1

(eBioscience; Cat #: 35-5931-80), BV605-conjugated anti-CD115 (CSF1R) (Biolegend; Cat #: 135517), PE-Cy7-conjugated anti-CD11b (eBioscience; Cat #: 25-0112-82), APC-conjugated anti-Ly6C (eBioscience; Cat #: 17-5932-82), APC-Cy7-conjugated anti-Ly6G (Biolegend; Cat #: 127624), BB515-conjugated anti-CCR5 (CD195) (BD; Cat #: 566208), PE-Cy7-conjugated anti-CD3e (eBioscience; Cat #: 25-0031-81). Primary and secondary Abs for Flow were used at 1:100 dilution. Recombinant murine TGFβ1 (#7666-MB) and recombinant mouse ICAM-1 (Cat #: 796-IC-050) were from R&D Systems. Rabbit monoclonal anti-Ki-67 (Cat #: ab15580), anti-p16 (Cat #: ab241543) primary Abs and Alex Flour488-conjugated goat anti-rabbit IgG H&L (Cat #: ab150077) secondary Ab were purchased from Abcam. Anti-PE MicroBeads (Cat #: 130-048-801) and Neutrophil Isolation Kits (Cat #: 130-097-658) were purchased from Miltenyi Biotec. ELISA kits for osteocalcin (Cat #: LS-F22474) were from LifeSpan BioScience, Inc., for TRACP5b (Cat #: MBS763504) from MyBioSource, Inc., for TGFβ1 (Cat #: DY1679-05) from R&D Systems. Maraviroc (Cat #: PZ0002) was purchased from Sigma.

## Enzyme-linked immunosorbent assay (ELISA)

Serum levels of osteocalcin and TRACP5b, as well as levels of RANKL and TGFβ1 in protein lysates of bone, BM, and BM plasma were tested by ELISA, according to the manufacturer's instructions. For sample preparation, BM cells were flushed from 1 tibia and 1 femur from each mouse with 250 μl PBS containing PIC (Roche, Cat #: 04693159001; 1 tablet in 10 ml PBS). BM cells were spun down immediately, and supernatant containing intracellular fluid was collected. Cortical bone was chopped and homogenized and immediately lysed in 250 μl T-per lysis buffer containing PIC. BM cells were also lysed in 250 μl T-per with PIC. Levels of active TGFβ1 expressed by Ly6G$^+$ BM cells were measured using a mouse TGFβ1 DuoSet ELISA kit (#DY1679) from R&D Systems, following the manufacturer's instructions. Briefly, levels of endogenously active TGF-β1 in these samples were measured by skipping the sample activation procedure, which uses an acidic solution. Cell protein lysates containing 25 μg total protein in 25 μl lysate were diluted 1:4 in assay diluent for measurement of active TGFβ1. ELISA plates were analyzed by investigators blinded to sample identity by reading absorbance using a microplate reader set to 450 nm with wavelength correction set to 540 nm. Protein from bone, BMCs, and BM plasma was extracted in a fixed volume (250 μl) of protein lysate; thus, values observed in ELISA tests were normalized by lysate volumes of the various samples.

## Flow cytometry

BM was flushed out with FACS buffer (2% FBS/PBS) from leg bones dissected from young and old C57 mice that had been treated with vehicle or maraviroc, as well as P-cKO and TRII-cKO mice and their respective WT littermates. To detect TGFβ1$^+$ granulocytes, $1 \times 10^7$ cells were stained with primary Abs, including 1 μl anti-CCR5-BB515, anti-F4/80-PE-Cy5, anti-CD11b-PE-Cy7, anti-Ly6C-APC, anti-Ly6G-APC-Cy7, anti-CD115-BV605, and 3 μl anti-TGFβ1-PE in 100 μl FACS buffer at 4 °C for 30 min. To test for intracellular expression of T-bet and TGFβ1, BM cells stained with anti-surface antigen Abs were fixed and permeabilized using BD Cytofix/Cytoperm solution kit (BD; Cat #: 51-2090/2091KZ), followed by incubation with 3 μl anti-TGFβ1-PE in 100 μl Perm/Wash buffer at 4 °C for 30 min. After two washes using BD Perm/Wash buffer and FACS buffer consecutively, these cells were ready for FACS analysis. We first gated BM cells using a "J"-shaped gate and to exclude DAPI-positive dead cells and then used forward scatter height (FSC-H) versus forward scatter area (FSC-A) plot, and side scatter height (SSC-H) vs side scatter area (SSC-A) plot to exclude doublets. To further analyze TCNs, TGFβ1-positive BM cells were gated out based on full-minus-one (FMO) control, and the CD11b-positive fraction (>90%) was next gated out from TGFβ1$^+$ BM cells, and then Ly6C and Ly6G expression on TGFβ1$^+$CD11b$^+$ BM cells weas analyzed before

measurement of CCR5 expression (Suppl. Fig. 8). Data on cell surface and intracellular marker expression were then collected by FACSDiva software using a FACS LSR II system (BD Biosciences). These gating strategies were used in all flow analyses in this study. FlowJo 7.6.5 software was used for data analysis.

## Cell sorting by FACS and magnetic columns

To sort granulocytes with TGFβ1 expression, $4 \times 10^7$ primary BM cells were incubated with 2 μl (per $10^7$ cells) anti-CD11b, -TGFβ1, -Ly6C, and -Ly6G Abs at 4 °C for 30 min. After being washed twice with 0.5% BSA-PBS and resuspended in FACS buffer at a concentration of $2 \times 10^7$ cells/ml, the targeted cell populations were sorted by FACS for various purposes, including RNA extraction, culture and cyto-spinning using BD FACSAria II sorting equipment.

Magnetic sorting was also used to isolate Ly6G$^+$ cells from mouse BM. Briefly, $8 \times 10^7$ BM cells were incubated with biotinylated anti-Ly6G Ab (2 μl per $10^7$ cells) (Millipore; Cat #: MABF1480) at 4 °C for 30 min, followed by anti-biotin MicroBeads (3 μl per $10^7$ cells) (Miltenyi; Cat #: 130-090-485) at 4 °C for 20 min. After being washed twice with 0.5% BSA-PBS, each sample was sorted using LS columns. Isolated Ly6G$^+$ cells were lysed in RIPA lysis buffer with PIC for TGFβ1 ELISA testing.

## NSG mice injected with sorted TGFβ-expressing neutrophils (TNs) or TGFβ/CCR5-expressing neutrophils (TCNs)

3- and 20–22-mon-old C57 and ROSA$^{mTmG}$ male mice were sacrificed for isolation of TNs and TCNs from BM. TGFβ1-expressing BM cells were first enriched through positive selection using a magnetic isolation system with PE-conjugated anti-TGFβ1 Ab as primary and anti-PE MicroBeads as secondary labeling. Ly6G-positive cells or CCR5/Ly6G double-positive cells were further sorted by BD FACSAria II sorting equipment from the positively-selected TGFβ1-expressing BM cells. The purity of TNs and TCNs in post-sort fractions was over 90%, which was confirmed using a BD FACS LSRII cytometer, and the mean fluorescence intensity (MFI) of TGFβ1 expression by these enriched TNs and TCNs was over 4-fold higher than Ly6G$^+$ neutrophils. Male NSG mice, aged 3-mon-old were injected with one million TNs or TCNs from young or aged mice via tail vein every other day for a total of 3 injections. NSG recipients injected with aged TCNs were treated with maraviroc (2 mg/kg body weight) by I.P. injection once per day from the day of the first TCN injection for 10 days and the mice were sacrificed the following day for bone phenotype analysis. We could have injected TNs or TCNs into same background WT recipients after lethal or lower dose radiation, rather than into NSG mice. However, increasing evidence suggests that neutrophils are antigen-presenting cells, which can activate adaptive immune cells, especially T cells[69], and activation of adaptive immune cells[70] or radiation[71] are associated with bone loss, which would have increased the complexity of the approach.

## Co-culture of immune cells with mesenchymal or osteoclast progenitor cells

$1.4 \times 10^5$ 2$^{nd}$ passage of BdMPCs from 3-mon-old male C57 mice were seeded in each well of 48-well plates and maintained in 10% FBS culture medium overnight. On the second day, $2 \times 10^7$ BM cells harvested from 3- and 21-mon-old male C57 mice were incubated with 4 μl biotinylated anti-Ly6G Ab (Millipore; Cat #: MABF1480) at 4 °C for 30 min, followed by 6 μl anti-biotin MicroBeads (Miltenyi; Cat #:130-090-485) at 4 °C for 20 min before sorting using LS columns. Positively-selected cells were confirmed to be Ly6G$^+$ cells (>96%) using FACS. Appropriate amounts of Ly6G$^+$ cells were gently resuspended in OB-inducing culture medium to make different concentrations (1.4, 2.8, 5.6, 11.2, 22.4 $\times 10^5$ cells/ml) of cells, and 500 μl of these various concentrated cell solutions were added gently in each well of BdMPC-coated plates and maintained for 4 d. Cell culture was terminated by fixation with 10% formalin for 5 min. Alkaline phosphatase (ALP) staining was performed for measurement of the area of ALP$^+$ cells.

For TGFβ-expressing neutrophil (TN) isolation, 3- and 20–22-month-old C57 male mice were sacrificed, and BM cells were harvested. Briefly, BM cells were first labeled with PE-conjugated anti-TGFβ1 Ab and anti-PE MicroBeads in order, according to the manufacturer's instructions, and then followed by magnetically-positive selection to isolate TGFβ1-expressing BM cells. These selected cells were next labeled with APC-Cy7-conjugated anti-Ly6G Ab and sorted by flow for co-culture experiments.

For co-culture with MPCs, $1 \times 10^6$ bone marrow cells from 3-month-old C57 mice were seeded in each well of 24-well culture plates in MPC culture medium (α-MEM with 15% FBS, 1% Pen/strep, 1% L-glutamine) for 7 d without medium change. On day 8, the culture medium was discarded, and $4 \times 10^5$ TNs were resuspended in 1 ml osteoblast-inducing medium and added into each MPC-precoated well. The fresh medium containing freshly-sorted TNs was added on day 11 and this culture was terminated on day 15. ALP staining was performed for osteoblast (ALP+) area assay.

For co-culture with osteoclast precursors (OCPs), $2.5 \times 10^4$ BM cells from 1-month-old C57 mice were seeded in each well of 96-well plates with OCP culture medium (α-MEM with 10% FBS, 1% Pen/strep, 1% NEAA and 40 ng/ml M-CSF) for 2 d. On day 3, the culture medium was discarded, and $2.5 \times 10^4$ TNs were resuspended in 0.2 ml osteoclast differentiation medium (containing 40 ng/ml M-CSF and 10 ng/ml RANKL) and added on top of each OCP-precoated well. Fresh medium containing-freshly sorted TNs was added every other day. On day 15, the co-culture was terminated, and TRAP staining was performed for TRAP+ osteoclast number assay.

### In vivo ectopic bone formation assay

3- and 24-month-old male ROSA^mTmG male mice were sacrificed, and BM cells were flushed out using sterile 0.5% BSA/PBS. After lysis of red blood cells, cell numbers were counted, and 3 μl of biotin-conjugated anti-Ly6G Ab (Millipore; Cat #: MABF1480) was added to $1 \times 10^7$ cells and incubated at 4 °C for 20 min. Cells were washed twice with sterile 0.5% BSA/PBS and 5 μl streptavidin-conjugated MicroBeads (Miltenyi; Cat #: 130-090-485) per $1 \times 10^7$ cells were added and incubated at 4 °C for 20 min. After two washes with sterile 0.5% BSA/PBS, cells were magnetically sorted using LS columns, and post-sort analysis was performed to confirm the purity of these sorted BM Ly6G+ cells. After this enrichment process, the Ly6G+ cells were further incubated with anti-TGFβ1, -CD11b, and -Ly6C Abs following the procedures described in the cell sorting by FACS section to sort TNs. Meanwhile, 3rd passage compact bone-derived MPCs (BdMPCs) from WT (TGFβRII^f/f) and TRII-cKO (TGFβRII^f/f;Prx1^Cre) mice were digested from cell culture dishes, and cell numbers were counted. $0.5 \times 10^6$ BdMPCs were mixed with $4 \times 10^6$ sorted TNs in 50 μl 10% FBS/PBS, and these cells were absorbed onto pieces of chopped Gelfoam (Pfizer; Cat #: 00300090315085; $5 \times 5 \times 7$ mm in size) and implanted into the dorsal subcutaneous tissue of 3-mon-old NOD scid gamma (NSG) mice. At 4 wk post-implantation, the implants were dissected from the recipient mice, fixed using 10% formalin at 4 °C for 24 h, processed through paraffin, and 4-μm-thick sections were cut using a microtome for H&E and TRAP staining.

### Micro-CT and bone histomorphometric analysis

For in vivo assessment of bone formation, mice were given injections of calcein (10 mg/kg) 5 and 1 d before sacrifice according to our standard protocol[16]. Right tibiae and T12–L1 or L2–L3 vertebrae were fixed in 10% neutral-buffered formalin for 2 d. Micro-CT scanning with a resolution of 10.5 μm was performed using a vivaCT 40 instrument (Scanco Medical), and trabecular bone parameters were assessed in a region 1–4 mm from the edge of the growth plate in the metaphyses of femora and in the entire trabecular bone area of the first lumbar vertebrae. The bone specimens were then processed sequentially through two changes of 95% ethanol for 1 h each, two changes of 100% ethanol for 1 h each under vacuum conditions, two changes of LR white

hydrophilic medium (Polyscience; Cat #: 17411-500) for 1 h each without vacuum suction, two changes of LR white hydrophilic medium for 12 h each under vacuum conditions, and then changed into fresh LR white hydrophilic medium and heated at 60 °C overnight to cure the plastic. 10 d later, 4-μm-thick sections were cut from the plastic blocks using a Shandon Finesse ME microtome. Calcein double-labeling was assessed in unstained slides under fluorescence microscopy, and dynamic parameters of bone formation were measured to allow calculation of mineral apposition rate (MAR) and bone formation rate (BFR)[16,72]. For von Kossa staining, 4-μm-thick vertebral coronal plastic sections were stained using 1% aqueous silver nitrate solution placed under ultraviolet light for 20 min. Left tibiae and L2- or L3–L5 vertebrae were fixed in 10% neutral buffered formalin for 2 d, maintained in 70% ethanol for 2 d, decalcified in 10% EDTA for 14 d, embedded in paraffin, and 4-μm-thick sections were cut. Static parameters of osteoblasts and osteoclasts were assessed blindly in H&E- and TRAP-stained sections of femoral primary spongiosae and vertebral trabecular bone by an investigator who was not involved in the sample collection and group assignment using an OsteoMeasure Image Analysis System (Osteometrics, Decatur, GA) and assessment of immunofluorescence-stained sections presented in Suppl. Fig. 11 was done using ImageJ 1.8.0 software (National Institutes of Health, USA).

### Cell migration velocity assay

To assess neutrophil migration in response to chemokines, Millicell EZ SLIDE 8-well glass chambers (Millipore, Cat #: PEZGS0816) were first incubated with 200 μl of 10 μg/ml recombinant mouse ICAM-1 plus 200 ng/ml of CCL4, CCL5, or CXCL12 overnight at 4 °C. On the 2nd day, both neutrophils and TGFβ1-expressing neutrophils (TNs) were freshly isolated from BM of 2- and 22-month-old C57 mice using a Neutrophil Isolation Kit, and $1 \times 10^4$ isolated cells were resuspended in 200 μl of L-15 medium (Gibco, Cat #: 11415114) and added into each glass chamber. Video recording of cell migration in a velocity assay was performed after incubation at 37 °C for 10 min. The migration video was recorded using a ×20 objective lens in a Nikon TE2000-U microscope, and image processing and velocity assay were performed using Volocity 6.3 Image Analysis Software by PerkinElmer, Inc. (PerkinElmer, Waltham, MA).

### Transwell assay for cell migration

Migration was determined by using a transwell migration assay (Corning Life Sciences, Acton, MA) with a pore size of 0.5 μm on the membrane (Cat #: 3421), according to the manufacturer's instructions. 3rd passage WT and P-cKO MPCs were seeded in 24-well plates at a concentration of $5 \times 10^4$ per well and maintained for 2 d without medium change. Meanwhile, TGFβ1+CD11b+Ly6G^hiLy6C+ cells (TNs) sorted from BM of 3- (young) and 20–22-mon-old (aged) mice were treated in 1% heat-inactivated fetal bovine serum (FBS) with 50 μM maraviroc for 2 h. $3 \times 10^4$ of these pre-treated cells were resuspended in 100 μl RPMI 1640 medium containing 1% FBS plus 50 μM maraviroc and added to each transwell insert. The inserts then were placed in the lower chambers, which were coated with MPCs. After incubation for 6 h, cells were fixed in 4% PFA and the inside of each inset was gently rubbed with cotton swabs to remove cells that failed to migrate into the membrane, according to the manufacturer's instructions. Finally, crystal violet staining was performed for cell counting using a light microscope.

### Cryosection staining

Leg bones of NSG mice that were injected with TNs from 20 to 22-month-old ROSA^mTmG mice and treated with maraviroc were decalcified and placed in 30% sucrose at 4 °C for 2–3 d. The bones were embedded in OCT and 10-μm-thick sections were cut. Briefly, cryosections were washed 3 times with PBS for 5 min and once with PBST containing 0.2% Triton X-100. The sections were next blocked using 10% normal goat

serum in PBST at room temperature for 30 min, followed by incubation in 3% Affinipure Fab fragment anti-mouse IgG (H + L)/anti-mouse IgM (Cat #: 115-007-003/115-006-020; Jackson ImmunoResearch Laboratories) at room temperature for 1 h. After 3 washes with PBS (10 min each), sections were incubated with primary Abs, including rabbit monoclonal Ab to Ki-67 (dilution: 1/1000) and rabbit monoclonal Ab to p16[Ink4a] (dilution: 1/100) at 4 °C overnight. On the 2nd day, sections were washed 3 times with PBS for 10 min and incubated with Alexa Fluor® 488-conjugated goat anti-rabbit IgG H&L (dilution: 1/400) at room temperature for 1 h. After 3 washes with PBS, sections were mounted with Vectashield medium with DAPI (Cat #: H-1200; Vectashield). Sections were imaged using an Olympus FV1200MPE confocal microscope.

## Western blotting

Mouse long bones were homogenized and lysed with T-Per lysis buffer containing a Protease Inhibitor Cocktail (PIC) tablet (Roche; #11697498001). BM cells and cultured cells treated with various reagents were lysed in PIC-containing RIPA lysis buffer (Millipore; Cat #: 20-188). After incubation at 4 °C for 30 min, protein lysates were centrifuged at $16.2 \times 10^3$ G for 15 min and the supernatant was collected. These protein lysates were boiled with 4 × loading buffer and adjusted to a final protein concentration of 1 µg/µl. 20 µg of protein lysates were loaded in 10% SDS-PAGE gels and transferred onto polyvinylidene difluoride (PVDF) membranes. Membranes were incubated with the primary Ab overnight at 4 °C and then with horseradish peroxidase-linked secondary Ab (Bio-Rad) for 2 h at room temperature. The membranes were incubated with Enhanced Chemiluminescence (ECL) substrate, and targeted proteins were detected using an imaging system from Bio-Rad. Densitometry analysis was performed using the Bio-Rad Image Lab 5.1 software.

## Quantitative real-time PCR

1 µg RNA extracted from sorted/cultured murine cells and human vertebral specimens was reverse-transcribed to cDNA using an iSCRIPT cDNA Synthesis kit (Bio-Rad). The expression level of the entire set of chemokine genes was measured using an iCycler RT qPCR machine (Bio-Rad), which detected the level of fluorescence signals emitted as iQ SYBR SuperMix (Bio-Rad) binding to dsDNA during DNA extension. Primer sequences of mouse genes are as follows: *Ccl5*, forward, 5′-GACACCACACCCTGCTGCT-3′, and reverse, 5′-TACTCCTTGATGTGGG CACG-3′; *Tnfsf11 (Rankl)*, forward, 5′-CAGAAGGAACTGCAACACAT-3′, and reverse, 5′-CAGAGTGACTTTATGGGAACC-3′; *Opg*, forward, 5′-TACCTGGAGATCGAATTCTGCTT-3′, and reverse, 5′-CCATCTGGA-CATTTTTTGCAAA-3′; *Gapdh*, forward, 5′-GGTCGGTGTGAACGGAT TTG-3′, and reverse, 5′-ATGAGCCCTTCCACAATG-3′. Primer sequences of human genes are as follows: *Ccl5*, forward, 5′-CCATGAAGGT CTCCGCGGCAC-3′, and reverse, 5′-CCTAGCTCATCTCCAAAGAG-3′; *Gapdh*, forward, 5′-AAGGTGAAGGTCGGAGTCAAC-3′, and reverse, 5′-GGGGTCATTGATGGCAACAATA-3′. The relative abundance (ΔCT) of each gene was calculated by subtracting the GAPDH CT value from the corresponding CT value of specific genes, and ΔΔCT values were obtained by subtracting the ΔCT values of control samples from the others, and then raised to the power 2 ($2^{-\Delta\Delta CT}$) to yield fold changes relative to the controls. The relative expression levels of chemokine genes in CD45- BM cells from 9-mon-old P-cKO and 24-mon-old WT mice were compared with those from 9-mon-old WT mice. A heatmap of chemokine gene expression was generated using a heatmap function natively provided in R, a language and environment for statistical computing developed by R Core Team.

## Cytokine array

BM was flushed from left femora using 1 ml PBS with 1% protease inhibitors and homogenized in an electronic homogenizer. 10 µl of Triton X-100 was added to the sample, which was then frozen overnight at −70 °C. The sample was thawed at room temperature, centrifuged at $10,000 \times g$ for 5 min to remove cellular debris. The volume of 100 µg protein tissue lysate from each sample was adjusted to 1 ml using array buffer 6 (R&D system; Cat #: ARY006) and then transferred to a separate tube containing 0.5 ml of array buffer 4. 15 µl of Mouse Cytokine Array Panel A Detection Antibody Cocktail (R&D system; Cat #: ARY006) were added to each prepared sample, which was mixed and incubated at room temperature for 1 h. These sample/Ab mixtures were transferred to wells of 4-well multi-dishes containing array membranes that had been blocked by a buffer 6 and incubated overnight at 4 °C. After thorough washing, the array membranes were incubated with Streptavidin-HRP for 30 min at room temperature followed by incubation with Chemi Reagent Mix for 1 min, and the targeted factors were then detected using an imaging system from Bio-Rad.

## Human specimen collection

We collected samples of vertebral bone that were removed from pediatric and adult patients undergoing elective surgery to correct spinal scoliosis and degenerative conditions, including cervical spondylosis, lumbar spinal stenosis, and disc herniation. We used a Rongeur to remove portions of the spinous processes in posterior cervical, thoracic, and lumbar spine procedures. In anterior cervical spine procedures, a Kerrison Rongeur was used to remove bony portions of the anterior overhang of the cervical vertebrae. These bone samples would typically have been discarded as part of the surgical procedure. The study enrolled 55 subjects, including 28 females and 27 males, ranging from 8- to 87-years-old, of which 26 subjects were 8–18-year-old children (10 males, 16 females) and the remaining 29 were middle-aged to elderly from 53 to 87 years (18 males, 11 females). Subjects with tumors, active systemic, immunologic, inflammatory, or metabolic disorders that might affect bone remodeling were excluded. Since the levels of chemokine mRNA in human bone samples had not been assessed when we started the study, we were unable to use a power analysis to determine the number of samples that would be required. We estimated that we would require a minimum of 15 samples from children and from adults to detect statistically significant differences in levels between them.

## ChIP assay

Transcription factor binding sites within 1 kb before the start codon of the murine *Ccl5* gene were searched using TFSEARCH software, and 3 κB binding sites were predicted in these regions on the *Ccl5* gene promotor. The binding of RelA and RelB to each of the κB binding sites was tested by ChIP assays following the procedure we published[16]. Briefly, the sheared chromatin from TRAF3[fl/fl] (WT) and Prx1[Cre];TRAF3[fl/fl] (P-cKO) BdMPCs that had been fixed with 1% formaldehyde was immunoprecipitated with anti-RelA or -RelB Abs, or rabbit IgG as a negative control. The precipitated DNA was used as a template for PCR using primers specifically designed to amplify a segment of 150−250 bp covering the putative κB binding sites. The sequences of the primers are: *Ccl5* site1/2, forward 5′-GGAGGGCAGTTAGAGGCAGA-3′ and reverse 5′-TGCAGCTCAGGCAGACCT-3; and *Ccl5* site 3, forward 5′-CTGGGAATCAGGATTACCTGGC-3′ and reverse 5′-CAGGTAGCAGG GAGCTGTTG-3. In addition, two pairs of unrelated primers were designed as control in the DNA region that is away from the κB binding sites on the *Ccl5* gene promotor. The primer sequences were as follows: pair 1, forward 5′-TGGTCGAAGCATGGCCAAC-3′ and reverse 5′-CATTGGTGTTCAGCCCTGC-3; and pair 2, forward 5′-GTGTCCAGCT GAGATGCACT-3′ and reverse 5′-GTGCCCACCAAGGTACAGAG-3. The procedure for Quantitative Real-time PCR described above was next performed to analyze the relative levels of RelA and RelB binding to these κB binding sites.

## Statistics and reproducibility

All results are given as the mean ± S.D. Variance was similar between groups for most parameters assessed. Comparisons between two groups were analyzed using Student's two-tailed unpaired $t$ test and those among 3 or more groups using one-way analysis of variance (ANOVA) followed by Tukey's post hoc multiple comparisons. $p$ values <0.05 were considered statistically significant. Statistical data in this study were analyzed and dot plots were generated using GraphPad Prism 6. The sample size for in vivo experiments is based on an unpaired t-test power analysis carried out by our statistician using SigmaStat Statistical Software: 5–8 mice were needed in each group where bone parameters were being assessed to detect significant differences from controls with an alpha error of 5%. The power is 0.98, i.e., there is 98% chance of detecting a specific effect with 95% confidence when alpha = 0.05. No data were excluded from the analyses. All in vitro experiments were repeated twice independently with similar results. All in vivo experiments were carried out once.

## Reporting summary

Further information on research design is available in the Nature Portfolio Reporting Summary linked to this article.

## Data availability

The raw data for all dot plots are in an Excel file, which is in the Data Source file for this paper along with uncropped blots for all protein bands, and gene expression levels for heatmap generation. Flow cytometry files have been deposited in FlowRepository with a Repository ID: FR-FCM-Z5VU, which is available following the URL: http://flowrepository.org/experiments/6142. All other data generated or analyzed during this study are included in this published article (and its supplementary information files). Source data are provided with this paper.

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

## Acknowledgements

Research reported in this publication was supported by the National Institute of Arthritis and Musculoskeletal and Skin Diseases of the National Institutes of Health under Award Numbers R01AR43510 (to B.F.B.), P30 AR069655 (to Edward Schwarz), AR063650 (to L.X.) and by the National Institute for Aging under Award Number R01AG049994 (to B.F.B. and Z.Y.) and R01AG076731 (to Z.Y.) and by the National Institutes of Health under Award Number 1S10RR027340 (to B.F.B.) and by the National Natural Science Foundation of China under Award Number 82172469 (to J.L.). The content is solely the responsibility of the authors and does not necessarily represent the official views of the National Institutes of Health. The TRAF3[fl/fl] mice were a gift from Dr. Gail Bishop, University of Iowa.

## Author contributions

J.L. supervised, conceived, and coordinated the studies, performed and analyzed the experiments shown in all Figures except Figs. 5b, 5c, 5h, 8a and Suppl. Fig. 6, and wrote the paper. Z.Y. conceived and coordinated the studies, performed and analyzed the experiments shown in Figs. in 5b, 5c and Suppl. Fig. 6. X.L. performed and analyzed the experiments

shown in Figs. in 5h and 8a. and analyzed µCT data. R.D. generated Prx1$^{Cre}$;TGFβRII$^{fl/fl}$ mice and analyzed histologic data. X.Y. and A.A. performed experiments with maraviroc. J.O.S. and A.M. provided human bone samples. L.X. conceived and coordinated the studies. B.F.B. supervised, conceived, and coordinated the studies, and wrote the paper. All authors reviewed the results and approved the final version of the manuscript.

## Competing interests

The authors declare no competing interests.
