## [Peer Review File · Nature Communications]

TGF β 1+CCR5+ neutrophil subset increases in bone marrow and causes age-related osteoporosis in male miceREVIEWER COMMENTS

Reviewer #1 (Remarks to the Author):

NCOMMS-21-19687

Novel TGFβ⁺/CCR5⁺ neutrophil population causes age-related osteoporosis in mice

In a line of studies by this research group, they have demonstrated that TGFβ¹ is a key molecule for inducing age-related bone loss through which it promotes degradation of TRAF3 in bone cells. This study identified a novel subset of TGFβ¹+CCR5⁺ neutrophils (TCNs) as the major source of TGFβ¹ in murine bone, of which increase in bone likely contributes age-related bone loss. This finding should be highly admired. However, there are some concerns in logic flow based on unsound evidence and somehow subjective interpretations. Specific comments are listed below.

Major comments;

In co-culture experiments shown in Fig.2, effects of LysG⁺ BM cells on osteoclastogenesis and bone resorption were not assessed. RANKL/OPG ratio in the coculture, and N.Oc (or at least one osteoclastic parameter) in the implant should be scored. Furthermore, the effects of TGF-beta inhibitor or its neutralization Abs on osteoblastogenesis and osteoclastogenesis are recommended to be conducted. These experiments will provide more sound evidence for reaching the message as represented by the title.

Functional evidence for the involvement of CCL5-CCR5 axis in the entire data set appears to be unsound. It is probable that functional dissociation of a given chemokine and its receptor from others encounter experimental difficulties because of their multiple cellular targets and functional redundancies. Nevertheless, more experimental evidence and unbiased discussion would strengthen this research work.

In data sets shown in Fig.4, validation of mRNA and proteins of Ccl-12, Cxcl12, Ccl24 as well as Cxcl4, Ccl26 and Ccl2 should be demonstrated, though some of them are not significantly changed.

In in vitro cell movement assay shown in Supplementary Fig.2c, the effect of Maraviroc treatment will be informative to see how CCR5 specifically contributed to this functional assay.

To further assess functional contribution of CCR5-mediated signal for TCNs to highly accumulate in bone marrow, assays to see the impact of CCR5 signal on cell survival should be designed.

As for the in vivo and vitro experiments of Maraviroc treatments shown in Fig.5 and Supplementary Fig.4, Lee et al. previously demonstrated the effects of Maraviroc on human osteoclast differentiation, and the impact of Ccr5-deficiency in osteoclastogenesis and bone with multiple careful assays (Nat Commun. 2017 Dec 20;8(1):2226. doi: 10.1038/s41467-017-02368-5), therefore, it sounds quite naïve to describe the sentence of “ In contrast, maraviroc has no significant direct effects on osteoblast or osteoclast formation in vitro (Supplementary Fig. 4a & 4b), suggesting that its effects on osteoclast and osteoblast numbers in vivo are mediated by other cell types.” Differentiation and functional regulation of osteoblasts and osteoclasts are multiple steps, merely counting ALP⁺ cell area or TRAP⁺ cell area do

not exclude the effects of CCR5-blockade on these bone cells, therefore, more detailed histomorphometry would be required. Genetic deletion of Ccr5 and Maraviroc treatment may exhibit somehow distinct bone phenotypes, Maraviroc may be less effective in mouse cells since designed targeting human CCR5. These points would also be taken for further consideration in the interpretation and discussion of these experimental data. Previously published works of CCL5-CCR5 (and other CC-chemokines and receptors, if necessary) in the function of bone cells should be fairly referenced and discussed.

Bone phenotype of TRII-cKO shown in Fig6. is quite striking. It could be wondered why vertebral bones of T12-L2 showed strong phenotype even though Prx1-driver is specific in limb mesenchyme, possibly needs explanation for this.

Related to this, some in vitro functional experiments to assess osteogenic potential of TRII-deficient mesenchymal cells such as co-culture with TCNs as shown in Fig.1 will strengthen the message.

Minor points;

Line 112 and M&Ms, “mTmG” and/or ROSA mTmG needs more specific explanation for readers.

Line 165, TGFβ1+CCR5+ neutrophils TCNs for short may be rephrased as TGFβ1+CCR5+ neutrophils (TCNs for short).

Lines 182, and Fig.6 legend title, “increased bone loss” may be rephrased as “increased bone mass” or else.

Reviewer #2 (Remarks to the Author):

Low-level chronic inflammation caused by aging, inflammaging, is considered one of the causes of age-related bone loss, osteoporosis. In the present study, the authors demonstrated that neutrophils are the major source of TGFβ in bone marrow in aged mice, and TGFβ-expressing neutrophils from aged mice inhibited osteoblast differentiation from mesenchymal progenitor cells (MPCs). Conditional deletion of TGFβ receptor II (TRII) in cells of mesenchymal lineage abrogated the inhibitory effect of TGFβ-expressing neutrophils on osteoblast differentiation. TGFβ is implicated in the degradation of TRAF3, and TRAF3 deletion in mesenchymal lineage cells reduced osteoblast differentiation and caused osteoporosis. The authors found that CCL5 levels are high in bone marrow of TRAF3 cKO mice, and the expression of CCL5 mRNA was high in MPCs from aged mice. The expression of CCR5, the CCL5 receptor, was high in TGFβ1+ neutrophils, and the increase in the number of TGFβ1+CCR5+ neutrophils (TCNs) were observed in bone marrow of aged mice. CCR5 antagonist maraviroc increased bone formation and reduced osteoclast number in aged mice, while maraviroc has no direct effects on osteoclasts or osteoblasts in vitro. TRII-cKO mice exhibited decreased number of TCNs in bone marrow and increased bone volume as compared with wild type littermates. Based on these findings, the authors concluded that TCNs attracted to bone marrow through the CCL5-CCR5 axis suppressed bone formation and

promoted bone resorption by TGF β -TRII pathways in aged mice. The experiments are carefully performed, and the conclusion appears sound. However, there are several points to be addressed.

Comments

- 1) As the authors commented, Seo et al. previously reported that Prx1-Cre-mediated conditional deletion of TGF β RII resulted in defective limb development (reference #45). This appears reasonable since Prx1 enhancer is mainly activated in the early limb bud mesenchyme (and in a subset of craniofacial mesenchyme) (Logan et al., *Genesis*. 2002 Jun;33(2):77-80; also see The Jackson Laboratory website <https://www.jax.org/strain/005584>). Therefore, the lack of abnormalities in the limbs, in contrast to the report by Seo et al, raises serious concerns regarding the specificity of the Prx1-Cre mice that the authors used. The authors should confirm that Cre is specifically expressed in MPCs by crossing the mice with Rosa26 reporter mice, for example. It is also critical if TGF β RII and TRAF3 is in fact specifically deleted in MPCs of “vertebrae” (not long bones) in their cKO mice since Prx enhancer is usually not activated at axial bones.
- 2) To further confirm the central role of TCNs in the bone loss in aged mice, the authors should perform transplantation of TNCs from old mice into young mice.
- 3) What is the mechanism underlying the increased bone resorption in P-cKO mice?

Reviewer #4 (Remarks to the Author):

Summary

With their manuscript entitled “Novel TGF β + /CCR5+ neutrophil population causes age-related osteoporosis in mice”, Li et al aims to provide a cellular and molecular mechanism, involving neutrophils, TGF β 1/TGF β RII, TRAF3, and mesenchymal progenitor cells, that can, at least in part, explain age-related osteoporosis. The authors show that TGF β 1-expressing neutrophils increase with age in mice and that bone marrow neutrophils from older mice, versus from younger mice, suppress bone formation to a higher degree in vitro and in vivo using an ectopic bone formation assay. The authors also provide evidence that the neutrophil-induced suppression of bone formation was mediated through TGF β RII. Moreover, the authors show that mice with conditional deletion of TRAF3 results in decreased bone mass and increased TGF β 1-expression neutrophils in the bone marrow. The authors further aim to link an age-related increased bone marrow CCL5 gene and protein expression to the accumulation of TGF β 1-expressing neutrophil in the bone marrow. Additionally, the authors show that therapeutically targeting CCR5, which is expressed by neutrophils (and likely other cells) and is one of the receptors for CCL5, modestly increases bone mass in aged mice, while simultaneously slightly decreasing TGF β 1-expressing neutrophils in the bone marrow. Finally, the authors show that conditionally deleting TGF β II in Prx1-expressing cells (which reportedly is specific to mesenchymal progenitor cells) results in substantially increased bone mass and decreased TGF β 1-expressing neutrophils. The manuscript contains many interesting and thought-provoking results that could help clarify the underlying mechanisms of age-related osteoporosis (which is complex and the likely result of diverse biological processes that may differ between individuals). The authors use several different mouse models and a variety of tools and

experimental readouts to investigate their research question. The manuscript is well written and the results communicated clearly, for which the authors should be commended. However, there are a few key issues that need to be addressed prior to publication. Most of these comments can be addressed through experiments that have already been performed, adjusting the text, or by relating their work to publicly available datasets.

Major comments

Apart from figure 2, the authors show mainly correlative connections between TGFb1-expressing neutrophil increases or decreases and bone volume; i.e. the authors lack sufficient data to back up the title of the paper, which states that TGFb1+ neutrophils cause age-related osteoporosis in mice. In figure 2, it is also unclear from the figure whether the authors used TGFb1-expressing neutrophils in these experiments or not. The authors should consider adjusting the title of the paper to more accurately reflect the data.

What is the proportion of neutrophils that express TGFb1? i.e. what does it look like if instead of pre-gating on TGFb1, the authors first gate on neutrophils, then look at TGFb1 expression. Is it a gradient or two split populations (high vs low)? Is TGFb1 expression a feature of the majority of bone marrow neutrophils or a smaller fraction? This would be relevant to understand considering that all BM Ly6g+ neutrophils may have been used for Figure 2 for example and to clarify whether it is a large versus small fraction of neutrophils that express TGFb1. If most neutrophils express TGFb1, then calling these cells a neutrophil subset would be less appropriate. Rather, one could consider the scenario that all neutrophils in the bone marrow upregulate TGFb1 expression with age. Also, does TGFb1 expression relate to neutrophil maturity? See the next comment, which is related to this point.

The authors posit that they have defined a novel neutrophil subset. In general, the authors may want to be more cautious in making this claim. How does the 'new' subset of TGFb1+CCR5+ neutrophils compare to previously reported neutrophil subsets? For example, has any other studies shown that bone marrow neutrophils express TGFb1? and/or express CCR5? What about compared to bone marrow neutrophil states (Evrard et al, Immunity 2018; Xie et al, Nature Immunology 2020), N1/N2 neutrophils as described in cancer (Fridlender Cancer Cell 2009; this paper reports on TGFb effects on neutrophils that are relevant to this manuscript), or Siglefhigh tumor-associated neutrophils, which relate to osteoblasts (Zilionis et al Immunity 2019; Engblom et al. Science 2017), and so on. Most of these papers have publicly available data that allows looking into TGFb1 expression among neutrophils (also see https://kleintools.hms.harvard.edu/tools/springViewer_1_6_dev.html?datasets/mouse_HPCs/basal_bone_marrow/full).

From the data, it is not entirely clear, why the authors only focused on neutrophils vs monocytes? The total number and frequency of TGFb1+ monocytes is still comparable to neutrophil numbers/frequency in old mice, and the monocytes even increase in numbers. This reviewer understands that one has to limit the scope of the paper, but the authors should acknowledge the possible contribution of

monocytes to their proposed mechanism. It is also possible that the monocytes versus neutrophils are physically more proximal to the MPC in the bone marrow, which could affect their relative ability to modulate MPC differentiation/osteoblastic bone formation. (A recent paper by Lin et al. (Bone Research 2021) showed that alveolar bone-derived monocytes/macrophages could stimulate osteogenesis in vitro, which could be relevant to discuss in relation to the authors' findings).

The authors show that several different chemokine increase in the bone marrow with age, but end up focusing on CCL5 after running a protein array. The authors should include the CCL12, CXCL12, and CCL2 protein array data since these are well known mediators of neutrophil chemotaxis and important to show how they change with age given the topic of the article. Also, positive (reference spots) and negative controls for the protein array should be included in supplement. The mechanism of age-related neutrophil accumulation in the bone marrow likely involve several chemokines and could also be related to retention, proliferation, cell turnover, in addition to recruitment of blood neutrophils.

What is the expression level of CCR5 in TGFb1 negative neutrophils? What other cells than neutrophils express CCR5 in the bone marrow? The effect of the CCR5 inhibitor presented in figure 5 are rather mild and it is not clear whether the effect of maraviroc on bone mass and neutrophils are related to each other, separate, or due to the involvement of other cells. '

Figure 2: What happens to osteoblast differentiation when TGFb is blocked during MPC co-culture with neutrophils? The authors show that TGFbRII is needed on the MPC, but adding data on whether TGFb1 expression specifically by neutrophils is needed to suppress osteoblast differentiation in vitro or in vivo would improve the manuscript. [L SEP]

Figures 3 and especially 6 provide compelling in vivo results relating TGFbRII and TRAF3 expression by Prx1+ cells to bone mass. However, the authors also report that TGFb1-expressing neutrophil numbers are increased or decreased in conditionally TGFbRII-deficient or TRAF3-deficient mice, respectively. These results are correlative and not causative; ie the authors have not shown that the numbers of TGFb1-expressing neutrophils are functionally related to the bone phenotypes (it could be due to other reasons). How specific is Prx1-targeted gene deletion to MPC? Is TGFbRII/TRAF3 expression completely absent in MPC from these conditional knockout mice? Is TGFbRII/TRAF3 expression normal in non-MPC cells in the Bone marrow? Is TGFbRII deleted in other cells? The authors can also refer to previously published articles if these conditional knockout mice have been thoroughly characterised previously. In general, the manuscript would benefit from a bit more discussion on alternative explanations to their data.

Figure 6: title of figure and text appears incorrect. The title says that "Mice with TGFbRII deleted specifically in mesenchymal lineage cells have decreased numbers of TCNs in BM and increased bone loss". However, the data shows a clear increase in bone volume and mass, as well as a decrease in osteoclasts, which is also discussed in the text.

Minor comments

Figure 1A: stats between bone and BM cells. It is not exactly clear how these values were normalized.

Figure 1D: missing x and y axis labels

Order of figures: the authors should present the panels for each figure in order (eg Fig 1e and f should be presented before Fig 1g). Could simply re-label the figures to match the text.

Fig 3f-g: it is not clear from the figure if these cell populations were pre-gated on TGFb1+ cells. Please clarify in the legend and figure.

The use of acronyms: if only mentioned once, there is no need for an acronym, for example "LLCI". Also the use of many acronyms in the abstract makes it somewhat difficult to follow.

Use of dash in abstract (treat/prevent) should be avoided.

Edit sentence in the introduction: "monocytes are myeloid lineage cells that express" Instead more appropriate to refer to all myeloid cells, introduce the subsets, and then to specify whether it is known or unknown which ones make TGFb1 and that you sought to answer this question (TGFb1 expression is not a prominent feature that is used to define monocytes). Additionally, monocytes do consist of multiple subsets (Ly6Chigh vs low, etc) but these are not the focus of the paper and therefore may not need to be discussed in the introduction.

We thank the reviewers for their careful and helpful review of our manuscript and Figures and for their suggestions to improve our paper. We have addressed these suggestions below in blue text. We have made changes in the manuscript in red text, leaving the original changes in blue so that the previous and new changes are easily identified.

Reviewer #1 (Remarks to the Author):

NCOMMS-21-19687

Novel TGF β +CCR5+ neutrophil population causes age-related osteoporosis in mice

In a line of studies by this research group, they have demonstrated that TGF β 1 is a key molecule for inducing age-related bone loss through which it promotes degradation of TRAF3 in bone cells. This study identified a novel subset of TGF β 1+CCR5+ neutrophils (TCNs) as the major source of TGF β 1 in murine bone, of which increase in bone likely contributes age-related bone loss. This finding should be highly admired. However, there are some concerns in logic flow based on unsound evidence and somehow subjective interpretations. Specific comments are listed below.

Major comments:

In co-culture experiments shown in Fig.2, effects of Ly6G+ BM cells on osteoclastogenesis and bone resorption were not assessed. RANKL/OPG ratio in the coculture, and N.Oc (or at least one osteoclastic parameter) in the implant should be scored. Furthermore, the effects of TGF-beta inhibitor or its neutralization Abs on osteoblastogenesis and osteoclastogenesis are recommended to be conducted. These experiments will provide more sound evidence for reaching the message as represented by the title.

Thanks for these suggestions. We have tested the effects of TGF β -expressing neutrophils (TNs) on OC formation along with treatment with the neutralizing Ab, 1D11 in co-culture experiments and have presented the data in Fig. 2f & 2g. We added a statement in the Results on page 6, "In addition, TNs from aged mice stimulated osteoclast formation from progenitor cells with no changes in the ratio of *Rankl/Opg* transcription levels, and this was inhibited by 1D11 (Fig. 2f-g, Suppl. Fig. 1b).".

In addition, we have measured osteoclastogenesis in the implants and the data are shown in Fig. 2j & 2k. We added a statement in the Results on page 7, "Osteoclastic bone resorption in this new bone was significantly higher when WT MPCs were co-implanted with TNs from aged than from young mice (Fig. 2j-k), while this stimulation of osteoclast formation caused by aged TNs was significantly reduced when the cells were co-implanted with bone-derived MPCs from TRII-cKO mice (Fig. 2j-k).".

To determine the effects of the TGF β neutralization Ab on osteoblast differentiation, we co-cultured TGF β -expressing neutrophils from aged mice with MPCs from WT and TGF β RII-cKO mice, or with WT MPCs plus 1D11. The data are presented in Fig. 2d & 2e and we added a statement in the Results on page 6, "To test if TGF β -expressing neutrophils (TNs) regulate osteoblast differentiation directly and if this process functions

through TGF β , we sorted TNs (TGF β 1⁺CD11b⁺Ly6C⁺6G⁺) from 3- and 20-m-old C57 male mice using FACS, and co-cultured them with MPCs from BM of 3-m-old C57 mice. We found that TNs from aged mice more efficiently inhibited osteoblast differentiation than TNs from young mice, and that this inhibition was blocked by 1D11, a TGF β neutralizing antibody (Fig. 2d).”.

To test if this process functions through TGF β signaling in MPCs, we generated mice with conditional KO of TGF β receptor II in mesenchymal lineage cells by crossing TGF β RII^{fl/fl} with Prx1^{Cre} mice (TGF β RII^{fl/fl};Prx1^{Cre}, which we call TRII-cKO mice). We added this comment on page 6, “We found that TNs sorted from aged C57 mice effectively inhibited osteoblast differentiation from WT MPCs, but not from TRII-cKO MPCs (Fig. 2e).”.

Functional evidence for the involvement of CCL5-CCR5 axis in the entire data set appears to be unsound. It is probable that functional dissociation of a given chemokine and its receptor from others encounter experimental difficulties because of their multiple cellular targets and functional redundancies. Nevertheless, more experimental evidence and unbiased discussion would strengthen this research work.

We appreciate this concern. CCL5 has four different chemokine C-C motif receptors (CCRs): CCR1, CCR3, CCR4 and CCR5. To examine for functional involvement of CCL5-CCR5 and possible roles of other chemokines, we tested the migration velocity of TGF β -expressing neutrophils (TNs) in response to multiple chemokines in Fig. 5h. We found that, in response to CCL4, CCL5 and CXCL12, TNs from aged C57 mice were attracted at a similar speed to that as cells from young C57 mice (Fig. 5h; Supplementary Fig. 3c). However, TNs were attracted faster in response to CCL4 and CCL5, but not to CXCL12, than vehicle-treated cells (Fig. 5h). We added a statement in the Results on page 10, “To examine if there are differences in cell migration of TNs from young and aged mice, we performed a cell migration assay using chemokines. We found that, in response to CCL4, CCL5 and CXCL12, both neutrophils and TNs from aged C57 mice were attracted at a similar speed as cells from young C57 mice (Fig. 5h; Suppl. Fig. 7c). In contrast, in response to CCL4 and CCL5, but not to CXCL12, TNs from both young and aged mice were attracted faster than vehicle-treated cells. (Fig. 5h).” These data suggest that TNs express high levels of CCR1 or CCR5 because both CCRs are co-receptors for CCL4 and CCL5. Further investigation will be required to determine if CCR1 plays a critical role in this process. However, we believe that our data convincingly demonstrate that blockade of CCR5 by maraviroc effectively limited TCN migration in vivo (Fig. 7a&7b) and in vitro (Fig. 5l&5m) and prevented bone loss in the NSG model (Fig. 6) and aged C57 mice (Fig. 7). Our data support our posit that the CCL5-CCR5 axis is necessary for TCN recruitment into mouse BM during aging, although it is likely not the only axis.

In data sets shown in Fig.4, validation of mRNA and proteins of Ccl-12, Cxcl12, Ccl24 as well as Cxcl4, Ccl26 and Ccl2 should be demonstrated, though some of them are not significantly changed.

The data in the original Fig. 4a (Fig. 5a in the revised version) show Real-time qPCR

test results in a heatmap, but we did not do mRNA Seq to make further mRNA expression validation of these genes. However, we have measured protein expression of these genes to strengthen our findings. In the antibody array in Fig. 5b and 5c, we also tested CCL12, CXCL12 and CCL2 protein levels in addition to CCL5 and found no difference in the levels of these three proteins. We were unable to examine protein levels of CCL24, CXCL4 and CCL26, which were not present in this array, because we had exhausted the material doing additional experiments. Thus, further investigation will be required to fully address this issue. We added this statement on page 9, "However, protein levels of CCL12, CXCL12 and CCL2 were very low, and no obvious changes were observed (data not shown; CCL24, CXCL4 and CCL26 were not included in the array). We next performed ELISA assay and confirmed that CCL12, CXCL12 and CCL2 protein levels were comparable between 8-month old WT and P-cKO mice. Protein levels of CCL12, but not CXCL12 or CCL2, were significantly higher in BM of 18-month-old mice than 6-month-old mice (Suppl. Fig. 6).".

In in vitro cell movement assay shown in Supplementary Fig.2c, the effect of Maraviroc treatment will be informative to see how CCR5 specifically contributed to this functional assay.

Thanks for this suggestion. To determine the specific contribution of CCR5 in promoting TCN migration toward MPCs, we tested the effects of maraviroc on TCN migration. The data from cell migration assays in transwell experiments are presented in Fig. 5l & 5m. We added a statement in the Results on page 10, "To determine if CCR5 expression by TNs facilitated their recruitment toward MPCs in P-cKO mice, we next performed transwell experiments and found that MPCs from P-cKO mice attracted significantly more TNs than MPCs from WT mice. Of note, the increased attraction induced by P-cKO MPCs was efficiently blocked by addition of maraviroc, an FDA-approved CCR5 antagonist (Fig. 5l-m).".

To further assess functional contribution of CCR5-mediated signal for TCNs to highly accumulate in bone marrow, assays to see the impact of CCR5 signal on cell survival should be designed.

Thanks for this suggestion. We have tested the effects of CCR5 blockade by maraviroc on viability of TCNs that are recruited from the peripheral blood into BM. We sorted TCNs from BM of 20-22-month-old ROSA^{mTmG} male mice and injected 3 million of these sorted cells into immune-deficient NSG mice via tail vein (1 million/d for 3 d). We found that NSG recipients treated with maraviroc had significantly fewer donor-derived tdTomato⁺Ly6G⁺ cells in BM than vehicle-treated recipients. The percentages of Ki-67⁺ cells (proliferating) and p16⁺ cells (senescent) in donor-derived tdTomato⁺ TCNs in BM of maraviroc-treated NSG recipients were comparable with those in vehicle-treated recipients (Suppl. Fig.5). These findings suggest that maraviroc prevented TCN accumulation into BM from peripheral blood, but it did not cause significant changes in the proliferation or senescence status of these accumulated TCNs. We described the generation and treatment of these mice in the Methods on page 23,

“NSG mice injected with sorted TGFβ/CCR5-expressing neutrophils (TCNs)

3- and 20-22-m-old C57 and ROSA^{mTmG} male mice were sacrificed for BM TCN isolation. TGFβ1-expressing BM cells were first enriched through positive selection using a magnetic isolation system with PE-conjugated anti-TGFβ1 Ab as primary and anti-PE MicroBeads as secondary labeling. Ly6G-positive cells or CCR5/Ly6G double-positive cells were further sorted by flow from the positively-selected TGFβ1-expressing BM cells. Male NSG mice, aged 3-mon-old were injected with one million TCNs from young or aged mice via tail vein every other day for a total of 3 injections. NSG recipients injected with aged TCNs were treated with maraviroc (2 mg/kg body weight) by I.P. injection once per day from the day of the first TCN injection for 10 days and the mice were sacrificed the following day for bone phenotype analysis.”. We added a statement in the Results on page 11, “To determine the effects of maraviroc on TCN recruitment into BM and on bone homeostasis, we injected TCNs from BM of aged ROSA^{mTmG} male mice into NSG mice and treated the mice with vehicle or maraviroc. We found that NSG mice treated with maraviroc had significantly fewer donor-derived TCNs in BM than vehicle-treated mice (Suppl. Fig. 10a-d), which was not associated with altered Ki-67-related cell proliferation (Suppl. Fig. 10e-f), or p16-related cell senescence (Suppl. Fig. 10g-h). Of note, maraviroc-treated recipients had significantly higher trabecular bone volume (BV/TV; Fig. 6a-b), trabecular number (Tb.N; Fig. 6c) and bone mineral density (BMD; Fig. 6e), and lower trabecular separation (Tb.Sp; Fig. 6d) than vehicle-treated recipients. However, no significant changes were observed in the thickness of trabecular or cortical bone between NSG recipients treated with vehicle or maraviroc (Suppl. Fig. 10i-j). Histomorphometric analysis showed that maraviroc-treated recipients had higher osteoblast surfaces (Fig. 6f&g) and lower osteoclast numbers and surfaces on trabecular surfaces in tibial metaphyses (Fig. 6f, h&i). These findings reveal that blockade of CCR5 by maraviroc efficiently prevented recruitment of donor-derived TCNs into BM and bone loss in NSG recipients.”.

As for the in vivo and vitro experiments of Maraviroc treatments shown in Fig.5 and Supplementary Fig.4, Lee et al. previously demonstrated the effects of Maraviroc on human osteoclast differentiation, and the impact of Ccr5-deficiency in osteoclastogenesis and bone with multiple careful assays (Nat Commun. 2017 Dec 20;8(1):2226. doi: 10.1038/s41467-017-02368-5), therefore, it sounds quite naïve to describe the sentence of “In contrast, maraviroc has no significant direct effects on osteoblast or osteoclast formation in vitro (Supplementary Fig. 4a & 4b), suggesting that its effects on osteoclast and osteoblast numbers in vivo are mediated by other cell types.” Differentiation and functional regulation of osteoblasts and osteoclasts are multiple steps, merely counting ALP+ cell area or TRAP+ cell area do not exclude the effects of CCR5-blockade on these bone cells, therefore, more detailed histomorphometry would be required. Genetic deletion of Ccr5 and Maraviroc treatment may exhibit somehow distinct bone phenotypes, Maraviroc may be less effective in mouse cells since designed targeting human CCR5. These points would also be taken for further consideration in the interpretation and discussion of these experimental data. Previously published works of CCL5 -CCR5 (and other CC-chemokines and receptors, if necessary) in the function of bone cells should be

fairly referenced and discussed.

Thanks for these helpful comments and suggestions. As Lee et al. reported in *Nat Commun* in in-vitro experiments, anti-human CCR5 antibody treatment stimulated osteoclast formation during the early stage (Day 0 to Day 2), but it inhibited osteoclast formation during the later stage (Day 2 to Day 6) of these assays. In addition, osteoclast precursors from CCR5 global knockout mice generated fewer osteoclasts than cells from WT mice. As we presented in Supplementary Fig. 6, the area of TRAP⁺ osteoclasts was reduced in the 100 ng/ml maraviroc-treated group vs. vehicle group, but only when these two groups were compared using an inappropriate unpaired Student's t test ($p=0.0454$) for these multiple group comparisons. In contrast, no significant changes were detected in the multiple maraviroc-treated groups vs. vehicle group when one-way ANOVA with Tukey's post-hoc test was used appropriately for statistical analysis. Therefore, our data may be partially consistent with Lee's findings on inhibition of osteoclast formation, but maraviroc could have effects different from anti-human CCR5 neutralizing Ab, since the binding sites are different (Chang, X.L., et al. *Nat Commun* 12, 3343 (2021). <https://doi.org/10.1038/s41467-021-23697-6>) and CCR5 neutralizing Ab probably affects the viability of targeted cells, given antibody-mediated cell depletion is common. As Lee et al. pointed out in the Introduction of their paper, the effects of CCR5 signaling in bone are controversial and contradictory.

Ccr5 global KO mice and mice treated with maraviroc have distinct bone phenotypes: 7-week-old Ccr5 KO mice had normal BV/TV, while the 22-m-old C57 mice we treated with maraviroc for 1 month had increased BV/TV. Because CCR5 is preferentially expressed in T cells, monocytes, and macrophages, global genetic deletion of CCR5 from the embryonic stage could affect many more types of cells in the bone environment than bone cells alone. Global genetic deletion of Ccr5 could result in phenotypes different from the effects of short-term blockage of CCR5, and the difference in mouse ages increases the complexity. Interestingly, Lee's group found that Ccr5 KO mice have normal BV/TV, but the KO effectively prevented RANKL-induced bone loss, suggesting that RANKL-mediated osteoclastic bone resorption is CCR5-mediated. In this study, we propose that osteoclastic bone resorption in aged mice causes TRAF3-dependent recruitment of TCNs into BM, mediated by CCL5/CCR5 and leading to bone loss (Fig. 9). Our findings are compatible with the interesting phenotype of CCR5 KO mice treated with RANKL.

To address these concerns, we added a statement in the Discussion on page 15:

"Lee et al. reported that a CCR5 Ab inhibited the function, but not formation, of human osteoclasts, and that Ccr5 global knockout mice have dysfunctional osteoclasts⁵⁵. However, they also noted that the role of CCL5/CCR5 signaling in bone cells is controversial. The effects we observed with maraviroc could be different from those of an anti-human CCR5 Ab or from genetic global deletion of CCR5, since it binds to a different site on CCR5 from the CCR5 Ab⁵⁶, and the range of its inhibitory effects is different from global CCR5 gene deletion. Lee et al. also reported that CCR5 mediates RANKL-induced bone resorption⁵⁵. Our data suggest that CCL5/CCR5 chemotaxis signaling mediates accumulation of TCNs in BM resulting in enhanced bone resorption and inhibited bone formation, and bone loss

during aging (Fig. 9) compatible with the interesting phenotype of CCR5 KO mice being resistant to RANKL⁵⁵.”.

Bone phenotype of TRII-cKO shown in Fig6. is quite striking. It could be wondered why vertebral bones of T12-L2 showed strong phenotype even though Prx1-driver is specific in limb mesenchyme, possibly needs explanation for this.

Although Prx1 is expressed predominantly in appendicular bones in embryos (ten Berge, D. et al. 1998 Development), it is also expressed in developing vertebrae, and mice with global knockout of Prx1 have defects in the skull, limbs, and vertebrae (ten Berge, D. et al. 1998 Development; Martin J.F. et al. 1995 Genes Dev; Li, J et al. 2019 Nat Common). Prx1Cre mice have been used to conditionally delete genes in MPCs to investigate their roles in skeletogenesis as well as in bone modeling and remodeling (Seo, H. S. et al. 2007 Dev. Biol.). To further address this issue, we generated a Prx1^{Cre};Rosa26^{mTmG} mouse model and confirmed Prx1-driven mesenchymal lineage distribution in vertebrae and limb bones, which is presented in Suppl. Fig. 4. We added a statement in the Results on page 8, “To examine the specificity of targeted gene deletion driven by Prx1Cre, we crossed Prx1^{Cre} with Rosa26^{mTmG} reporter mice, in which Prx1^{Cre}+ cells switch tdTomato to GFP protein expression. We found that Prx1^{Cre}+ GFP cells were very rare in skeletal muscle, but widely distributed in bone cells, including in hypertrophic chondrocytes in growth plates, mesenchymal/osteoblastic cells on metaphyseal bone surfaces, and a relatively small fraction of osteocytes and spindle cells in BM (Suppl. Fig. 4a-4c), which is consistent with the profiles of Prx1-tracing mesenchymal lineage cells reported by others⁴⁶. In addition, osteoblastic cells on surfaces of newly generated bone surfaces were GFP-positive in fracture callus of 3-m-old Prx1^{Cre};Rosa26^{mTmG} mice on Day14 post-fracture, consistent with the osteogenic potential of the Prx1^{Cre}-tracing cells in our mouse model (Suppl. Fig. 4d). We also found significantly higher Cre transcription levels (~50-700x) in vertebrae of Prx1^{Cre} mice than in WT mice from 2 wk to 12-mon of age (Suppl. Fig. 4e). Many Prx1^{Cre}+ GFP cells were present on endosteal and trabecular bones surfaces of 3-m-old Prx1^{Cre};Rosa26^{mTmG} mice (Suppl. Fig. 4f). We also found that Prx1^{Cre} drove *Traf3* gene deletion in MPCs, but not in B cells, T cells, leukocytes, or osteoclast precursor cells, in BM of 15-m-old P-cKO mice (Suppl. Fig. 5). These findings verify the specificity of Prx1-driven gene deletion in mesenchymal lineage cells and suggest that TRAF3 in mesenchymal lineage cells limits the numbers of TGFβ1+ neutrophils in BM and bone loss in adult mice.”.

Related to this, some in vitro functional experiments to assess osteogenic potential of TRII-deficient mesenchymal cells such as co-culture with TCNs as shown in Fig.1 will strengthen the message.

To address this concern, we added in-vitro functional experiments in which MPCs from TRII-cKO and WT littermates were co-cultured with TGFβ-expressing neutrophils sorted from BM of aged C57 mice. We found that the osteogenic potential of MPCs from TRII-cKO and WT mice was similar, and that TGFβ-expressing neutrophils efficiently inhibited

osteoblast differentiation from WT MPCs, but not from TRII-cKO MPCs. We added a statement in the Results on page 6, “We found that TNs sorted from aged C57 mice effectively inhibited osteoblast differentiation from WT MPCs, but not from TRII-cKO MPCs (Fig. 2e). In addition, TNs from aged mice stimulated osteoclast formation from progenitor cells with no changes in the ratio of *Rankl/Opg* transcription levels, and this was inhibited by 1D11 (Fig. 2f-g, Suppl. Fig. 1b).”.

Minor points:

Line 112 and M&Ms, “mTmG” and/or ROSA mTmG needs more specific explanation for readers.

We have added a statement in the Methods on page 19 to explain, “ROSA^{mT/mG} mice (Jackson Lab #007676) have cell membrane-localized tdTomato (mT) fluorescence expressed widely in cells/tissues prior to Cre recombination, while, in Cre recombinase-expressing cells (and future cell lineages derived from these cells), cell membrane-localized EGFP (mG) fluorescence expression will be triggered and replace the red fluorescence.”.

Line 165, TGFβ1+CCR5+ neutrophils TCNs for short may be rephrased as TGFβ1+CCR5+ neutrophils (TCNs for short).

It has been rephrased accordingly.

Lines 182, and Fig.6 legend title, “increased bone loss” may be rephrased as “increased bone mass” or else.

We thank reviewers for spotting out this error. We have revised it to “increased bone mass”.

Reviewer #2 (Remarks to the Author)

Low-level chronic inflammation caused by aging, inflammaging, is considered one of the causes of age-related bone loss, osteoporosis. In the present study, the authors demonstrated that neutrophils are the major source of TGFβ in bone marrow in aged mice, and TGFβ-expressing neutrophils from aged mice inhibited osteoblast differentiation from mesenchymal progenitor cells (MPCs). Conditional deletion of TGFβ receptor II (TRII) in cells of mesenchymal lineage abrogated the inhibitory effect of TGFβ-expressing neutrophils on osteoblast differentiation. TGFβ is implicated in the degradation of TRAF3, and TRAF3 deletion in mesenchymal lineage cells reduced osteoblast differentiation and caused osteoporosis. The authors found that CCL5 levels are high in bone marrow of TRAF3 cKO mice, and the expression of CCL5 mRNA was high in MPCs from aged mice. The expression of CCR5, the CCL5 receptor, was high in TGFβ1+ neutrophils, and the increase in the number of TGFβ1+CCR5+ neutrophils (TCNs) were observed in bone marrow of aged mice. CCR5 antagonist maraviroc increased bone formation and reduced osteoclast number in aged mice, while maraviroc has no direct effects on osteoclasts or osteoblasts in vitro. TRII-cKO mice exhibited decreased number of TCNs in bone marrow

and increased bone volume as compared with wild type littermates. Based on these findings, the authors concluded that TCNs attracted to bone marrow through the CCL5-CCR5 axis suppressed bone formation and promoted bone resorption by TGF β -TRII pathways in aged mice. The experiments are carefully performed, and the conclusion appears sound. However, there are several points to be addressed.

Comments

1) As the authors commented, Seo et al. previously reported that Prx1-Cre-mediated conditional deletion of TGF β RII resulted in defective limb development (reference #45). This appears reasonable since Prx1 enhancer is mainly activated in the early limb bud mesenchyme (and in a subset of craniofacial mesenchyme) (Logan et al., *Genesis*. 2002 Jun;33(2):77-80; also see The Jackson Laboratory website <https://www.jax.org/strain/005584>). Therefore, the lack of abnormalities in the limbs, in contrast to the report by Seo et al, raises serious concerns regarding the specificity of the Prx1-Cre mice that the authors used. The authors should confirm that Cre is specifically expressed in MPCs by crossing the mice with Rosa26 reporter mice, for example. It is also critical if TGF β RII and TRAF3 is in fact specifically deleted in MPCs of “vertebrae” (not long bones) in their cKO mice since Prx enhancer is usually not activated at axial bones.

To address this issue and to trace Cre expression driven by the Prx1 promoter in the Prx1^{Cre} mouse model we used, we crossed Prx1^{Cre} with Rosa26^{mTmG} reporter mice, in which Prx1-Cre-expressing cells (and future cell lineages derived from these cells) express membrane-localized EGFP (mG) fluorescence replacing the red fluorescence. We confirmed Prx1 expression in limb mesenchyme. We tested Cre expression driven by the Prx1 promoter in vertebrae and added a statement in the Results on page 8, “To examine the specificity of targeted gene deletion driven by Prx1Cre, we crossed Prx1^{Cre} with Rosa26^{mTmG} reporter mice, in which Prx1^{Cre+} cells switch tdTomato to GFP protein expression. We found that Prx1^{Cre+} GFP cells were very rare in skeletal muscle, but widely distributed in bone cells, including in hypertrophic chondrocytes in growth plates, mesenchymal/osteoblastic cells on metaphyseal bone surfaces, and a relatively small fraction of osteocytes and spindle cells in BM (Suppl. Fig. 4a-c), which is consistent with the profiles of Prx1-tracing mesenchymal lineage cells reported by others⁴⁶. In addition, osteoblastic cells on surfaces of newly generated bone surfaces were GFP-positive in fracture callus of 3-m-old Prx1^{Cre};Rosa26^{mTmG} mice on day14 post-fracture, consistent with the osteogenic potential of the Prx1^{Cre}-tracing cells in our mouse model (Suppl. Fig. 4d). We also found significantly higher Cre transcription levels (~50-700x) in vertebrae of Prx1^{Cre} mice than in WT mice from 2 wk to 12-mon of age (Suppl. Fig. 4e). Many Prx1^{Cre+} GFP cells were present on endosteal and trabecular bones surfaces of 3-m-old Prx1^{Cre};Rosa26^{mTmG} mice (Suppl. Fig. 4f). We also found that Prx1^{Cre} drove *Traf3* gene deletion in MPCs, but not in B cells, T cells, leukocytes, or osteoclast precursor cells, in BM of 15-m-old P-cKO mice (Suppl. Fig. 5). These findings verify the specificity of Prx1-driven gene deletion in mesenchymal lineage cells and suggest that TRAF3 in mesenchymal lineage cells limits the numbers of TGF β 1+ neutrophils in BM and bone loss

in adult mice.”.

2) To further confirm the central role of TCNs in the bone loss in aged mice, the authors should perform transplantation of TNCs from old mice into young mice.

To address this issue and elucidate the role of TCNs in the bone loss that occurs during aging, we sorted TGF β 1-expressing neutrophils (TNs) from BM of young and aged mice and infused these cells into young immune-deficient NSG mice. The results of these experiments are presented in Fig. 3 and described on page 7, “To examine if TNs have systemic effects on bone homeostasis, we sorted TNs from BM of 3- and 20-mon-old male C57 mice and transplanted them into 3-mon-old NSG male mice via tail vein. We found that NSG recipients transplanted with either young or aged TNs developed significant bone loss (Fig. 3a). Of note, NSG recipients transplanted with aged TNs had significantly lower trabecular bone volume (Fig. 3b) and trabecular number (Fig. 3c) and increased trabecular separation (Fig. 3d) than NSG recipients transplanted with young TNs; however, no significant changes were observed in trabecular or cortical bone thickness (Suppl. Fig. 2a and 2b). Consistent with this, NSG recipients transplanted with either young or aged TNs had significantly fewer osteoblasts and more osteoclasts on trabecular bone surfaces than NSG mice without transplantation (Fig. 3e-h). NSG recipients transplanted with aged TNs had significantly more osteoclasts than recipients transplanted with young TNs (Fig. 3g-h).”. In addition, we also sorted TGF β 1⁺CCR5⁺ neutrophils (TCNs) from aged ROSA^{mTmG} mice and injected these cells into NSG mice following treatment with vehicle or maraviroc. The results of these experiments are presented in Fig.6 and described on page 11, “To determine the effects of maraviroc on TCN recruitment into BM and bone homeostasis, we injected TCNs from BM of aged ROSA^{mTmG} male mice into NSG mice and treated the mice with vehicle or maraviroc. We found that NSG mice treated with maraviroc had significantly fewer donor-derived TCNs in BM than vehicle-treated mice (Suppl. Fig. 10a-d), which was not associated with Ki67-related cell proliferation (Suppl. Fig. 10e-f), or p16-related cell senescence (Suppl. Fig. 10g-h). Of note, maraviroc-treated recipients had significantly higher trabecular bone volume (BV/TV; Fig. 6a-b), trabecular number (Tb.N; Fig. 6c) and bone mineral density (BMD; Fig. 6e), and lower trabecular separation (Tb.Sp; Fig. 6d) than vehicle-treated recipients. However, no significant changes were observed in the thickness of trabecular or cortical bone between NSG recipients treated with vehicle or maraviroc (Suppl. Fig. 10i-j). Histomorphometric analysis showed that maraviroc-treated recipients had higher osteoblast surfaces (Fig. 6f-g) and lower osteoclast numbers and surfaces on trabecular surfaces in tibial metaphyses (Fig. 6f, h-i). These findings reveal that blockade of CCR5 efficiently prevents recruitment of donor-derived TCNs into BM and bone loss in NSG recipients.”. These findings indicate a critical role of TCNs to cause bone loss that is associated with a decrease in osteoblasts and an increase in osteoclasts.

3) What is the mechanism underlying the increased bone resorption in P-cKO mice?

P-cKO mice with TRAF3 specific deletion in mesenchymal lineage cells develop early onset osteoporosis, which is associated with inhibition of osteoblastic differentiation

through reduced b-catenin signaling. In addition, TRAF3 deletion in mesenchymal lineage cells stimulates NFkB-dependent RANKL transcription by MPCs, which promotes osteoclastogenesis and bone loss. We published these findings in 2019 (Li, J. 2019 Nat Commun).

Reviewer #4 (Remarks to the Author):

Summary

With their manuscript entitled “Novel TGFβ⁺/CCR5⁺ neutrophil population causes age-related osteoporosis in mice”, Li et al aims to provide a cellular and molecular mechanism, involving neutrophils, TGFβ1/TGFβRII, TRAF3, and mesenchymal progenitor cells, that can, at least in part, explain age-related osteoporosis. The authors show that TGFβ1-expressing neutrophils increase with age in mice and that bone marrow neutrophils from older mice, versus from younger mice, suppress bone formation to a higher degree in vitro and in vivo using an ectopic bone formation assay. The authors also provide evidence that the neutrophil-induced suppression of bone formation was mediated through TGFβRII. Moreover, the authors show that mice with conditional deletion of TRAF3 results in decreased bone mass and increased TGFβ1-expression neutrophils in the bone marrow. The authors further aim to link an age-related increased bone marrow CCL5 gene and protein expression to the accumulation of TGFβ1-expressing neutrophil in the bone marrow. Additionally, the authors show that therapeutically targeting CCR5, which is expressed by neutrophils (and likely other cells) and is one of the receptors for CCL5, modestly increases bone mass in aged mice, while simultaneously slightly decreasing TGFβ1-expressing neutrophils in the bone marrow. Finally, the authors show that conditionally deleting TGFβII in Prx1-expressing cells (which reportedly is specific to mesenchymal progenitor cells) results in substantially increased bone mass and decreased TGFβ1-expressing neutrophils. The manuscript contains many interesting and thought-provoking results that could help clarify the underlying mechanisms of age-related osteoporosis (which is complex and the likely result of diverse biological processes that may differ between individuals). The authors use several different mouse models and a variety of tools and experimental readouts to investigate their research question. The manuscript is well written and the results communicated clearly, for which the authors should be commended. However, there are a few key issues that need to be addressed prior to publication. Most of these comments can be addressed through experiments that have already been performed, adjusting the text, or by relating their work to publicly available datasets.

Major comments

Apart from figure 2, the authors show mainly correlative connections between TGFβ1-expressing neutrophil increases or decreases and bone volume; i.e. the authors lack sufficient data to back up the title of the paper, which states that TGFβ1⁺ neutrophils cause age-related osteoporosis in mice. In figure 2, it is also unclear from the figure whether the

authors used TGFb1-expressing neutrophils in these experiments or not. The authors should consider adjusting the title of the paper to more accurately reflect the data.

Thanks for these comments. Neutrophils (Ly6G⁺) were used in Fig. 2a-c, while TGFβ1-expressing neutrophils (TN) were used in all other experiments presented in Fig. 2. We agree with the reviewer's concerns, and hence we performed additional experiments using co-culture (Fig. 2d-g), ectopic implantation (Fig. 2h-k) and adoptive transfers (Fig. 3 and Fig. 6) to verify the causative effects of TNs on bone loss and to support the assertion in the title.

In Fig. 2, new co-culture experiments were added and presented in the Results on page 6, "To test if TGFβ-expressing neutrophils (TNs) regulate osteoblast differentiation directly and if this process functions through TGFβ, we sorted TNs (TGFβ1⁺CD11b⁺Ly6C⁺6G⁺) from 3- and 20-mon-old C57 male mice using FACS, and co-cultured them with MPCs from BM of 3-mon-old C57 mice. We found that TNs from aged mice more efficiently inhibited osteoblast differentiation than TNs from young mice, and that this inhibition was blocked by 1D11, a TGFβ neutralizing antibody (Fig. 2d)", and, also on page 6, "We found that TNs sorted from aged C57 mice effectively inhibited osteoblast differentiation from WT MPCs, but not from TRIL-cKO MPCs (Fig. 2e). In addition, TNs from aged mice stimulated osteoclast formation from progenitor cells with no changes in the ratio of *Rankl/Opg* transcription levels, and this was inhibited by 1D11 (Fig. 2f-g, Suppl. Fig. 1b)." Consistent with this, the results of ectopic implantation showed that the volume of new bone generated from WT bone-derived MPCs was significantly lower and osteoclast formation was significantly higher when they were co-implanted with TNs from aged than from young mice (Fig. 2h & 2i).

We also performed new experiments using adoptive transfers and presented the data in Fig. 3 and described the findings on page 7, "To examine if TNs have systemic effects on bone homeostasis, we sorted TNs from BM of 3- and 20-mon-old male C57 mice and transplanted them into 3-mon-old NSG male mice via tail vein. We found that NSG recipients transplanted with either young or aged TNs developed significant bone loss (Fig. 3a). Of note, NSG recipients transplanted with aged TNs had significantly lower trabecular bone volume (Fig. 3b) and trabecular number (Fig. 3c) and increased trabecular separation (Fig. 3d) than NSG recipients transplanted with young TNs; however, no significant changes were observed in trabecular or cortical bone thickness (Suppl. Fig. 2a-b). Consistent with this, NSG recipients transplanted with either young or aged TNs had significantly fewer osteoblasts and more osteoclasts on trabecular bone surfaces than NSG mice without transplantation (Fig. 3e-3h). NSG recipients transplanted with aged TNs had significantly more osteoclasts than recipients transplanted with young TNs (Fig. 3g-h)." Of note, the NSG recipients injected with aged TNs developed osteoporosis to a greater extent than recipients with young TNs, which was associated with a greater degree of inhibition of osteoblast differentiation and higher promotion of osteoclast formation. With these additional experiments, we believe that our in vivo and in vitro data support causative effects of TNs in age-related osteoporosis.

What is the proportion of neutrophils that express TGFb1? i.e. what does it look like if

instead of pre-gating on TGFb1, the authors first gate on neutrophils, then look at TGFb1 expression. Is it a gradient or two split populations (high vs low)? Is TGFb1 expression a feature of the majority of bone marrow neutrophils or a smaller fraction? This would be relevant to understand considering that all BM Ly6g+ neutrophils may have been used for Figure 2 for example and to clarify whether it is a large versus small fraction of neutrophils that express TGFb1. If most neutrophils express TGFb1, then calling these cells a neutrophil subset would be less appropriate. Rather, one could consider the scenario that all neutrophils in the bone marrow upregulate TGFb1 expression with age. Also, does TGFb1 expression relate to neutrophil maturity? See the next comment, which is related to this point.

We re-analyzed the flow data, following the reviewer's suggestion. We first gated on neutrophils and tested TGFβ1 expression and found that TGFβ1-expressing cells account for 11.96% and 15.66% of DAPI⁺CD11b⁺Ly6C⁺Ly6G⁺ neutrophils in 2- and 23-mon-old C57 mice, respectively. These data suggest that TGFβ1-expressing neutrophils are a small fraction of the total number of neutrophils in BM. Although the fraction of Ly6C⁺Ly6G⁺ neutrophils in CD11b⁺ cells in mouse BM did not change significantly with age, the percentages of CD11b⁺ cells in BM and of TGFβ1⁺ cells in Ly6G⁺ neutrophils in BM were significantly higher in aged than in young mice. As seen in Fig. 2c, active TGFβ1 expression levels were significantly higher in Ly6G⁺ neutrophils from aged than from young mice. Consistent with this, the percentage of TGFβ1-expressing neutrophils in aged mouse BM was significantly higher than in young mouse BM, no matter whether we pre-gated on neutrophils or on TGFβ1-expressing cells. In terms of TGFβ1 expression by neutrophils, we did observe that TGFβ1 is expressed in a gradient, rather than as a positive population clearly distinguished from a negative one. However, TGFβ1-expressing neutrophils might have higher expression of Ly6G than TGFb1-negative cells, which allows clear isolation of TGFβ1⁺Ly6G^{high} neutrophils from TGFβ1⁺Ly6G^{low} neutrophils, as in Suppl. Fig. 8. The correlation of TGFβ1 and Ly6G expression suggests that TGFβ1 expression in neutrophils may relate to cell maturity, given that Ly6G is a critical marker of neutrophil maturity. We added new data in Suppli. Fig 2 and described the findings on page 5, "We found that TGFβ1⁺ cells comprised ~12% of total BM Ly6C⁺Ly6G⁺ granulocytic cells in 2-mon-old young mice, that this fraction increased significantly to ~16% in aged mice, and that most TGFβ1⁺ cells had high expression of Ly6G, a marker of neutrophil differentiation (Suppl. Fig. 2)".

The authors posit that they have defined a novel neutrophil subset. In general, the authors may want to be more cautious in making this claim. How does the 'new' subset of TGFb1+CCR5+ neutrophils compare to previously reported neutrophil subsets? For example, has any other studies shown that bone marrow neutrophils express TGFb1? and/or express CCR5? What about compared to bone marrow neutrophil states (Evrard et al, Immunity 2018; Xie et al, Nature Immunology 2020), N1/N2 neutrophils as described in cancer (Fridlender Cancer Cell 2009; this paper reports on TGFb effects on neutrophils that are relevant to this manuscript), or Siglehigh tumor-associated neutrophils, which relate to osteoblasts (Zilionis et al Immunity 2019; Engblom et al. Science 2017), and so on. Most of these papers have publicly available data that allows looking into TGFb1 expression among

neutrophils (also see

https://kleintools.hms.harvard.edu/tools/springViewer_1_6_dev.html?datasets/mouse_HP_Cs/basal_bone_marrow/full).

Thank you for these comments. Neutrophils have been historically considered as a homogeneous population. However, as the reviewer points out, in recent years, several maturation stages and subsets with different phenotypic profiles and effector functions have been described both in physiological and pathological conditions, such as infections, autoimmunity, and cancer. CXCRs, including CXCR1, CXCR2 and CXCR4, are highly expressed by neutrophils and mediate directional migration. Mature neutrophils recirculate back into bone marrow for clearance and CXCR4 plays a critical role during this process (Xie et al, Nature Immunology 2020). In contrast, CCRs are less frequently expressed by BM and circulating neutrophils. Nevertheless, CCR5 has been reported to be expressed by neutrophils, and CCR5-expressing neutrophils were found in bronchoalveolar lavage fluid from infected lung and in rheumatoid arthritis joints. In addition, CCR5 is involved in the recruitment of murine neutrophils after the induction of ischemia-reperfusion injury, upon acute endotoxin-induced lung injury and during infectious pneumonia. However, CCR5+ neutrophils in bone marrow remain largely unreported.

Neutrophils are the most abundant leukocytes in the blood. TGF β 1 induces neutrophil recruitment and polarization of tumor-associated neutrophils (TANs). TGF- β blockade slows tumor growth by activating CD8+ cells and macrophages³, stimulating hypersegmentation of neutrophils, and increasing expression of neutrophil-attractants and proinflammatory cytokines⁴. Neutrophils are reported to be a source of TGF β . Airway and blood neutrophils from both asthmatic and normal subjects can express and release TGF β ⁵. Blood neutrophils from asthmatic subjects spontaneously released significantly higher levels of TGF β than those from normal subjects (p=.007)⁵. However, it is unclear if bone marrow neutrophils are an important cellular source of TGF β 1.

In this study, we show that TCNs are a major cellular source of TGF β 1 in murine bone marrow, and that ~90% of TGF β -expressing neutrophils highly express CCR5. We also show functional regulation of bone cell biology *in vitro* and *in vivo* by TCNs by adaptive transfer of this subset into NSG mice. We believe that these TCNs are distinct from any other reported immune cell subpopulation that is involved in the pathogenesis of age-associated osteoporosis. However, we take the reviewer's point and have deleted statements alluding to these being a novel subset, and now simply refer to them being a subset of neutrophils in the revised title and in the paper.

We added a statement in the Discussion on page 15, "CCR5 is involved in recruitment of neutrophils in murine models of ischemia-reperfusion injury³⁴, endotoxin-induced lung injury, and pneumonia³⁵, and neutrophils are a source of TGF β in tumors²⁷, the intestine²⁸ and in asthmatic subjects²⁹. This is the first report of CCR5+ neutrophils being a major cellular source of TGF β in murine BM and important regulators of bone homeostasis, as evidenced by *in vitro* co-culture and *in vivo* ectopic implantation and adoptive transfer of this subset into NSG mice. We believe that these TCNs are distinct from other reported immune cell subpopulations involved in the pathogenesis of age-associated osteoporosis."

From the data, it is not entirely clear, why the authors only focused on neutrophils vs monocytes? The total number and frequency of TGF β 1+ monocytes is still comparable to neutrophil numbers/frequency in old mice, and the monocytes even increase in numbers. This reviewer understands that one has to limit the scope of the paper, but the authors should acknowledge the possible contribution of monocytes to their proposed mechanism. It is also possible that the monocytes versus neutrophils are physically more proximal to the MPC in the bone marrow, which could affect their relative ability to modulate MPC differentiation/osteoblastic bone formation. (A recent paper by Lin et al. (Bone Research 2021) showed that alveolar bone-derived monocytes/macrophages could stimulate osteogenesis in vitro, which could be relevant to discuss in relation to the authors' findings).

Thanks for this suggestion. We should have clarified why neutrophils attracted our attention more than monocytes in this study. For example:

- (1) The percentage of monocytes (Ly6C^{hi}6G⁻) with either surface or intracellular TGF β 1 expression in TGF β 1+CD11b⁺ leukocytes in aged bone marrow was significantly lower than in young bone marrow (Fig. 1d,1e,1f and 1g); the number of TGF β 1+ monocytes (TGF β 1+CD11b+Ly6C^{hi}6G⁻) in bone marrow increased with age, in part, due to the increase in total bone volume accompanied by a significant expansion of total bone marrow cells in mice during aging. In contrast, the percentage of TGF β 1+ neutrophils (TGF β 1+CD11b+Ly6C⁻6G⁺) was higher in aged than in young bone marrow. In addition, we tested both surface and intracellular TGF β 1 expression. The numbers of monocytes (Ly6C^{hi}6G⁻) with surface TGF β 1 expression in bone marrow were comparable to neutrophils (Ly6C⁻6G⁺) during aging (Fig. 1h), while the numbers of monocytes with intracellular TGF β 1 expression in bone marrow were much lower (~25% of the numbers of neutrophils) than neutrophils during aging (Fig. 1i). Clearly, there were many fewer CD11b⁺ leukocytes with surface TGF β 1 expression than leukocytes with intracellular TGF β 1 expression (10⁴ for surface vs. 10⁶ for intracellular) (Fig. 1h and 1i). These data show that TGF β 1+ neutrophils were much more abundant than TGF β 1+ monocytes in BM of aged mice.
- (2) Lin et al.⁹ reported that alveolar bone monocytes/macrophages express a higher level of Oncostatin M than cells in long bones, and this promotes osteogenic differentiation of MSCs. Although both neutrophils and monocytes from bone marrow express TGF β 1, we focused here on studies to investigate if TGF β 1+ neutrophils inhibit bone formation and cause bone loss during aging because age-related osteoporosis is characterized by decreased bone formation, while bone-derived monocytes/macrophage likely stimulate bone formation as Lin et al. reported.
- (3) TGF β 1+ neutrophils, but not TGF β 1+ monocytes, highly express CCR5 (Fig. 5i and 5j), which enables TGF β 1+ neutrophils to be attracted toward increased CCL5 expression in the bone microenvironment of aged WT mice and Prx1^{Cre}TRAF3^{ff} (P-cKO) (Fig. 4f-g) mice. This feature makes TGF β 1+ neutrophils, but not TGF β 1+ monocytes, easily targeted and prevented from accumulating in BM during aging by available drugs, such as FDA-approved maraviroc.

We added a statement in the Discussion on page 14, “We found that monocytes (Ly6C^{hi}6G⁺) also express TGFβ1. However, we focused here on TCNs because they are much more abundant in BM during aging (Fig. 2) and age-related osteoporosis is characterized by decreased bone formation, while bone-derived monocytes/macrophage have been reported to stimulate bone formation³⁷.”

The authors show that several different chemokines increase in the bone marrow with age, but end up focusing on CCL5 after running a protein array. The authors should include the CCL12, CXCL12, and CCL2 protein array data since these are well known mediators of neutrophil chemotaxis and important to show how they change with age given the topic of the article. Also, positive (reference spots) and negative controls for the protein array should be included in supplement. The mechanism of age-related neutrophil accumulation in the bone marrow likely involves several chemokines and could also be related to retention, proliferation, cell turnover, in addition to recruitment of blood neutrophils.

Thanks for this suggestion. We have added experiments to confirm expression levels of CCL12, CXCL12, and CCL2. A statement was added in the Results on page 9, “However, protein levels of CCL12, CXCL12 and CCL2 were very low, and no obvious changes were observed (data not shown; CCL24, CXCL4 and CCL26 were not included in the array). We next performed an ELISA assay and confirmed that CCL12, CXCL12 and CCL2 protein levels were comparable between 8-mon-old WT and P-cKO mice. Protein levels of CCL12, but not CXCL12 or CCL2, were significantly higher in BM of 18-mon-old mice than 6-mon-old mice (Suppl. Fig. 6).”. We also added experiments to determine if BM TCNs were recruited from blood neutrophils and if cell proliferation or turnover was involved in this process. We injected TCNs from BM of aged ROSA^{mTmG} male mice into NSG mice via tail vein. We found that donor-derived EGFP⁺ neutrophils were effectively recruited into BM from peripheral blood, and this recruitment was partially prevented by maraviroc. In addition, Ki-67-related cell proliferation and p16-related cell senescence was measured on cryosections of leg bones of NSG recipient mice. We illustrated these findings in Fig. 6 and Suppl. Fig. 10 and described them in the Results on page 11, “To determine the effects of maraviroc on TCN recruitment into BM and bone homeostasis, we injected TCNs from BM of aged ROSA^{mTmG} male mice into NSG mice and treated the mice with vehicle or maraviroc. We found that NSG mice treated with maraviroc had significantly fewer donor-derived TCNs in BM than vehicle-treated mice (Suppl. Fig. 10a-d), which was not associated with altered Ki-67-related cell proliferation (Suppl. Fig. 10e-f), or p16-related cell senescence (Suppl. Fig. 10g-h). Of note, maraviroc-treated recipients had significantly higher trabecular bone volume (BV/TV; Fig. 6a-b) and trabecular number (Tb.N; Fig. 6c), lower trabecular separation (Tb.Sp; Fig. 6d), and higher bone mineral density (BMD; Fig. 6e) than vehicle-treated recipients. However, no significant changes were observed in the thickness of trabecular or cortical bone between NSG recipients treated with vehicle or maraviroc (Suppl. Fig. 10i-j). Histomorphometric analysis showed that maraviroc-treated recipients had higher osteoblast surfaces (Fig. 6f-g) and lower osteoclast numbers and surfaces on trabeculae in tibial metaphyses (Fig. 6f, h-i). These findings reveal that blockade of CCR5 by maraviroc efficiently prevented recruitment of donor-derived TCNs

into BM and bone loss in NSG recipients.”.

In addition, we have added blots showing reference spots and negative controls in the protein array, as requested, as Suppl. Fig. 6a. Reference spots as positive controls demonstrate the incubation with streptavidin-HRP during the assay procedure.

What is the expression level of CCR5 in TGF β 1 negative neutrophils? What other cells than neutrophils express CCR5 in the bone marrow? The effect of the CCR5 inhibitor presented in figure 5 are rather mild and it is not clear whether the effect of maraviroc on bone mass and neutrophils are related to each other, separate, or due to the involvement of other cells.

These are good points. We analyzed CCR5 expression levels in TGF β 1-negative neutrophils (TGF β 1⁻CD11b⁺Ly6C⁺Ly6G⁺ cells) and found that they were significantly higher than in other TGF β 1-negative leukocytes, including TGF β 1-negative monocyte/macrophages (TGF β 1⁻CD11b⁺Ly6C⁺Ly6G⁻ cells); however, CCR5 expression in TGF β 1-negative neutrophils was significantly lower than in TGF β 1-positive neutrophils. We added these results in a revised section on page 10, “We next tested the expression levels of CCR5, a co-receptor of CCL5 and CCL4, by various Ly6C and 6G subsets of TGF β 1⁺ myeloid cells, including TNs. We found that CCR5 expression by aged TNs was comparable to that in young TNs (Suppl. Fig. 7d), and that CCR5 expression was significantly higher in TGF β 1⁺ neutrophils (Ly6C⁺6G⁺) than in TGF β 1⁺ monocytes (Ly6C^{hi}6G⁻) (Fig. 5i-j; Suppl. Fig. 8a), which are characterized by higher expression of CSF-1R (Fig. 5k; Suppl. Fig. 8b). CCR5 expression by TGF β 1⁻ neutrophils was also higher than TGF β 1⁻ monocytes, but still much lower than TGF β 1⁺ neutrophils (Suppl. Fig. 9). Hence, we called this newly identified neutrophil subset TGF β 1⁺CCR5⁺ neutrophils (TCNs for short).”.

CCR5 is a chemokine receptor that is expressed constitutively by immune cells, including lymphocytes, monocytes, macrophages, and granulocytes, and endothelial cells. Monocytes and macrophages that are enriched in the Ly6C⁺Ly6G⁻ subpopulation have relatively low CCR5 expression. However, the CCR5 expression by lymphocytes and endothelial cells that are enriched in the Ly6C⁻Ly6G⁻ subpopulation was higher than the Ly6C⁺Ly6G⁻ (monocytes and macrophages) subpopulation. Of note, CCR5 expression by both of these subpopulations was significantly lower than neutrophils (Ly6C⁻Ly6G⁺) from the same parent gates. These data suggest that CCR5 expression by neutrophils is higher than in lymphocytes, which have higher expression than monocytes/macrophages (Suppl. Fig. 9).

To determine the effects of TGF β 1⁺ neutrophils on bone homeostasis, these cells were sorted and adoptively transferred into NSG immune-deficient mice. We found that the trabecular bone volume of NSG mice injected with TGF β 1⁺ neutrophils was significantly lower than in control mice, as presented in Fig. 3. In addition, the trabecular bone loss in NSG mice transplanted with TGF β 1⁺ neutrophils from aged mice was abolished when these mice were treated with maraviroc during the period after injection. Consistent with this, in Fig. 7 (Fig. 5 in original submitted version), 22-mon-old male C57 mice treated with maraviroc (10 mg/kg) once/day for 1 month had a significant decrease in TCNs and increased trabecular bone mass. We added a new paragraph describing these new data

on page 11,

“Maraviroc prevents bone loss caused by TCNs from aged mice.

To determine the effects of maraviroc on TCN recruitment into BM and bone homeostasis, we injected TCNs from BM of aged ROSA^{mTmG} male mice into NSG mice and treated the mice with vehicle or maraviroc. We found that NSG mice treated with maraviroc had significantly fewer donor-derived TCNs in BM than vehicle-treated mice (Suppl. Fig. 10a-d), which was not associated with altered Ki-67-related cell proliferation (Suppl. Fig. 10e-f), or p16-related cell senescence (Suppl. Fig. 10g-h). Of note, maraviroc-treated recipients had significantly higher trabecular bone volume (BV/TV; Fig. 6a-b) and trabecular number (Tb.N; Fig. 6c), lower trabecular separation (Tb.Sp; Fig. 6d), and higher bone mineral density (BMD; Fig. 6e) than vehicle-treated recipients. However, no significant changes were observed in the thickness of trabecular or cortical bone between NSG recipients treated with vehicle or maraviroc (Suppl. Fig. 10i-j). Histomorphometric analysis showed that maraviroc-treated recipients had higher osteoblast surfaces (Fig. 6f-g) and lower osteoclast numbers and surfaces on trabeculae in tibial metaphyses (Fig. 6f, h-i). These findings reveal that blockade of CCR5 by maraviroc efficiently prevented recruitment of donor-derived TCNs into BM and bone loss in NSG recipients.” Although the effects of maraviroc given to 22-mon-old mice may appear rather mild, they are nonetheless remarkable in our opinion since they involve effects to decrease bone resorption and increase bone formation, an effect that no current therapies for age-related osteoporosis have. Our new data showing that injection of TCNs into mice causes bone loss further strengthen our posit that these cells contribute significantly to age-related bone loss, which can be prevented by administration of maraviroc.

Figure 2: What happens to osteoblast differentiation when TGFb is blocked during MPC co-culture with neutrophils? The authors show that TGFbRII is needed on the MPC, but adding data on whether TGFb1 expression specifically by neutrophils is needed to suppress osteoblast differentiation in vitro or in vivo would improve the manuscript. ¹¹_{SEP}

In response to this suggestion, we carried out new experiments and added a statement in the Results on page 6, “To test if TGFβ-expressing neutrophils (TNs) regulate osteoblast differentiation directly and if this process functions through TGFβ, we sorted TNs (TGFβ1⁺CD11b⁺Ly6C⁺6G⁺) from 3- and 20-mon-old C57 male mice using FACS, and co-cultured them with MPCs from BM of 3-mon-old C57 mice. We found that TNs from aged mice more efficiently inhibited osteoblast differentiation than TNs from young mice and that this inhibition was blocked by 1D11, a TGFβ neutralizing antibody (Fig. 2d).”. Consistent with this, TGFβRII specific deletion in MPCs prevented inhibition of osteoblast differentiation caused by TNs. We added a description of these data in the Results on page 6, “We found that TNs sorted from aged C57 mice effectively inhibited osteoblast differentiation from WT MPCs, but not from TRII-ckO MPCs (Fig. 2e). In addition, TNs from aged mice stimulated osteoclast formation from progenitor cells with no changes in the ratio of *Rankl/Opg* transcription levels, and this was inhibited by 1D11 (Fig. 2f-g, Suppl. Fig. 1b).”. The experiments on ectopic implantation provide further *in vivo* evidence, as we presented in the Results on page 6, “We found that osteoblast differentiation and new bone formation

from WT bone-derived MPCs were significantly lower when they were co-implanted with TNs from aged than from young mice (Fig. 2h-i). Importantly, this inhibition of osteoblast differentiation caused by aged TNs was abolished when they were co-implanted with bone-derived MPCs from TRII-cKO mice, and these cells formed significantly more new bone (Fig. 2h-i).”.

These *in-vitro* and *in-vivo* findings support our proposal that TGFβ1 expression specifically by neutrophils suppresses osteoblast differentiation.

Figures 3 and especially 6 provide compelling *in vivo* results relating TGFβRII and TRAF3 expression by Prx1+ cells to bone mass. However, the authors also report that TGFβ1-expressing neutrophil numbers are increased or decreased in conditionally TGFβRII-deficient or TRAF3-deficient mice, respectively. These results are correlative and not causative; ie the authors have not shown that the numbers of TGFβ1-expressing neutrophils are functionally related to the bone phenotypes (it could be due to other reasons). How specific is Prx1-targeted gene deletion to MPC? Is TGFβRII/TRAF3 expression completely absent in MPC from these conditional knockout mice? Is TGFβRII/TRAF3 expression normal in non-MPC cells in the Bone marrow? Is TGFβRII deleted in other cells? The authors can also refer to previously published articles if these conditional knockout mice have been thoroughly characterised previously. In general, the manuscript would benefit from a bit more discussion on alternative explanations to their data.

Thanks for these suggestions. To determine if TGFβ1-expressing neutrophils (TNs) cause bone loss *in vivo*, we added new experiments using adoptive transfer of TNs from young and aged mice into immune-deficient NSG mice. The findings are presented in Fig. 3 and described on page 7,

“NSG mice injected with TGFβ1-expressing neutrophils develop osteoporosis.

To examine if TNs have systemic effects on bone homeostasis, we sorted TNs from BM of 3- and 20-mon-old male C57 mice and injected them into 3-mon-old NSG male mice via tail vein. We found that NSG recipients injected with either young or aged TNs developed significant bone loss (Fig. 3a). Of note, NSG recipients transplanted with aged TNs had significantly lower trabecular bone volume (Fig. 3b) and trabecular number (Fig. 3c) and increased trabecular separation (Fig. 3d) than NSG recipients transplanted with young TNs. However, no significant changes were observed in trabecular or cortical bone thickness (Suppl. Fig. 2a-b). Consistent with this, NSG recipients injected with either young or aged TNs had significantly fewer osteoblasts and more osteoclasts on trabecular bone surfaces than NSG mice without transplantation (Fig. 3e-h). NSG recipients transplanted with aged TNs had significantly more osteoclasts than recipients transplanted with young TNs (Fig. 3g-h).” These findings provide more compelling evidence that TNs cause bone loss in mice.

The Prx1-Cre mouse line is a powerful tool to delete loxp-sited target genes in mesenchymal lineage cells¹⁰, and we have previously reported the efficacy and specificity of targeting TRAF3 deletion driven by Prx1-Cre in mice (Li, et al. 2019 Nat. Commun.). In addition, we carried out new experiments to test the specificity of Prx1Cre in driving TRAF3 deletion in various cell types in bone marrow of Prx1^{Cre};TRAF3^{fl/fl} (P-cKO) mice. We

therefore added a statement in the Results on page 8, “We also found that Prx1Cre drove *Traf3* gene deletion in mesenchymal progenitor cells, but not in B cells, T cells, leukocytes and osteoclast precursor cells, in bone marrow of 15-mon-old P-cKO mice (Suppl. Fig. 5).”

To further verify the specificity of Prx1Cre-driven gene editing in mesenchymal lineage cells, we crossed Prx1^{Cre} with Rosa26^{mTmG} reporter mice, in which the Prx1^{Cre+} cells express GFP protein. The data are presented in the Results on page 8, “To examine the specificity of targeted gene deletion driven by Prx1Cre, we crossed Prx1^{Cre} with Rosa26^{mTmG} reporter mice, in which Prx1^{Cre+} cells switch tdTomato to GFP protein expression. We found that Prx1^{Cre+} GFP cells were very rare in skeletal muscle, but widely distributed in bone cells, including in hypertrophic chondrocytes in growth plates, mesenchymal/osteoblastic cells on metaphyseal bone surfaces, and a relatively small fraction of osteocytes and spindle cells in BM (Suppl. Fig. 4a-4c), which is consistent with the profiles of Prx1-tracing mesenchymal lineage cells reported by others⁴⁶. In addition, osteoblastic cells on surfaces of newly generated bone surfaces were GFP-positive in fracture callus of 3-m-old Prx1^{Cre};Rosa26^{mTmG} mice on day14 post-fracture, consistent with the osteogenic potential of the Prx1^{Cre}-tracing cells in our mouse model (Suppl. Fig. 4d). We also found significantly higher Cre transcription levels (~50-700x) in vertebrae of Prx1^{Cre} mice than in WT mice from 2 wk to 12-mon of age (Suppl. Fig. 4e). Many Prx1^{Cre+} GFP cells were present on endosteal and trabecular bones surfaces of 3-mon-old Prx1^{Cre};Rosa26^{mTmG} mice (Suppl. Fig. 4f).” These lineage tracing data verify the specificity of our Prx1Cre model.

Figure 6: title of figure and text appears incorrect. The title says that “Mice with TGFβRII deleted specifically in mesenchymal lineage cells have decreased numbers of TCNs in BM and increased bone loss”. However, the data shows a clear increase in bone volume and mass, as well as a decrease in osteoclasts, which is also discussed in the text.

Thank you spotting this error. We corrected the error as following: “Mice with TGFβRII deleted specifically in mesenchymal lineage cells have decreased numbers of TCNs in BM and increased bone mass”.

Minor comments

Figure 1A: stats between bone and BM cells. It is not exactly clear how these values were normalized.

We now have addressed this issue in the Methods on pages 21-22, “For sample preparation, BM cells were flushed from 1 tibia and 1 femur from each mouse with 250 μl PBS containing PIC (Roche, Cat #: 04693159001; 1 tablet in 10 ml PBS). BM cells were spun down immediately, and supernatant containing intracellular fluid was collected. Cortical bone was chopped and homogenized and immediately lysed in 250 μl T-per lysis buffer containing PIC. BM cells were also lysed in 250 μl T-per with PIC.”. On page 22, we added “Protein from bone, BMCs and BM plasma was extracted in a fixed volume (250 μl) of protein lysate; thus, values observed in ELISA tests were normalized by lysate volumes of the various samples.”.

Figure 1D: missing x and y axis labels

We added x and y axis labels.

Order of figures: the authors should present the panels for each figure in order (eg Fig 1e and f should be presented before Fig 1g). Could simply re-label the figures to match the text.

We have re-labeled the panels in Fig. 1 to match the text.

Fig 3f-g: it is not clear from the figure if these cell populations were pre-gated on TGFb1+ cells. Please clarify in the legend and figure.

In Fig. 3f-3g, Ly6C and Ly6G subpopulations were pre-gated on TGFβ1+CD11b+ cells. These figures and figure legends have been updated.

The use of acronyms: if only mentioned once, there is no need for an acronym, for example "LLCI". Also the use of many acronyms in the abstract makes it somewhat difficult to follow.

The acronym, LLCI, has been deleted. We edited acronyms in the Abstract accordingly, but it is difficult to stay close to the word limit without using them.

Use of dash in abstract (treat/prevent) should be avoided.

"/" has been replaced by "and".

Edit sentence in the introduction: "monocytes are myeloid lineage cells that express".... Instead more appropriate to refer to all myeloid cells, introduce the subsets, and then to specify whether it is known or unknown which ones make TGFb1 and that you sought to answer this question (TGFb1 expression is not a prominent feature that is used to define monocytes). Additionally, monocytes do consist of multiple subsets (Ly6Chigh vs low, etc) but these are not the focus of the paper and therefore may not need to be discussed in the introduction.

Thanks for bringing this to our attention. We revised the statement in the Introduction to read "Leukocytes are the most abundant cell type in BM and are heterogenous. As the most abundant sub-population of leukocytes, neutrophils express TGFβ1 in tumor cells²⁷, normal intestinal cells²⁸ and in respiratory cells in asthmatic subjects²⁹, but whether TGFβ1-expressing neutrophils or other immune cells are involved in inflammaging and age-related osteoporosis remains unclear."

References:

1. Lee, J. W.; Hoshino, A.; Inoue, K.; Saitou, T.; Uehara, S.; Kobayashi, Y.; Ueha, S.; Matsushima, K.; Yamaguchi, A.; Imai, Y.; Iimura, T., The HIV co-receptor CCR5 regulates osteoclast function. *Nat Commun* **2017**, *8* (1), 2226.
2. Chang, X. L.; Webb, G. M.; Wu, H. L.; Greene, J. M.; Abdulhaqq, S.; Bateman, K. B.; Reed, J. S.; Pessoa, C.; Weber, W. C.; Maier, N.; Chew, G. M.; Gilbride, R. M.; Gao, L.; Agnor, R.; Giobbi, T.; Torgerson, J.; Siess, D.; Burnett, N.; Fischer, M.; Shiel, O.; Moats,

- C.; Patterson, B.; Dhody, K.; Kelly, S.; Pourhassan, N.; Magnani, D. M.; Smedley, J.; Bimber, B. N.; Haigwood, N. L.; Hansen, S. G.; Brown, T. R.; Ndhlovu, L. C.; Sacha, J. B., Antibody-based CCR5 blockade protects Macaques from mucosal SHIV transmission. *Nat Commun* **2021**, *12* (1), 3343.
3. Nam, J. S.; Terabe, M.; Mamura, M.; Kang, M. J.; Chae, H.; Stuelten, C.; Kohn, E.; Tang, B.; Sabzevari, H.; Anver, M. R.; Lawrence, S.; Danielpour, D.; Lonning, S.; Berzofsky, J. A.; Wakefield, L. M., An anti-transforming growth factor beta antibody suppresses metastasis via cooperative effects on multiple cell compartments. *Cancer Res* **2008**, *68* (10), 3835-43.
4. Fridlender, Z. G.; Sun, J.; Kim, S.; Kapoor, V.; Cheng, G.; Ling, L.; Worthen, G. S.; Albelda, S. M., Polarization of tumor-associated neutrophil phenotype by TGF-beta: "N1" versus "N2" TAN. *Cancer Cell* **2009**, *16* (3), 183-94.
5. Chu, H. W.; Trudeau, J. B.; Balzar, S.; Wenzel, S. E., Peripheral blood and airway tissue expression of transforming growth factor beta by neutrophils in asthmatic subjects and normal control subjects. *J Allergy Clin Immunol* **2000**, *106* (6), 1115-23.
6. Stroo, I.; Stokman, G.; Teske, G. J.; Raven, A.; Butter, L. M.; Florquin, S.; Leemans, J. C., Chemokine expression in renal ischemia/reperfusion injury is most profound during the reparative phase. *Int Immunol* **2010**, *22* (6), 433-42.
7. Hartl, D.; Krauss-Etschmann, S.; Koller, B.; Hordijk, P. L.; Kuijpers, T. W.; Hoffmann, F.; Hector, A.; Eber, E.; Marcos, V.; Bittmann, I.; Eickelberg, O.; Griesse, M.; Roos, D., Infiltrated neutrophils acquire novel chemokine receptor expression and chemokine responsiveness in chronic inflammatory lung diseases. *J Immunol* **2008**, *181* (11), 8053-67.
8. Chen, F.; Yang, W.; Huang, X.; Cao, A. T.; Bilotta, A. J.; Xiao, Y.; Sun, M.; Chen, L.; Ma, C.; Liu, X.; Liu, C. G.; Yao, S.; Dann, S. M.; Liu, Z.; Cong, Y., Neutrophils Promote Amphiregulin Production in Intestinal Epithelial Cells through TGF-beta and Contribute to Intestinal Homeostasis. *J Immunol* **2018**, *201* (8), 2492-2501.
9. Lin, W.; Li, Q.; Zhang, D.; Zhang, X.; Qi, X.; Wang, Q.; Chen, Y.; Liu, C.; Li, H.; Zhang, S.; Wang, Y.; Shao, B.; Zhang, L.; Yuan, Q., Mapping the immune microenvironment for mandibular alveolar bone homeostasis at single-cell resolution. *Bone Res* **2021**, *9* (1), 17.
10. Duchamp de Lageneste, O.; Julien, A.; Abou-Khalil, R.; Frangi, G.; Carvalho, C.; Cagnard, N.; Cordier, C.; Conway, S. J.; Colnot, C., Periosteum contains skeletal stem cells with high bone regenerative potential controlled by Periostin. *Nat Commun* **2018**, *9* (1), 773.

REVIEWERS' COMMENTS

Reviewer #1 (Remarks to the Author):

In the revised manuscript, the authors have carefully and fully addressed the comments and concerns provided by the reviewers. This is an excellent work. I have no serious criticisms regarding this work.

Reviewer #2 (Remarks to the Author):

The authors satisfactorily addressed the concerns raised by the reviewers by performing additional experiments, and the manuscript was much improved.

Reviewer #4 (Remarks to the Author):

The authors have performed many additional experiments that have substantially strengthened the manuscripts, most specifically the adoptive transfer experiments of neutrophils from aged and young mice and the in vitro co-culture experiments with young vs old neutrophils and MPCs where the TGF β axis is blocked. The authors responded satisfactorily to this reviewer's comments. The manuscript could be published in its current state with a few minor clarifications:

The results presented in Fig. 3 and the supplementary data showing the inhibitory effect of aged versus young neutrophils on bone mass are impressive and support the authors' claims that neutrophils from aged mice induce bone loss. The authors state that the purity of the neutrophils transferred was over 90% and that they enriched for TGF β expressing neutrophils, but what was the TGF β expression level in the final sorted sample? This is a central experiment to support their claim that TGF β -expressing neutrophils, rather than neutrophils in general, inhibit bone growth, and therefore the authors should show the efficiency of the TGF β enrichment as well as the TGF β expression of the sorted TGF β -enriched sample vs the non-enriched. Alternatively, the authors could 'soften' these claims in the text.

Related to this experiment, it is unclear why the authors use NSG mice as the recipients of the neutrophil transfer experiments. Why not use WT mice of the same background? This reviewer appreciates that the adoptive transfer experiments using fluorescent reporters may cause rejection due to lymphocyte reactivity against the fluorescent molecules, but it would be beneficial to discuss potential caveats in the text. For example, are NSG mice more susceptible to bone loss? What would happen in an immunocompetent setting?

Fig. 2a: at the ratio of 0:1 neutrophils to pre-osteoblasts, there is already a difference between the groups with neutrophils from 3 vs 21 months? I interpret this as no neutrophils present, yet there is a difference in the pre-osteoblast ALP staining. Why is this the case? Please clarify.

In Fig. S9, what do the numbers in the separate panels labeled Q1-4 represent? One would expect them to add up to 100% or be higher based on the gating. Please clarify in the legend and/or figure.

All in all, the authors should be commended for the thorough response to all the reviewers and for their intriguing work that warrants publication.

REVIEWERS' COMMENTS

Reviewer #1 (Remarks to the Author):

In the revised manuscript, the authors have carefully and fully addressed the comments and concerns provided by the reviewers. This is an excellent work. I have no serious criticisms regarding this work.

Reviewer #2 (Remarks to the Author):

The authors satisfactorily addressed the concerns raised by the reviewers by performing additional experiments, and the manuscript was much improved.

Reviewer #4 (Remarks to the Author):

The authors have performed many additional experiments that have substantially strengthened the manuscripts, most specifically the adoptive transfer experiments of neutrophils from aged and young mice and the in vitro co-culture experiments with young vs old neutrophils and MPCs where the TGFB axis is blocked. The authors responded satisfactorily to this reviewer's comments. The manuscript could be published in its current state with a few minor clarifications:

The results presented in Fig. 3 and the supplementary data showing the inhibitory effect of aged versus young neutrophils on bone mass are impressive and support the authors' claims that neutrophils from aged mice induce bone loss. The authors state that the purity of the neutrophils transferred was over 90% and that they enriched for TGFB expressing neutrophils, but what was the TGFB expression level in the final sorted sample? This is a central experiment to support their claim that TGFB-expressing neutrophils, rather than neutrophils in general, inhibit bone growth, and therefore the authors should show the efficiency of the TGFB enrichment as well as the TGFB expression of the sorted TGFB-enriched sample vs the non-enriched. Alternatively, the authors could 'soften' these claims in the text.

Thanks for raising this issue for clarification. In the process of cell sorting, we first enriched TGF β 1⁺ BMCs using magnetic-activated cell sorting (MACS) and increased the percentage of TGF β 1⁺ cells by ~4-fold. Fluorescence-activated cell sorting (FACS) was next applied to sort CD11b⁺Ly6G⁺ cells from enriched TGF β 1⁺ BMCs. After positive selection, over 90% of the cells we finally harvested were TGF β 1-expressing neutrophils (TNs). We next compared the TGF β 1 expression by these enriched TNs with Ly6G⁺ neutrophils. We found that enriched TCNs had 4.3-fold higher TGF β 1 MFI (mean of fluorescence intensity) than non-enriched neutrophils. We added these data to Supplementary Fig. 3 and added this statement to page 8, "To examine if TNs have

systemic effects on bone homeostasis, we flow sorted TNs from BM of 3- and 20-mon-old male C57 mice and confirmed that they have high expression of TGF β , with MFI levels being 4.3-fold higher than in non-enriched neutrophils (Suppl. Fig. 3a-d)".

In addition, we measured TGF β 1 expression by sorted TCNs and non-enriched neutrophils and added these new data as Supplementary Fig. 10. We added this statement on page 12, "To determine the effects of maraviroc on TCN recruitment into BM and bone homeostasis, we sorted TCNs from BM of aged ROSA^{mTmG} male mice and confirmed their high expression of TGF β 1 (Suppl. Fig. 10). We injected these sorted TCNs into NSG mice and treated the mice with vehicle or maraviroc."

We also added this statement in the Methods section on page 22,

"NSG mice injected with sorted TGF β -expressing neutrophils (TNs) and TGF β /CCR5-expressing neutrophils (TCNs)

3- and 20-22-mon-old C57 and ROSA^{mTmG} male mice were sacrificed for isolation of TNs and TCNs from BM. TGF β 1-expressing BM cells were first enriched through positive selection using a magnetic isolation system with PE-conjugated anti-TGF β 1 Ab as primary and anti-PE MicroBeads as secondary labeling. Ly6G-positive cells or CCR5/Ly6G double-positive cells were further sorted by BD FACSAria II sorting equipment from the positively-selected TGF β 1-expressing BM cells. The purity of TNs and TCNs in post-sort fractions was over 90%, which was confirmed using a BD FACS LSRII cytometer, and the mean fluorescence intensity (MFI) of TGF β 1 expression by these enriched TNs and TCNs was over 4-fold higher than Ly6G⁺ neutrophils."

Related to this experiment, it is unclear why the authors use NSG mice as the recipients of the neutrophil transfer experiments. Why not use WT mice of the same background? This reviewer appreciates that the adoptive transfer experiments using fluorescent reporters may cause rejection due to lymphocyte reactivity against the fluorescent molecules, but it would be beneficial to discuss potential caveats in the text. For example, are NSG mice more susceptible to bone loss? What would happen in an immunocompetent setting?

Increasing evidence suggests that neutrophils, the most abundant innate immune cells, are antigen-presenting cells with respect to activating adaptive immune cells, especially T cells¹. In general, to study the functions of immune cells, these cells could be transplanted into same background WT recipients after lethal or lower dose radiation. Our understanding is that either activation of adaptive immune cells^{2, 3} or radiation⁴ is associated with bone loss, which would increase the complexity of the approach if we planned to transplant specific neutrophils to determine the direct effects of these cells on bone metabolism. To avoid this complexity, we used NSG mice as recipients to study the function of neutrophils isolated from various mice on bone metabolism for the following reasons:

- (1) In Fig. 2, we performed an in-vivo ectopic bone formation assay in which NSG mice were ideal recipients for injecting the mixture of neutrophils and mesenchymal progenitor cells from donor mice. To further verify the in-vivo effects of these neutrophils on bone metabolism systemically, NSG mice also used as recipients in

the experiments on TGF β 1-expressing neutrophil injection, the results of which are presented in Fig. 3. This consistent use of NSG mice as recipients helped to determine the effects of TGF β 1-expressing neutrophils on local and systemic bone metabolism.

- (2) In addition, to determine if these specific neutrophils injected into the circulation of recipients were recruited into bone marrow and if they were in proliferative or apoptotic status (Fig. S10), mTmG reporter mice were a powerful tool for this purpose, and therefore neutrophils isolated from these reporter mice were used in this study. As the reviewer mentioned, in order to minimize lymphocyte reactivity against the fluorescent molecules, as well as avoid potential bone loss caused by adaptive immune cell activation or irradiation in C57 recipients, we used NSG mice that lack of adaptive immune responses as recipients and mTmG mice as donors, which enabled us to track and determine the fate of transplanted neutrophils.
- (3) To our best knowledge, there is no direct published evidence reporting that NSG mice are more susceptible to bone loss. This would require further investigation. The NSG mouse line was generated by depletion of the gamma chain of the interleukin 2 receptor (*Il2rg*) in NOD/SCID mice, which have loss of the functionally mutated *Prkdc* gene that prevents T and B cell maturation. Song et al. reported that glucocorticoid-induced osteoporosis (GIOP) could not be induced in SCID mice that lack T cells, but it could be induced by adoptive transfer of splenic T cells from wild-type mice⁵. This study suggests that SCID mice and presumably NSG mice could be more resistance to pathological bone loss that is mediated by adaptive immunity.

Gene modifications in NSG mice disable B, T and natural killer (NK) cells, but not neutrophils; however, it is unclear if NSG mice are more susceptible to bone loss than wildtype mice with normal immune responses, in an immunocompetent setting. Some studies used humanized NSG mice to study effects of chronic *S. aureus* infection on bone, but these mice had established functional immune responses through transplantation of human hematopoietic stem cells before infection⁶. Given the deficiency of B and T cells in NSG mice, NSG mice probably are more resistance to bone loss, if the immunocompetent setting causes activation of certain type of inflammatory B and T cells, such as RANKL⁺ B cells⁷, TNFa⁺ T cells⁸ and CD40L⁺ T cells⁹. To address this issue, we added this statement in the Methods section on page 22, "We could have injected TNs or TCNs into same background WT recipients after lethal or lower dose radiation, rather than into NSG mice. However, increasing evidence suggests that neutrophils are antigen-presenting cells, which can activate adaptive immune cells, especially T cells⁶⁹, and activation of adaptive immune cells⁷⁰ or radiation⁷¹ are associated with bone loss, which would have increased the complexity of the approach."

Fig. 2a: at the ratio of 0:1 neutrophils to pre-osteoblasts, there is already a difference between the groups with neutrophils from 3 vs 21 months? I interpret this as no neutrophils present, yet there is a difference in the pre-osteoblast ALP staining. Why is

this the case? Please clarify.

It is correct that there were no neutrophils present in the co-culture at the ratio of 0:1 neutrophils:pre-osteoblasts and that there is less ALP staining of osteoblasts differentiated from pre-osteoblasts from 21-month-old mice than from 3-month-old mice. This difference in ALP expression indicates impaired osteoblast differentiation from pre-osteoblasts in 21-m-old than 3-m-old mice. In line with our finding here, other studies have reported this decline of osteoblast differentiation from aged mesenchymal progenitor cells^{10, 11}.

We added this comment on page 7 to clarify this point. "We found that osteoblasts derived from aged mice had lower alkaline phosphatase activity than those derived from young mice when cultured in the absence of neutrophils (Fig 2a), consistent with previous reports that MSCs from aged mice have reduced osteoblastic potential."

In Fig. S9, what do the numbers in the separate panels labeled Q1-4 represent? One would expect them to add up to 100% or be higher based on the gating. Please clarify in the legend and/or figure.

Thanks for pointing this out. The numbers in the separate Q1-4 panels in Fig. S9 represent the percentages of the gated populations in all cells of each Q1-4 panel. In Q1-4 panels, "log" axis was applied to show the clearly separated Ly6G and Ly6C subpopulations, and consequently most Ly6C-6G- double-negative cells and debris were hidden beyond the chart edges (as illustrated in the revised figure). Thus, the sum of the three percentage numbers was lower than 100%, which did not affect the accuracy of the results.

We added this comment in the figure legend, "The numbers represent the percentages of the gated populations in all cells of each panel (the middle row). In Q1-4 panels, "log" axis was applied to show the clearly separated Ly6G and Ly6C subpopulations, and consequently various types of cells were hidden beyond the chart edges (indicated by the yellow arrows)."

All in all, the authors should be commended for the thorough response to all the reviewers and for their intriguing work that warrants publication.

1. Li, Y. *et al.* The regulatory roles of neutrophils in adaptive immunity. *Cell Commun Signal* **17**, 147 (2019).
2. Wu, D. *et al.* T-Cell Mediated Inflammation in Postmenopausal Osteoporosis. *Front Immunol* **12**, 687551 (2021).
3. Komatsu, N. *et al.* Plasma cells promote osteoclastogenesis and periarticular bone loss in autoimmune arthritis. *J Clin Invest* **131** (2021).
4. Kielbassa, A.M., Hinkelbein, W., Hellwig, E. & Meyer-Luckel, H. Radiation-related damage to dentition. *Lancet Oncol* **7**, 326-335 (2006).

5. Song, L. *et al.* The critical role of T cells in glucocorticoid-induced osteoporosis. *Cell Death Dis* **12**, 45 (2020).
6. Muthukrishnan, G. *et al.* Humanized Mice Exhibit Exacerbated Abscess Formation and Osteolysis During the Establishment of Implant-Associated *Staphylococcus aureus* Osteomyelitis. *Front Immunol* **12**, 651515 (2021).
7. Meednu, N. *et al.* Production of RANKL by Memory B Cells: A Link Between B Cells and Bone Erosion in Rheumatoid Arthritis. *Arthritis Rheumatol* **68**, 805-816 (2016).
8. Yu, M. *et al.* Ovariectomy induces bone loss via microbial-dependent trafficking of intestinal TNF+ T cells and Th17 cells. *J Clin Invest* **131** (2021).
9. Li, J.Y. *et al.* Ovariectomy dysregulates osteoblast and osteoclast formation through the T-cell receptor CD40 ligand. *Proc Natl Acad Sci U S A* **108**, 768-773 (2011).
10. Farr, J.N. *et al.* Targeting cellular senescence prevents age-related bone loss in mice. *Nat Med* **23**, 1072-1079 (2017).
11. Li, H. *et al.* FOXP1 controls mesenchymal stem cell commitment and senescence during skeletal aging. *J Clin Invest* **127**, 1241-1253 (2017).